# Optimal Gradient-based Algorithms for Non-concave Bandit Optimization

**Baihe Huang**[1*], **Kaixuan Huang**[2*], **Sham M. Kakade**[3,4*], **Jason D. Lee**[2*]
**Qi Lei**[2*], **Runzhe Wang**[2*], **Jiaqi Yang**[5*]

[1]Peking University   [2]Princeton University   [3]University of Harvard
[4]Microsoft Research   [5]Tsinghua University

## Abstract

Bandit problems with linear or concave reward have been extensively studied, but relatively few works have studied bandits with non-concave reward. This work considers a large family of bandit problems where the unknown underlying reward function is non-concave, including the low-rank generalized linear bandit problems and two-layer neural network with polynomial activation bandit problem. For the low-rank generalized linear bandit problem, we provide a minimax-optimal algorithm in the dimension, refuting both conjectures in [55, 43]. Our algorithms are based on a unified zeroth-order optimization paradigm that applies in great generality and attains optimal rates in several structured polynomial settings (in the dimension). We further demonstrate the applicability of our algorithms in RL in the generative model setting, resulting in improved sample complexity over prior approaches. Finally, we show that the standard optimistic algorithms (e.g., UCB) are sub-optimal by dimension factors. In the neural net setting (with polynomial activation functions) with noiseless reward, we provide a bandit algorithm with sample complexity equal to the intrinsic algebraic dimension. Again, we show that optimistic approaches have worse sample complexity, polynomial in the extrinsic dimension (which could be exponentially worse in the polynomial degree).

## 1   Introduction

Bandits [51] are a class of online decision-making problems where an agent interacts with the environment, only receives a scalar reward, and aims to maximize the reward. In many real-world applications, bandit and RL problems are characterized by large or continuous action space. To encode the reward information associated with the action, function approximation for the reward function is typically used, such as linear bandits [20]. Stochastic linear bandits assume the mean reward to be the inner product between the unknown model parameter and the feature vector associated with the action. This setting has been extensively studied, and algorithms with optimal regret are known [20, 51, 2, 13].

However, linear bandits suffer from limited representation power unless the feature dimension is prohibitively large. A comprehensive empirical study [60] found that real-world problems required non-linear models and thus non-concave rewards to attain good performance on a testbed of bandit problems. To take a step beyond the linear setting, it becomes more challenging to design optimal algorithms. Unlike linear bandits, more sophisticated algorithms beyond optimism are necessary. For instance, a natural first step is to look at quadratic [44] and higher-order polynomial [35] reward. In

---

*Alphabetical order. Correspondence to: Qi Lei, `qilei@princeton.edu`, Jason D. Lee, `jasonlee@princeton.edu`.

the context of phase retrieval, which is a special case for the quadratic bandit, people have derived algorithms that achieve minimax risks in the statistical learning setting [9, 52, 15]. However, the straightforward adaptation of these algorithms results in sub-optimal dimension dependency.

In the bandit domain, existing analysis on the nonlinear setting includes eluder dimension [62], subspace elimination [55, 43], etc. Their results also suffer from a larger dimension dependency than the best known lower bound in many settings. (See Table 1 and Section 3 for a detailed discussion of these results.) Therefore in this paper, we are interested in investigating the following question:

> *What is the optimal regret for non-concave bandit problems, including structured polynomials (low-rank etc.)? Can we design algorithms with optimal dimension dependency?*

**Contributions:** In this paper, we answer the questions and close the gap (in problem dimension) for various non-linear bandit problems.

1. First, we design stochastic zeroth-order gradient-like[2] ascent algorithms to attain minimax regret for a large class of structured polynomials. The class of structured polynomials contains bilinear and low-rank linear bandits and symmetric and asymmetric higher-order homogeneous polynomial bandits with action dimension $d$. Though the reward is non-concave, we combine techniques from two bodies of work, non-convex optimization and numerical linear algebra, to design robust gradient-based algorithms that converge to global maxima. Our algorithms are also computationally efficient, practical, and easily implementable.
   In all cases, our algorithms attain the optimal dependence on dimension $d$, which was not previously attainable using existing optimism techniques. As a byproduct, our algorithm refutes[3] the conjecture from [43] on the bilinear bandit, and the conjecture from [55] on low-rank linear bandit by giving an algorithm that attains the optimal dimension dependence.
2. We demonstrate that our techniques for non-concave bandits extend to RL in the generative setting, improving upon existing optimism techniques.
3. When the reward is a general polynomial without noise, we prove that solving polynomial equations achieves regret equal to the intrinsic algebraic dimension of the underlying polynomial class, which is often linear in $d$ for interesting cases. In general, this complexity cannot be further improved.
4. Furthermore, we provide a lower bound showing that all UCB algorithms have a sample complexity of $\Omega(d^p)$, where $p$ is the degree of the polynomial. The dimension of *all homogeneous polynomials* of degree $p$ in dimension $d$ is $d^p$, showing that UCB is oblivious to the polynomial class and highly sub-optimal even in the noiseless setting.

## 1.1 Related Work

**Linear Bandits.** Linear bandit problems and their variants are studied in [20, 51, 1, 13, 2, 20, 61, 34, 33]. The matching upper bound and minimax lower bound achieves $O(d\sqrt{T})$ regret. Structured linear bandits, including sparse linear bandit [3] which developed an online-to-confidence-set technique. This technique yields the optimal $O(\sqrt{sdT})$ rate for sparse linear bandit. However [55] employed the same technique for low-rank linear bandits giving an algorithm with regret $O(\sqrt{d^3\mathrm{poly}(k)T})$ which we improve to $O(\sqrt{d^2\mathrm{poly}(k)T})$, which meets the lower bound given in [55].

**Eluder Dimension.** [62] proposed the eluder dimension as a general complexity measure for nonlinear bandits. However, the eluder dimension is only known to give non-trivial bounds for linear function classes and monotone functions of linear function classes. For structured polynomial classes, the eluder dimension simply embeds into an ambient linear space of dimension $d^p$, where $d$ is the dimension and $p$ is the degree. This parallels the linearization/NTK line in supervised learning [71, 30, 7] which show that linearization also incurs a similarly large penalty of $d^p$ sample complexity, and more advanced algorithm design is need to circumvent linearization [10, 17, 25, 72, 27, 59, 29, 57, 36, 68, 19].

**Non-concave Bandits.** To our knowledge, there is no general study of non-concave bandits, likely due to the difficulty of globally maximizing non-concave functions. A natural starting point of

---

[2]Our algorithm estimates the gradient, but with some irreducible bias for the tensor case. Note that our algorithms converge linearly despite the bias.

[3]Both papers conjectured regret of the form $O(\sqrt{d^3\mathrm{poly}(k)T})$ based on convincing but potentially misleading heuristics.

studying the non-concave setting are quadratic rewards such as the Rayleigh quotient, or namely bandit PCA [47, 28]. In the bilinear [43] and low-rank linear setting [55] with rank $k$ parameter matrices achieved $\widetilde{O}(\sqrt{d^3\mathrm{poly}(k)T})$ regret. Other literature considered related but different settings that are not comparable to our results [44, 42, 32, 48]. We note that the regret of all previous work is at least $O(\sqrt{d^3T})$. This includes the subspace exploration and refinement algorithms from [43, 55], or from eluder dimension [62]. Recently [35] considers online problem with a low-rank tensor associated with axis-aligned set of arms, which corresponds to finding the largest entry of the tensor. Finally, [47] study the bandit PCA problem in the adversarial setting, attaining regret of $O(\sqrt{d^3T})$. We leave adapting our results to the adversarial setting as an open problem.

Due to space limitation we defer the discussion of some equally important literature to the appendix.

## 2 Preliminaries

### 2.1 Setup: Structured Polynomial Bandit

We study structured polynomial bandit problems where the reward function is from a class of structured polynomials (see below). A player plays actions for $T$ rounds; at each round $t \in [T]$, the player chooses one action $\boldsymbol{a}_t$ from the feasible action set $\mathcal{A}$ and receives the reward $r_t$ afterward.

We consider both the stochastic case where $r_t = f_{\boldsymbol{\theta}}(\boldsymbol{a}_t) + \eta_t$ where $\eta_t$ is the random noise, and the noiseless case $r_t = f_{\boldsymbol{\theta}}(\boldsymbol{a}_t)$. Specifically the function $f_{\boldsymbol{\theta}}$ is unknown to the player, but lies in a known function class $\mathcal{F}$. We use the notation $\mathrm{vec}(\boldsymbol{M})$ to denote the vectorization of a matrix or a tensor $\boldsymbol{M}$, and $\boldsymbol{v}^{\otimes p}$ to denote the $p$-order tensor product of a vector $\boldsymbol{v}$. For vectors we use $\|\cdot\|_2$ or $\|\cdot\|$ to denote its $\ell_2$ norm. For matrices $\|\cdot\|_2$ or $\|\cdot\|$ stands for its spectral norm, and $\|\cdot\|_F$ is Frobenius norm. For integer $n$, $[n]$ denotes set $\{1, 2, \cdots n\}$. For cleaner presentation, in the main paper we use $\widetilde{O}$, $\widetilde{\Theta}$ or $\widetilde{\Omega}$ to hide universal constants, polynomial factors in $p$ and polylog factors in dimension $d$, error $\varepsilon$, eigengap $\Delta$, total round number $T$ or failure rate $\delta$.

We now present the outline with the settings considered in the paper:

**The stochastic bandit eigenvector case**, $\mathcal{F}_{\mathrm{EV}}$, considers action set $\mathcal{A} = \{\boldsymbol{a} \in \mathbb{R}^d : \|\boldsymbol{a}\|_2 \leq 1\}$ as shown in Section 3.1, and

$$\mathcal{F}_{\mathrm{EV}} = \left\{ \begin{array}{l} f_{\boldsymbol{\theta}}(\boldsymbol{a}) = \boldsymbol{a}^T \boldsymbol{M} \boldsymbol{a}, \boldsymbol{M} = \sum_{j=1}^k \lambda_j \boldsymbol{v}_j \boldsymbol{v}_j^\top, \text{ for orthonormal } \boldsymbol{v}_j \\ \boldsymbol{M} \in \mathbb{R}^{d \times d}, 1 \geq \lambda_1 \geq |\lambda_2| \geq \cdots \geq |\lambda_k| \end{array} \right\}.$$

**The stochastic low-rank linear reward case**, $\mathcal{F}_{\mathrm{LR}}$ considers action sets on bounded matrices $\mathcal{A} = \{\boldsymbol{A} \in \mathbb{R}^{d \times d} : \|\boldsymbol{A}\|_{\mathrm{F}} \leq 1\}$ in Section 3.2, and

$$\mathcal{F}_{\mathrm{LR}} = \left\{ \begin{array}{l} f_{\boldsymbol{\theta}}(\boldsymbol{A}) = \langle \boldsymbol{M}, \boldsymbol{A} \rangle = \mathrm{vec}(\boldsymbol{M})^\top \mathrm{vec}(\boldsymbol{A}), \\ \boldsymbol{M} \in \mathbb{R}^{d \times d}, \mathrm{rank}(\boldsymbol{M}) = k, \boldsymbol{M} = \boldsymbol{M}^\top, \|\boldsymbol{M}\|_{\mathrm{F}} \leq 1 \end{array} \right\}.$$

We illustrate how to apply the established bandit oracles to attain a better sample complexity for RL problems with the simulator in Section 3.2.1.

**The stochastic homogeneous polynomial reward case** is presented in Section 3.3. For the symmetric case in Section 3.3.1, the action sets are $\mathcal{A} = \{\boldsymbol{a} \in \mathbb{R}^d : \|\boldsymbol{a}\|_2 \leq 1\}$, and

$$\mathcal{F}_{\mathrm{SYM}} = \left\{ \begin{array}{l} f_{\boldsymbol{\theta}}(\boldsymbol{a}) = \sum_{j=1}^k \lambda_j (\boldsymbol{v}_j^\top \boldsymbol{a})^p \text{ for orthonormal } \boldsymbol{v}_j, \\ 1 \geq r^* = \lambda_1 > |\lambda_2| \geq \cdots \geq |\lambda_k| \end{array} \right\};$$

in the asymmetric case in Section D.3.1, the action sets are
$\mathcal{A} = \{\boldsymbol{a} = \boldsymbol{a}(1) \otimes \boldsymbol{a}(2) \otimes \cdots \otimes \boldsymbol{a}(p) \in \mathbb{R}^{d^p} : \forall q \in [p], \|\boldsymbol{a}(q)\|_2 \leq 1\}$, and

$$\mathcal{F}_{\mathrm{ASYM}} = \left\{ \begin{array}{l} f_{\boldsymbol{\theta}}(\boldsymbol{a}) = \sum_{j=1}^k \lambda_j \prod_{q=1}^p (\boldsymbol{v}_j(q)^\top \boldsymbol{a}(q)) \text{ for orthonormal } \boldsymbol{v}_j(q) \text{ for each } q, \\ 1 \geq r^* = |\lambda_1| \geq |\lambda_2| \geq \cdots \geq |\lambda_k| \end{array} \right\}.$$

In the above settings, there is stochastic noise on the observed rewards. We also consider noiseless settings as below:

**The noiseless polynomial reward case** is presented in Section 3.4. The action sets $\mathcal{A}$ are subsets of $\mathbb{R}^d$, and $\mathcal{F}_{\mathrm{P}} = \{f_{\boldsymbol{\theta}}(\boldsymbol{a}) = \langle \boldsymbol{\theta}, \widetilde{\boldsymbol{a}}^{\otimes p} \rangle : \boldsymbol{\theta} \in \mathcal{V}, \widetilde{\boldsymbol{a}} = [1, \boldsymbol{a}^\top]^\top, \mathcal{V} \subseteq (\mathbb{R}^{d+1})^{\otimes p}$ is an algebraic variety$\}$.

| | | | $\mathcal{F}_{\mathrm{HPS}}$ | $\mathcal{F}_{\mathrm{HPA}}$ | $\mathcal{F}_{\mathrm{B}}$ | $\mathcal{F}_{\mathrm{LL}}$ |
|---|---|---|---|---|---|---|
| Upper Bound from LinUCB/eluder | | | $\widetilde{O}(d^{p+1}k\epsilon^{-2})(k>p)$ | $\widetilde{O}(d^{p+1}k\epsilon^{-2})$ | $\widetilde{O}(d^3k\epsilon^{-2})$ | $\widetilde{O}(d^3k\epsilon^{-2})$ |
| Our Results | NPM | Gap-dependent | $\widetilde{O}(\kappa d^p k\Delta^{-2}\epsilon^{-2})$ (even $p$) | N/A | $\widetilde{O}(\kappa d^2\Delta^{-2}\epsilon^{-2})$ | $\widetilde{O}(d^2k^2\lambda_k^{-2}\epsilon^{-2})$ |
| | | Gap-independent | $\widetilde{O}(d^p k\epsilon^{-2})$ (odd $p$) | $\widetilde{O}((Ck)^p d^p\epsilon^{-2})$ | $\widetilde{O}(\min\{d^2\epsilon^{-5}, d^2k^5\epsilon^{-2}\})$ | N/A |
| | | Lower bounds | $\Omega(d^p\epsilon^{-2})$ | $\Omega(d^p\epsilon^{-2})$ | $\Omega(d^2\epsilon^{-2})$ | $\Omega(d^2k^2\epsilon^{-2})^*$ |

Table 1: Baselines and Our Main Results (for stochastic settings). Eigengap $\Delta = \lambda_1 - |\lambda_2|$, $\kappa = \lambda_1/\Delta$. $C$ is a constant that depends on failure probability. The result with $*$ is from [55].

Additionally, $\mathcal{F}$ needs to be admissible (Definition 3.20). This class includes two-layer neural networks with polynomial activations (i.e. structured polynomials). We study the fundamental limits of all UCB algorithms in Section G.0.1 as they are $\Omega(d^{p-1})$ worse than our algorithm presented in Section 3.4.1.

In the above settings, we are concerned with the cumulative regret $\mathfrak{R}(T)$ for $T$ rounds. Let $f_{\boldsymbol{\theta}}^* = \sup_{\boldsymbol{a}\in\mathcal{A}} f_{\boldsymbol{\theta}}(\boldsymbol{a})$, $\mathfrak{R}_{\boldsymbol{\theta}}(T) := \sum_{t=1}^{T}(f_{\boldsymbol{\theta}}^* - f_{\boldsymbol{\theta}}(\boldsymbol{a}_t))$. Since the parameters can be chosen adversarially, we are bounding $\mathfrak{R}(T) = \sup_{\boldsymbol{\theta}} \mathfrak{R}_{\boldsymbol{\theta}}(T)$ in this paper. In all the stochastic settings above, we make the standard assumption on stochasticity that $\eta_t$ is conditionally zero-mean 1-sub-Gaussian random variable regarding the randomness before $t$.

## 2.2 Warm-up: Adapting Existing Algorithms

In all of the above settings, the function class can be viewed as a generalized linear function. Namely, there is fixed feature maps $\psi, \phi$ so that $f_{\boldsymbol{\theta}}(\boldsymbol{a}) = \psi(\boldsymbol{\theta})^T \phi(\boldsymbol{a})$. Thus it is straightforward to adapt linear bandit algorithms like the renowned LinUCB [53] to our settings. Furthermore, another baseline is given by the eluder dimension argument [62][67] which gives explicit upper bounds for general function classes. We present the best upper bound by adapting these methods as a baseline in Table 1, together with our newly-derived lower bound and upper bound in this paper.
The best-known statistical rates are based on the following result.

**Theorem 2.1** (Proposition 4 in [62]). *With $\alpha = O(T^{-2})$ appropriately small, given the $\alpha$-covering-number $N$ (under $\|\cdot\|_\infty$) and the $\alpha$-eluder-dimension $d_E$ of the function class $\mathcal{F}$, Eluder UCB (Algorithm 2) achieves regret $\widetilde{O}(\sqrt{d_E T \log N})$.*

In the first row of Table 1, we further elaborate on the best results obtained from Theorem 2.1 in individual settings. More details can be found in Appendix B.

## 3 Main results

We now present our main results. We consider 4 different stochastic settings (see Table 1) and one noiseless setting with structured polynomials.
In the cases of stochastic reward, all our algorithms can be unified as gradient-based optimization. At each stage with a candidate action $\boldsymbol{a}$, we define the estimator $G_n(\boldsymbol{a}) := \frac{1}{n}\sum_{i=1}^{n}(f_{\boldsymbol{\theta}}((1-\zeta)\boldsymbol{a}+\zeta\boldsymbol{z}_i)+\eta_i)\boldsymbol{z}_i$, with $\boldsymbol{z}_i \sim \mathcal{N}(0, \sigma^2 \boldsymbol{I}_d)$ and proper step-size $\zeta$ [26]. Therefore $\mathbb{E}_{\boldsymbol{z}}[G(\boldsymbol{a})] = \sigma^2\zeta\nabla f_{\boldsymbol{\theta}}((1-\zeta)\boldsymbol{a})+O(\zeta^2) = \zeta(1-\zeta)^{p-1}\sigma^2\nabla f_{\boldsymbol{\theta}}(\boldsymbol{a})+O(\zeta^2)$ for $p$-th order homogeneous polynomials. Therefore with enough samples, we are able to implement noisy gradient ascent with bias.
In the noiseless setting, our algorithm solves for the parameter $\boldsymbol{\theta}$ with randomly sampled actions $\{\boldsymbol{a}_t\}$ and the noiseless reward $\{f_{\boldsymbol{\theta}}(\boldsymbol{a}_t)\}$, and then determines the optimal action by computing $\arg\max_{\boldsymbol{a}} f_{\boldsymbol{\theta}}(\boldsymbol{a})$.

## 3.1 Stochastic Eigenvalue Reward ($\mathcal{F}_{\mathbf{EV}}$)

Now consider bandits with stochastic reward $r(\boldsymbol{a}) = \boldsymbol{a}^\top \boldsymbol{M}\boldsymbol{a} + \eta$ with action set $\mathcal{A} = \{\boldsymbol{a}|\|\boldsymbol{a}\|_2 \leq 1\}$. $\boldsymbol{M} = \sum_{i=1}^{r} \lambda_i \boldsymbol{v}_i \boldsymbol{v}_i^\top$ is symmetric and satisfies $r^* = \lambda_1 > |\lambda_2| \geq |\lambda_3| \cdots \geq |\lambda_r|, \eta \sim \mathcal{N}(0,1)$. Denote by $\boldsymbol{a}^*$ the optimal action ($\pm\boldsymbol{v}_1$), the leading eigenvector of $\boldsymbol{M}$, $(\boldsymbol{a}^*)^\top \boldsymbol{M}\boldsymbol{a}^* = \lambda_1$. Let $\Delta = \lambda_1 - |\lambda_2| > 0$ be the eigengap and $\kappa := \lambda_1/\Delta$ be the condition number.

**Remark 3.1** (Negative leading eigenvalue). *For a symmetric matrix $\boldsymbol{M}$, we will conduct noisy power method to recover its leading eigenvector, and therefore we require its leading eigenvalue $\lambda_1$*

*to be positive. It is straightforward to extend to the setting where the nonzero eigenvalues satisfy:*
$r^* \equiv \lambda_1 > \lambda_2 \geq \lambda_l > 0 > \lambda_{l+1} \cdots \geq \lambda_k$, *and* $|\lambda_k| > \lambda_1$. *For this problem, we can shift $M$ to get*
$M + |\lambda_k|I$ *and the eigen-spectrum now becomes* $\lambda_1 + |\lambda_k|, \lambda_2 + |\lambda_k|, \cdots 0$; *therefore, we can still recover the optimal action with dependence on the new condition number* $(\lambda_1 + |\lambda_k|)/(\lambda_1 - \lambda_2)$.

**Remark 3.2** (Asymmetric matrix). *Our algorithm naturally extends to the asymmetric setting:*
$f(\boldsymbol{a}_1, \boldsymbol{a}_2) = \boldsymbol{a}_1^\top \widetilde{\boldsymbol{M}} \boldsymbol{a}_2$, *where* $\widetilde{\boldsymbol{M}} = \boldsymbol{U}\boldsymbol{\Sigma}\boldsymbol{V}^\top$. *This setting can be reduced to the symmetric case via defining*

$$\boldsymbol{M} = \begin{bmatrix} 0 & \widetilde{\boldsymbol{M}}^\top \\ \widetilde{\boldsymbol{M}} & 0 \end{bmatrix} = \frac{1}{2}\begin{bmatrix} \boldsymbol{V} & \boldsymbol{V} \\ \boldsymbol{U} & -\boldsymbol{U} \end{bmatrix}\begin{bmatrix} \boldsymbol{\Sigma} & 0 \\ 0 & -\boldsymbol{\Sigma} \end{bmatrix}\begin{bmatrix} \boldsymbol{V} & \boldsymbol{V} \\ \boldsymbol{U} & -\boldsymbol{U} \end{bmatrix}^\top,$$

*which is a symmetric matrix, and its eigenvalues are* $\pm\sigma_i(\widetilde{\boldsymbol{M}})$, *the singular values of* $\widetilde{\boldsymbol{M}}$. *Therefore our analysis on symmetric matrices also applies to the asymmetric setting and will equivalently depend on the gap between the top singular values of* $\widetilde{\boldsymbol{M}}$. *A formal asymmetric to symmetric conversion algorithm is presented in Algorithm 1 in [28].*

**Algorithm.** We note that by conducting zeroth-order gradient estimate $1/n \sum_{i=1}^n (f(\boldsymbol{a}/2 + \boldsymbol{z}_i/2) + \eta_i)\boldsymbol{z}_i$ with step-size $1/2$ [26] and sample size $n$, we get an estimate for $\mathbb{E}_{\eta,\boldsymbol{z}}[(f(\boldsymbol{a}/2 + \boldsymbol{z}/2) + \eta)\boldsymbol{z}] = \frac{\sigma^2}{2}\boldsymbol{M}\boldsymbol{a}$ when $\boldsymbol{z} \sim \mathcal{N}(0, \sigma^2)$. Therefore we are able to use noisy power method to recover the top eigenvector. We present the complete algorithm in Algorithm 1 and attain a gap-dependent risk bound:

---

**Algorithm 1** Noisy power method for bandit eigenvalue problem.

---

1: **Input:** Quadratic function $f : \mathcal{A} \to \mathbb{R}$ with noisy reward, failure probability $\delta$, error $\varepsilon$.
2: **Initialization:** Initial action $\boldsymbol{a}_0 \in \mathbb{R}^d$ randomly sampled on the unit sphere $\mathbb{S}^{d-1}$. We set $\alpha = |\lambda_2/\lambda_1|$, sample size per iteration $n = C_n d^2 \log(d/\delta)(\lambda_1\alpha)^{-2}\varepsilon^{-2}$, sample variance $m = C_m d\log(n/\delta)$, total iteration $L = \lfloor C_L \log(d/\varepsilon) \rfloor + 1$.
3: **for** Iteration $l$ from 1 to $L$ **do**
4:     Sample $\boldsymbol{z}_i \sim \mathcal{N}(0, 1/mI_d), i = 1, 2, \cdots n$. (Re-sample the whole batch if exists $\boldsymbol{z}_i$ with norm greater than 1.)
5:     **Noisy power method:**
6:         Take actions $\widetilde{\boldsymbol{a}}_i = \frac{\boldsymbol{a}_{l-1}+\boldsymbol{z}_i}{2}$ and observe $r_i = f(\boldsymbol{a}_i) + \eta_i, \forall i \in [n]$
7:         Update normalized action $\boldsymbol{a}_l \leftarrow \frac{m}{n}\sum_{i=1}^n r_i \boldsymbol{z}_i$, and normalize $\boldsymbol{a}_l \leftarrow \boldsymbol{a}_l/\|\boldsymbol{a}_l\|_2$.
8: **Output:** $\boldsymbol{a}_L$.

---

**Theorem 3.3** (Regret bound for noisy power method (NPM)). *In Algorithm 1, we set* $\varepsilon \in (0, 1/2)$, $\delta = 0.1/(L_0 S)$ *and let* $C_L, C_S, C_n, C_m$ *be large enough universal constants. Then with high probability 0.9 we have: the output* $\boldsymbol{a}_L$ *satisfies* $\tan\theta(\boldsymbol{a}^*, \boldsymbol{a}_L) \leq \varepsilon$ *and yields* $r^*\varepsilon^2$-*optimal reward; and the total number of actions we take is* $\widetilde{O}(\frac{\kappa d^2}{\Delta^2\varepsilon^2})$. *By explore-then-commit (ETC) the cumulative regret is at most* $\widetilde{O}(\sqrt{\kappa^3 d^2 T})$.

All proofs in this subsection are in Appendix C.1.

**Remark 3.4** (Intuition of [43] and how to overcome the conjectured lower bound via the design of adaptive algorithms). *Let us consider the rank 1 case of* $r(\boldsymbol{a}) = (\boldsymbol{a}^T\boldsymbol{\theta}^*)^2 + \eta$. *A random action* $\boldsymbol{a} \sim Unif(\mathbb{S}^{d-1})$ *has* $f(\boldsymbol{a}) \asymp 1/d^2$, *and the noise has standard deviation* $O(1)$. *Thus the signal-to-noise-ratio is* $O(1/d^2)$ *and the optimal action* $\boldsymbol{\theta}^*$ *requires* $d$ *bits to encode. If we were to play non-adaptively, this would require* $O(d^3)$ *queries and result in regret* $\sqrt{d^3 T}$ *which matches the result of [43].*

*To go beyond this, we must design algorithms that are adaptive, meaning the information in* $f(\boldsymbol{a}) + \eta$ *is **strictly larger than** $\frac{1}{d^2}$. As an illustration of why this is possible, consider batching the time-steps into $d$ stages so that each stage decode 1 bit of* $\boldsymbol{\theta}^*$. *At the first stage, random exploration* $\boldsymbol{a} \sim Unif(\mathbb{S}^{d-1})$ *gives signal-to-noise-ratio* $O(1/d^2)$. *Suppose $k$ bits of $\boldsymbol{\theta}$ are decoded at $k$-th stage by* $\widehat{\boldsymbol{\theta}}$, *adaptive algorithms can boost the signal-to-noise-ratio to* $O(k/d^2)$ *by using* $\widehat{\boldsymbol{\theta}}$ *as bootstrap (e.g. exploring with* $\widehat{\boldsymbol{\theta}} \pm a$ *where $a$ is random exploration in the unexplored subspace). In this way*

*adaptive algorithms only need $d^2/k$ queries in $(k+1)$-th stage and so the total number of queries sums up to $\sum_{k=1}^{d} d^2/k \approx d^2 \log d$.*

*Gradient descent and power method offer a computationally efficient and seamless way to implement the above intuition. For every iterate action $\boldsymbol{a}$, we estimate $\boldsymbol{M}\boldsymbol{a}$ from noisy observations and take it as our next action $\boldsymbol{a}^+$. With $d^2/(\Delta^2 \varepsilon^2)$ samples, noisy power method enjoys linear progress $\tan\theta(\boldsymbol{a}^+, \boldsymbol{a}^*) \leq \max\{c\tan(\boldsymbol{a}, \boldsymbol{a}^*), \varepsilon\}$, where $c < 1$ is a constant that depends on $\lambda_1, \lambda_2$, and $\varepsilon$. Therefore even though every step costs $d^2$ samples, overall we only need logarithmic (in $d, \varepsilon, \lambda_1, \lambda_2$) iterations to find an $\varepsilon$-optimal action.*

**Remark 3.5** (Connection to phase retrieval and eluder dimension). *For rank-1 case $\boldsymbol{M} = \boldsymbol{x}\boldsymbol{x}^\top$, the bilinear bandits can be viewed as phase retrieval, where one observes $y_r = (\boldsymbol{a}_r^\top \boldsymbol{x})^2 = \boldsymbol{a}_r^\top \boldsymbol{M} \boldsymbol{a}_r$ plus some noise $\eta_r \sim \mathcal{N}(0, \sigma^2)$. The optimal (among non-adaptive algorithms) sample complexity to recover $\boldsymbol{x}$ is $\sigma^2 d/\epsilon^2$ [16, 15] where they play $\boldsymbol{a}$ from random Gaussian $\mathcal{N}(0, \boldsymbol{I})$. However, in bandit, we need to set the variance of $\boldsymbol{a}$ to at most $1/d$ to ensure $\|\boldsymbol{a}\| \leq 1$. Our problem is equivalent to observing $y_r/d$ where their $\boldsymbol{a}/\sqrt{d} \sim \mathcal{N}(0, 1/d\boldsymbol{I})$ and noise level $\eta_r/d \sim \mathcal{N}(0, 1)$, i.e., $\sigma^2 = d^2$. Therefore one gets $d^3/\epsilon^2$ even for the rank-1 problems. On the other hand, for all rank-1 $\boldsymbol{M}$ the condition number $\kappa = 1$; and thus our results match the lower bound (see Section 3.3.2) up to logarithmic factors, and also have fundamental improvements for phase retrieval problems by leveraging adaptivity.*

*For UCB algorithms based on eluder dimension, the regret upper bound is $O(\sqrt{d_E \log(N)T}) = O(\sqrt{d^3 T})$ as presented in Theorem 2.1, where the dependence on $d$ is consistent with [43, 55] and is non-optimal.*

The previous result depends on the eigen-gap. When the matrix is ill-conditioned, i.e., $\lambda_1$ is very close to $\lambda_2$, we can obtain gap-free versions with a modification: The first idea stems from finding higher reward instead of recovering the optimal action. Therefore when $\lambda_1$ and $\lambda_2$ are very close (gap being smaller than desired accuracy $\varepsilon$), it is acceptable to find any direction in the span of $(\boldsymbol{v}_1, \boldsymbol{v}_2)$. More formally, we care about the convergence speed of identifying any action in the space spanned by any top eigenvectors (whose associated eigenvalues are higher than $\lambda_1 - \varepsilon$). Therefore the convergence speed will depend on $\varepsilon$ instead of $\lambda_1 - \lambda_2$.

**Corollary 3.6** (Gap-free regret bound). *For positive semi-definite matrix $\boldsymbol{M}$, by setting $\alpha = 1 - \varepsilon^2/2$ in Algorithm 1 and performing ETC afterwards, one can obtain cumulative regret of $\widetilde{O}(\lambda_1^{3/5} d^{2/5} T^{4/5})$.*

Again, the PSD assumption is not essential. For general symmetric matrices with $\lambda_1 \geq \lambda_2 \geq \cdots > 0 > \cdots \lambda_k$. We can still conduct shifted power method on $\boldsymbol{M} - \lambda_k \boldsymbol{I}$, yielding a cumulative regret of $(\lambda_1 + |\lambda_k|)^{3/5} d^{2/5} T^{4/5}$.

Another novel gap-free algorithm requires to identify any top eigenspace $\boldsymbol{V}_{1:l}, l \in [k]$: $\boldsymbol{V}_{1:l}$ is the column span of $\{\boldsymbol{v}_1, \boldsymbol{v}_2, \cdots \boldsymbol{v}_l\}$. Notice that in traditional subspace iteration, the convergence rate of recovering $\boldsymbol{V}_l$ depends on the eigengap $\Delta_l := |\lambda_l| - |\lambda_{l+1}|$. Meanwhile, since $\sum_{l=1}^k \Delta_l = \lambda_1$, at least one eigengap is larger or equal to $\lambda_1/k$. Suppose $\Delta_{l^*} \geq \lambda_1/k$; we can, therefore, set $\alpha = 1 - 1/k$ and recover the top $l^*$ subspace up to $\lambda_1 \epsilon$ error, which will give an $\epsilon$-optimal reward in the end. We don't know $l^*$ beforehand and will try recovering the top subspace $\boldsymbol{V}_1, \boldsymbol{V}_2, \cdots \boldsymbol{V}_k$ respectively, which will only lose a $k$ factor. With the existence of $l^*$, at least one trial (on recovering $\boldsymbol{V}_{l^*}$) will be successful with the parameters of our choice, and we simply output the best action among all trials.

**Theorem 3.7** (Informal statement: (gap-free) subspace iteration). *By running subspace iteration (Algorithm 3) with proper choices of parameters, we attain a cumulative regret of $\widetilde{O}(\lambda_1^{1/3} k^{4/3}(dT)^{2/3})$. Algorithm 3) with another set of parameters can also recover the whole eigenspace, and achieve cumulative regret of $\widetilde{O}((\lambda_1 k)^{1/3}(\widetilde{\kappa} dT)^{2/3})$, where $\widetilde{\kappa} = \lambda_1/|\lambda_k|$.*

## 3.2 Stochastic Low-rank Linear Bandits ($\mathcal{F}_{\mathbf{LR}}$)

In the low-rank linear bandit, the reward function is $f(\boldsymbol{A}) = \langle \boldsymbol{A}, \boldsymbol{M} \rangle$, with noisy observations $r_t = f(\boldsymbol{A}) + \eta_t$, and the action space is $\{\boldsymbol{A} \in \mathbb{R}^{d \times d} : \|\boldsymbol{A}\|_F \leq 1\}$. Without loss of generality we assume $k$, the rank of $\boldsymbol{M}$ satisfies $k \leq \frac{d}{2}$, since when $k$ is of the same order as $d$, the known upper and lower bound are both $\sqrt{d^2 k^2 T} = \Theta(\sqrt{d^4 T})$ [55] and there is no room for improvement. We

| $\mathcal{F}_{\text{EV}}$ | LB ($k=1$) | [43] | NPM | Gap-free NPM | Subspace Iteration |
|---|---|---|---|---|---|
| Regret | $\sqrt{d^2 T}$ | $\sqrt{d^3 k \lambda_k^{-2} T}$ | $\sqrt{\kappa^3 d^2 T}$ | $d^{2/5} T^{4/5}$ | $\min(k^{4/3}(dT)^{2/3}, k^{1/3}(\widetilde{\kappa} dT)^{2/3})$ |

| $\mathcal{F}_{\text{LR}}$ | LB ([55]) | UB ([55]) | | Subspace Iteration |
|---|---|---|---|---|
| Regret | $\Omega(\sqrt{d^2 k^2 T})$ | $\sqrt{d^3 k T}^{*}$ or $\sqrt{d^3 k \lambda_k^{-2} T}$ | | $\min(\sqrt{d^2 k \lambda_k^{-2} T}, (dkT)^{2/3})$ |

Table 2: Summary of results for quadratic reward. All red expressions are our results. LB, UB, NPM stands for lower bound, upper bound, and noisy power method respectively. $\Delta = \lambda_1 - |\lambda_2|$ is the eigengap, and $\kappa = \lambda_1/\Delta$, $\widetilde{\kappa} = \lambda_1/|\lambda_k|$ are the condition numbers. The result with $*$ is not computationally tractable. For low-rank setting in this table, we treat $\|M\|_F$ as a constant for simplicity and leave its dependence in the theorems. Our upper bounds match the lower bound in terms of dimension and substantially improve over existing algorithms that are computationally efficient.

write $r^* = \|M\|_F \leq 1$, therefore the optimal action is $A^* = M/r^*$. In this section, we write $X(s)$ to be the $s$-th column of any matrix $X$.

As presented in Algorithm 4, we conduct noisy subspace iteration to estimate the right eigenspace of $M$. Subspace iteration requires calculating $MX_t$ at every step. This can be done by considering a change of variable of $g(X) := f(XX^\top) = \langle XX^\top, M \rangle$ whose gradient[4] is $\nabla g(X) = 2MX$. The zeroth-order gradient estimator can then be employed to stochastically estimate $MX$. We instantiate the analysis with symmetric $M$ while extending to asymmetric setting is straightforward since the problem can be reduced to symmetric setting (suggested in Remark 3.2).

With stochastic observations and randomly sampled actions, we achieve the next iterate $Y_l$ that satisfies $MX_{l-1} \equiv \mathbb{E}[Y_l]$ in Algorithm 4. Let $M = V\Sigma V^\top$. With proper concentration bounds presented in the appendix, we can apply the analysis of noisy power method [37] and get:

**Theorem 3.8** (Informal statement; sample complexity for low-rank linear reward). *With Algorithm 4, $X_L$ satisfies $\|(I - X_L X_L^\top)V\| \leq \varepsilon/4$, and output $\widehat{A}$ satisfies $\|\widehat{A} - A^*\|_F \leq \varepsilon\|M\|_F$ with sample size $\widetilde{O}(d^2 k \lambda_k^{-2} \varepsilon^{-2})$.*

We defer the proofs for low-rank linear reward in Appendix C.2.

**Corollary 3.9** (Regret bound for low-rank linear reward). *We first call Algorithm 4 with $\varepsilon^4 = \widetilde{\Theta}(d^2 k \lambda_k^{-2} T^{-1})$ to obtain $\widehat{A}$; we then play $\widehat{A}$ for the remaining steps. The cumulative regret satisfies*

$$\mathfrak{R}(T) \leq \widetilde{O}(\sqrt{d^2 k (r^*)^2 \lambda_k^{-2} T}),$$

*with high probability* $0.9$.

To be more precise, we need $T \geq \widetilde{\Theta}(d^2 k/\lambda_k^2)$ for Algorithm 4 to take sufficient actions; however, the conclusion still holds for smaller $T$. Since simply playing $0$ for all $T$ actions will give a sharper bound of $\mathfrak{R}(T) \leq r^* T \leq \widetilde{O}(r^*\sqrt{d^2 k \lambda_k^{-2} T}))$. For cleaner presentation, we won't stress this for every statement.

Notice that $k \leq \|M\|_F^2/\lambda_k^2 \leq k\widetilde{\kappa}^2$ is order $k$. Thus for well-conditioned matrices $M$, our upper bound of $\widetilde{O}(\sqrt{d^2 k \|M\|_F^2/\lambda_k^2 T})$ matches the lower bound $\sqrt{d^2 k^2 T}$ except for logarithmic factors.

In the previous setting with the bandit eigenvalue problem, estimating $M$ up to an $\epsilon$-error (measured by **operator norm**) gives us an $\epsilon$-optimal reward. Therefore the sample complexity for eigenvalue reward with similar subspace iteration is $\|M\|_2^2 d^2 k/(\lambda_k^2 \epsilon^2)$. In this section, on the other hand, we need **Frobenius norm** bound $\|A_L - A^*\|_F \leq \epsilon$; naturally the complexity becomes $\|M\|_F^2 d^2 k/(\lambda_k^2 \epsilon^2)$.

**Theorem 3.10** (Regret bound for low-rank linear reward: gap-free case). *Set $\varepsilon^6 = \Theta(\frac{d^2 k^2}{(r^*)^2 T})$, $n = \widetilde{\Theta}(\frac{d^2 k^2}{(r^*)^2 \varepsilon^4})$, $L = \Theta(\log(d/\varepsilon))$ and $k' = 2k$ in Algorithm 4 and get $\widehat{A}$. Then we play it for the remaining steps, the cumulative regret satisfies:*

$$\mathfrak{R}(T) \leq \widetilde{O}((dkT)^{2/3}(r^*)^{1/3}).$$

---

[4]Directly performing projected gradient descent on $f(A)$ would not work, since this is not an adaptive algorithm as the gradient of a linear function is constant. This would incur regret $\sqrt{d^3 \text{poly}(k)T}$.

We summarize all our results and prior work for quadratic reward in Table 2.

### 3.2.1 RL with Simulator: $Q$-function is Quadratic and Bellman Complete

In this section we demonstrate how our results for non-concave bandits also apply to reinforcement learning. Let $\mathcal{T}_h$ be the Bellman operator applied to the Q-function $Q_{h+1}$ defined as:

$$\mathcal{T}_h(Q_{h+1})(s,a) = r_h(s,a) + \mathbb{E}_{s' \sim \mathbb{P}(\cdot|s,a)}[\max_{a'} Q_{h+1}(s',a')].$$

**Definition 3.11** (Bellman complete). Given MDP $\mathcal{M} = (\mathcal{S}, \mathcal{A}, \mathbb{P}, r, H)$, function class $\mathcal{F}_h : \mathcal{S} \times \mathcal{A} \mapsto \mathbb{R}, h \in [H]$ is called Bellman complete if for all $h \in [H]$ and $Q_{h+1} \in \mathcal{F}_{h+1}$, $\mathcal{T}_h(Q_{h+1}) \in \mathcal{F}_h$.

**Assumption 3.12** (Bellman complete for low-rank quadratic reward). We assume the function class $\mathcal{F}_h = \{f_{\boldsymbol{M}} : f_{\boldsymbol{M}}(s,a) = \phi(s,a)^\top \boldsymbol{M} \phi(s,a), \text{rank}(\boldsymbol{M}) \leq k, \text{ and } 0 < \lambda_1(\boldsymbol{M})/\lambda_{\min}(\boldsymbol{M}) \leq \widetilde{\kappa}.\}$ is a class of quadratic function and the MDP is Bellman complete. Here $f_{\boldsymbol{M}}(\phi(s,a)) = \sum_{j=1}^k \lambda_j (\boldsymbol{v}_j^\top \phi(s,a))^2$ when $\boldsymbol{M} = \sum_{j=1}^k \lambda_j \boldsymbol{v}_j \boldsymbol{v}_j^\top$. Write $Q_h^* = f_{\boldsymbol{M}_h^*} \in \mathcal{F}_h$.

*Observation:* When querying $s_{h-1}, a_{h-1}$, we observe $s_h' \sim \mathbb{P}(\cdot|s_{h-1}, a_{h-1})$ and reward $r_{h-1}(s_{h-1}, a_{h-1})$.

*Oracle to recover parameter $\widehat{\boldsymbol{M}}$:* Given $n \geq \widetilde{\Theta}(d^2 k^2 \widetilde{\kappa}^2(\boldsymbol{M})/\varepsilon^2)$, if one can play $\geq n$ samples $\boldsymbol{a}_i$ and observe $y_i \sim \boldsymbol{a}_i^\top \boldsymbol{M} \boldsymbol{a}_i + \eta_i, i \in [n]$ with 1-sub-gaussian and mean-zero noise $\eta$, we can recover $\widehat{\boldsymbol{M}} = \widehat{\boldsymbol{M}}(\{(\boldsymbol{a}_i, y_i)\})$ such that $\|\widehat{\boldsymbol{M}} - \boldsymbol{M}\|_2 \leq \varepsilon$. This oracle is implemented via our analysis from the bandit setting.

With the oracle, at time step $H$, we can estimate $\widehat{\boldsymbol{M}}_H$ that is $\epsilon/H$ close to $\boldsymbol{M}_H^*$ in spectral norm through noisy observations from the reward function with $\widetilde{O}(\widetilde{\kappa}^2 d^2 H^2/\epsilon^2)$ samples. Next, for each time step $h = H - 1, H - 1, \cdots, 1$, sample $s_i' \sim \mathbb{P}(\cdot|s,a)$, we define $\eta_i = \max_{a'} f_{\widehat{\boldsymbol{M}}_{h+1}}(s_i', a') - \mathbb{E}_{s' \sim \mathbb{P}(\cdot|s,a)} \max_{a'} f_{\widehat{\boldsymbol{M}}_{h+1}}(s', a')$. $\eta_i$ is mean-zero and $O(1)$-sub-gaussian since it is bounded. Denote $\boldsymbol{M}_h$ as the matrix that satisfies $f_{\boldsymbol{M}_h} := \mathcal{T} f_{\widehat{\boldsymbol{M}}_{h+1}}$, which is well-defined due to Bellman completeness.

We estimate $\widehat{\boldsymbol{M}}_h$ from the noisy observations $y_i = r_h(s,a) + \max_{a'} f_{\widehat{\boldsymbol{M}}_{h+1}}(s_i', a') = \mathcal{T} f_{\widehat{\boldsymbol{M}}_{h+1}} + \eta_i =: f_{\boldsymbol{M}_h} + \eta_i$. Therefore with the oracle, we can estimate $\widehat{\boldsymbol{M}}_h$ such that $\|\widehat{\boldsymbol{M}}_h - \boldsymbol{M}_h\|_2 \leq \epsilon/H$ with $\Theta(\widetilde{\kappa}^2 d^2 k^2 H^2/\epsilon^2)$ bandits. More details are deferred to Algorithm 5 and the Appendix. We state the theorem on sample complexity of finding $\epsilon$-optimal policy here:

**Theorem 3.13.** *Suppose $\mathcal{F}$ is Bellman complete associated with parameter $\widetilde{\kappa}$. With probability $1 - \delta$, Algorithm 5 learns an $\epsilon$-optimal policy $\pi$ with $\widetilde{\Theta}(d^2 k^2 \widetilde{\kappa}^2 H^3/\epsilon^2)$ samples.*

Existing approaches require $O(d^3 H^2/\epsilon^2)$ trajectories, or equivalently $O(d^3 H^3/\epsilon^2)$ samples, though they operate in the online RL setting [75, 23, 41], which is worse by a factor of $d$.

It is an open problem to attain $d^2$ sample complexity in the online RL setting. The quadratic Bellman complete setting can also be easily extended to any of the polynomial settings of Section 3.3.

## 3.3 Stochastic High-order Homogeneous Polynomial Reward

Next we move on to homogeneous high-order polynomials.

### 3.3.1 The symmetric setting

Let reward function be a $p$-th order stochastic polynomial function $f : \mathcal{A} \to \mathbb{R}$, where the action set $\mathcal{A} := \{B_1^d = \{\boldsymbol{a} \in \mathbb{R}^d, \|\boldsymbol{a}\| \leq 1.\}$. $f(\boldsymbol{a}) = \boldsymbol{T}(\boldsymbol{a}^{\otimes p}), r_t = f(\boldsymbol{a}) + \eta_t$, where $\boldsymbol{T} = \sum_{j=1}^k \lambda_j \boldsymbol{v}_j^{\otimes p}$ is an orthogonally decomposable rank-$k$ tensor. $\{\boldsymbol{v}_1, \cdots \boldsymbol{v}_k\}$ form an orthonormal basis. Optimal reward $r^*$ satisfies $1 \geq r^* = \lambda_1 \geq |\lambda_2| \cdots \geq |\lambda_k|$. Noise $\eta_t \sim \mathcal{N}(0,1)$.

In this setting, the problem is fundamentally more challenging than quadratic reward functions. On the one hand, it has a higher noise-to-signal ratio with larger $p$. One can tell from the rank-1 setting where $\boldsymbol{T} = \lambda_1 \boldsymbol{v}_1^{\otimes p}$. For a randomly generated action $\boldsymbol{a}$ on the unit ball, $\mathbb{E}[\|\boldsymbol{a}^\top \boldsymbol{v}_1\|^2] = 1/d$. Therefore on average the signal strength is only $(\boldsymbol{a}^\top \boldsymbol{v}_1)^p \sim d^{-p/2}$, much smaller than the noise level 1. Intuitively this demonstrates why higher complexity is needed for high-order polynomials. On the

other hand, it is also technically more challenging. Unlike the matrix case, the expected zeroth-order update is no longer equal to any tensor product. Therefore existing tensor decomposition arguments do not apply. Fortunately, we prove that zeroth-order optimization still pushes the iterated actions toward the optimal action with linear convergence, given a good initialization. We show the bandit optimization procedure in Algorithm 6 and present the result in Theorem 3.14:

**Theorem 3.14** (Staged progress). *For each stage $s$, with high probability $\mathcal{A}_s$ is not empty; and at least one action $\boldsymbol{a} \in \mathcal{A}_s$ satisfies:* $\tan\theta(\boldsymbol{a}, \boldsymbol{a}^*) \leq \widetilde{\varepsilon}_s = 2^{-s}$.

We defer the proofs together with formal statements to Appendix D.1.1 and D.1.

**Remark 3.15** (Choice of step-size). *Here we choose step-size $\zeta = 1/2p$. Note that $\mathbb{E}[\boldsymbol{y}] = \zeta(1-\zeta)^p \sigma^2 \nabla_{\boldsymbol{a}} f(\boldsymbol{a})$. The scaling $(1-\zeta)^p \geq 1/\sqrt{e}$ ensures the signal to noise ratio not too small. The choice of weighted action is a delicate balance between making progress in optimization and controlling the noise to signal ratio.*

**Corollary 3.16** (Regret bound for tensors). *Algorithm 6 yields an regret of:* $\mathfrak{R}(T) \leq \widetilde{O}\left(\sqrt{kd^p T}\right)$, *with high probability.*

**Corollary 3.17** (Regret bound with burn-in period). *In Algorithm 6, we first set $\varepsilon = 1/p, n_s = \widetilde{\Theta}(d^p/\lambda_1^2)$. We can first estimate the action $\boldsymbol{a}$ such that $\boldsymbol{v}_1^\top \boldsymbol{a} \geq 1 - 1/p$ with $\widetilde{O}(kd^p/\lambda_1^2)$ samples. Next we change $\varepsilon = \widetilde{\Theta}(k^{1/4} d^{1/2} \lambda_1^{-1/2} T^{-1/4})$, and $n_s = \widetilde{\Theta}(d^2 \varepsilon^{-2} \lambda_1^{-2})$ in Algorithm 6. This procedure suffices to find $\lambda_1 \varepsilon^2$-optimal reward with $\widetilde{O}(kd^2/\lambda_1^2 \varepsilon^2)$ samples in the candidate set with size at most $\widetilde{O}(k)$. Finally with the UCB algorithm altogether we have a regret bound of: $\widetilde{O}(\frac{kd^p}{\lambda_1} + \sqrt{kd^2 T})$.*

In Section 3.3.2 we also demonstrate the necessity of the sample complexity in this burn-in period.

### 3.3.2 Lower Bounds for Stochastic Polynomial Bandits

In this section we show lower bound for stochastic polynomial bandits.

$$f(\boldsymbol{a}) = \prod_{i=1}^p (\boldsymbol{\theta}_i^\top \boldsymbol{a}) + \eta, \text{ where } \eta \sim N(0,1), \|\boldsymbol{a}\|_2 \leq 1 \text{ and } f(\boldsymbol{a}) \leq 1. \tag{1}$$

**Theorem 3.18.** *Define minimax regret as follow*

$$\mathfrak{R}(d,p,T) = \inf_\pi \sup_{(\boldsymbol{\theta}_1,\ldots,\boldsymbol{\theta}_p)} \mathbb{E}_{(\boldsymbol{\theta}_1,\ldots,\boldsymbol{\theta}_p)} \left[ T \max_{\boldsymbol{a}} \prod_{i=1}^p (\boldsymbol{\theta}_i^\top \boldsymbol{a}) - \sum_{t=1}^T \prod_{i=1}^p (\boldsymbol{\theta}_i^\top \boldsymbol{a}^{(t)}) \right].$$

*For all algorithms $\mathcal{A}$ that adaptively interact with bandit (Eq (1)) for $T$ rounds, we have $\mathfrak{R}(d,p,T) \geq \Omega(\sqrt{d^p T}/p^p)$.*

From the theorem, we can see even when the problem is rank-1, any algorithm incurs at least $\Omega(\sqrt{d^p T}/p^p)$ regret. This further implies algorithm requires sample complexity of $\Omega((d/p^2)^p/\epsilon^2)$ to attain $\epsilon$-optimal reward. This means our regret upper bound obtained in Corollary 3.16 is optimal in terms of dependence on $d$. In Appendix G we also show a $\Omega(\sqrt{d^p T})$ lower bound for asymmetric actions setting, which our upper bound up to poly-logarithmic factors.

We note that with burn-in period, our algorithm also obtains a cumulative regret of $kd^p/\lambda_1 + \sqrt{kd^2 T}$, as shown in Corollary 3.17. Here for a fixed $\lambda_1$ and very large $T$, this is better result than the previous upper bound. We note that there is no contradiction with the lower bound above, since this worst case is achieved with a specific relation between $T$ and $\lambda_1$.

The burn-in period requires $kd^p/\lambda_1^2$ samples to get a constant of $r^*$. We want to investigate whether our dependence on $\lambda_1 \equiv r^*$ is optimal. Next we show a gap-dependent lower bound for finding an arm that is close to the optimal arm by a constant factor.

**Theorem 3.19.** *For all algorithms $\mathcal{A}$ that adaptively interact with bandit (Eq (1)) for $T$ rounds and output a vector $\boldsymbol{a}^{(T)} \in \mathbb{R}^d$, it requires at least $T = \Omega(d^p/\|\boldsymbol{\theta}\|^{2p})$ rounds to find an arm $\boldsymbol{a}^{(T)} \in \mathbb{R}^d$ such that $\prod_{i=1}^p (\boldsymbol{\theta}_i^\top \boldsymbol{a}^{(T)}) \geq \frac{3}{4} \cdot \max_{\boldsymbol{a}} \prod_{i=1}^p (\boldsymbol{\theta}_i^\top \boldsymbol{a})$.*

In the rank-1 setting $r^* = \|\boldsymbol{\theta}\|^p$ and the lower bound for achieving a constant approximation for the optimal reward is $\Omega(d^p/(r^*)^2)$. Therefore our burn-in sample complexity is also optimal in the dependence on $d$ and $r^*$.

### 3.4 Noiseless Polynomial Reward

In this subsection, we study the regret bounds for learning bandits with **noiseless** polynomial rewards. First we present the definition of admissible polynomial families.

**Definition 3.20** (Admissible Polynomial Family). For $\boldsymbol{a} \in \mathbb{R}^d$, define $\widetilde{\boldsymbol{a}} = [1, \boldsymbol{a}^\top]^\top$. For a algebraic variety $\mathcal{V} \subseteq (\mathbb{R}^{d+1})^{\otimes p}$, define $\mathcal{R}_\mathcal{V} := \{r_{\boldsymbol{\theta}}(\boldsymbol{a}) = \langle \boldsymbol{\theta}, \widetilde{\boldsymbol{a}}^{\otimes p} \rangle : \boldsymbol{\theta} \in \mathcal{V}\}$ as the polynomial family with parameters in $\mathcal{V}$. We define the dimension of the family $\mathcal{R}_\mathcal{V}$ as the algebraic dimension of $\mathcal{V}$. Next, define $\mathcal{X} := \{\widetilde{\boldsymbol{a}}^{\otimes p} : \boldsymbol{a} \in \mathbb{R}^d\}$. An polynomial family $\mathcal{R}_\mathcal{V}$ is said to be admissible[5] w.r.t. $\mathcal{X}$ if for any $\boldsymbol{\theta} \in \mathcal{V}, \dim(\mathcal{X} \cap \{X \in \mathcal{X} : \langle X, \boldsymbol{\theta} \rangle = 0)\}) < \dim(\mathcal{X}) = d$.

#### 3.4.1 Upper Bounds via Solving Polynomial Equations

We show that if the action set $\mathcal{A}$ is of positive measure with respect to the Lebesgue measure $\mu$, then by playing actions randomly, we can uniquely solve for the ground-truth reward function $r_{\boldsymbol{\theta}}(\boldsymbol{a})$ almost surely with samples of size that scales with the intrinsic algebraic dimension of $\mathcal{V}$, provided that $\mathcal{V}$ is an admissible algebraic variety.

**Theorem 3.21.** *Assume that the reward function class is an admissible polynomial family $\mathcal{R}_\mathcal{V}$, and the maximum reward is upper bounded by $1$. If $\mu(\mathcal{A}) > 0$, where $\mu$ is the Lebesgue meaure, then by randomly sample actions $\boldsymbol{a}_1, \ldots, \boldsymbol{a}_T$ from $\mathbb{P}_{\boldsymbol{a} \sim \mathcal{N}(0, I_d)}(\cdot | \boldsymbol{a} \in \mathcal{A})$, when $T \geq 2\dim(\mathcal{V})$, we can uniquely solve for the ground-truth $\boldsymbol{\theta}$ and thus determine the optimal action almost surely. Therefore, the cumulative regret at round $T$ can be bounded as*

$$\Re(T) \leq \min\{T, 2\dim(\mathcal{V})\}.$$

We state two important examples of admissible polynomial families with $O(d)$ dimensions.

**Example 3.22** (low-rank polynomials). The function class $\mathcal{R}_\mathcal{V}$ of possibly inhomogeneous degree-$p$ polynomials with $k$ summands $\mathcal{R}_\mathcal{V} = \{r(\boldsymbol{a}) = \sum_{i=1}^k \lambda_i \langle \boldsymbol{v}_i, \boldsymbol{a} \rangle^{p_i} \mid \lambda_i \in \mathbb{R}, \boldsymbol{v}_i \in \mathbb{R}^d\}$ is admissible with $\dim(\mathcal{R}_\mathcal{V}) \leq dk$, where $p = \max\{p_i\}$. Neural network with monomial/polynomial activation functions are low-rank polynomials.

**Example 3.23** ([18]). The function class $\mathcal{R}_\mathcal{V} = \{r(\boldsymbol{a}) = q(\boldsymbol{U}\boldsymbol{a}) \mid \boldsymbol{U} \in \mathbb{R}^{k \times d}, \deg q(\cdot) \leq p\}$ is admissible with $\dim(\mathcal{V}) \leq dk + (k+1)^p$.

## 4 Conclusion

In this paper, we design minimax-optimal algorithms for a broad class of bandit problems with non-concave rewards. For the stochastic setting, our algorithms and analysis cover the low-rank linear reward setting, bandit eigenvector problem, and homogeneous polynomial reward functions. We improve the best-known regret from prior work and attain the optimal dependence on problem dimension $d$. Our techniques naturally extend to RL in the generative model settings. Furthermore, we obtain the optimal regret, dependent on the intrinsic algebraic dimension, for general polynomial reward without noise. Our regret bound demonstrates the fundamental limits of UCB algorithms, being $\Omega(d^{p-1})$ worse than our result for cases of interest.

We leave to future work several directions. First, the gap-free algorithms for the low-rank linear and bandit eigenvector problem do not attain $\sqrt{T}$ regret. We believe $\sqrt{d^2 T}$ regret without any dependence on gap is impossible, but do not have a lower bound. Secondly, our study of degree $p$ polynomials only covers the noiseless setting or orthogonal tensors with noise. We believe entirely new algorithms are needed for general polynomial bandits to surpass eluder dimension. As a first step, designing optimal algorithms would require understanding the stability of algebraic varieties under noise, so we leave this as a difficult future problem. Finally, we conjecture our techniques can be used to design optimal algorithms for representation learning in bandits and MDPs [74, 39].

## Acknowledgment

JDL acknowledges support of the ARO under MURI Award W911NF-11-1-0303, the Sloan Research Fellowship, NSF CCF 2002272, and an ONR Young Investigator Award. QL is supported by NSF

---

[5] Intuitively, admissibility means the dimension of $\mathcal{X}$ decreases by one when there is an additional linear constraint $\langle \boldsymbol{\theta}, X \rangle = 0$

2030859 and the Computing Research Association for the CIFellows Project. SK acknowledges funding from the NSF Award CCF-1703574 and the ONR award N00014-18-1-2247. The authors would like to thank Qian Yu for numerous conversations regarding the lower bound in Theorem 3.23. JDL would like to thank Yuxin Chen and Anru Zhang for several conversations on tensor power iteration, Simon S. Du and Yangyi Lu for explaining to him to the papers of [43, 55], and Max Simchowitz and Chao Gao for several conversations regarding adaptive lower bounds.

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
