# A Omitted Related Work

**Neural Kernel Bandits.** [66] initiated the study of kernelized linear bandits, showing regret dependent on the information gain. The work of [76, 73] specialized this to the Neural Tangent Kernel (NTK) [54, 24, 40, 27, 10], where the algorithm utilizes gradient descent but remains close to initialization and thus remains a kernel class. Furthermore, NTK methods require $d^p$ samples to express a degree $p$ polynomial in $d$ dimensions [31], similar to eluder dimension of polynomials, and so lack the inductive biases necessary for real-world applications of decision-making problems [60].

**Neural Bandits.** For bandits with practical neural networks (instead of overparameterized NTKs) as the function approximator, we are not aware of any previous paper that gives provably efficient bandit algorithm for this case. Our paper gives the first provably efficient algorithm for neural bandits with noiseless reward and deterministic activation. We note that however, previous paper has already solved neural bandits when the neural network happens to be convex [21].

**Concave Bandits.** There has been a rich line of work on concave bandits starting with [26, 46]. [4] attained the first $\sqrt{T}$ regret algorithm for concave bandits though with a large $\mathrm{poly}(d)$ dependence. In the adversarial setting, a line of work [38, 14, 49] have attained polynomial-time algorithms with $\sqrt{T}$ regret with increasingly improved dimension dependence. The sharp dimension dependence remains unknown.

**Noiseless Bandits.** In the noiseless setting, there is some investigation in phase retrieval borrowing the tools from algebraic geometry (see e.g. [69]). In this paper, we will study the bandit problem with more general reward functions: neural nets with polynomial activation (structured polynomials) including phase retrieval. [45] study similar structured polynomials, also using tools from algebraic geometry, but they only study the expressivity of those polynomials and do not consider the learning problems. [21] study noiseless bandits with bounded Sequential Rademacher Complexity, but focus on attaining local optimality.

**Concurrent work.** [50] address the phase retrieval bandit problem which is equivalent to a symmetric rank 1 variant of the bilinear bandit of [43] and attain $\widetilde{O}(\sqrt{d^2 T})$ regret. Our work in Section 3.1 specialized to the rank 1 case attains the same regret.

**Matrix/Tensor Power Method.** Our analysis stems from noisy power methods for matrix/tensor decomposition problems. Robust power method, subspace iteration, and tensor decomposition that tolerate noise first appeared in [37, 8]. Follow-up work attained the optimal rate for both gap-dependence and gap-free settings for matrix decomposition [58, 6]. An improvement on the problem dimension for tensor power method is established in [70]. [63] considers the convergence of tensor power method in the non-orthogonal case.

# B Additional Preliminaries

In this section we show that adapting the eluder UCB algorithms from [62] would yield the sample complexity in Theorem 2.1. Especially we give the rates in Table 1 for our stochastic settings.

**The algorithm** [62] consider Algorithm 2 for the stochastic generalized linear bandit problem. Assume that $\boldsymbol{\theta}^*$ is the true parameter of the reward model. The reward is $r_t = f_{\boldsymbol{\theta}^*}(\boldsymbol{a}_t) + \eta_t$ for $f_{\boldsymbol{\theta}^*} \in \mathcal{F}$. Let $N$ be the $\alpha$-covering-number (under $\|\cdot\|_\infty$) of $\mathcal{F}$, $d_E$ be the $\alpha$-eluder-dimension of $\mathcal{F}$ (see Definition 3,4 in [62]). Let $C = \sup\limits_{f \in \mathcal{F}, a \in \mathcal{A}} |f(a)|$. We set $\alpha = \frac{1}{T^2}$ in the algorithm.

**The regret analysis** Choosing $\alpha = 1/T^2$, proposition 4 in [62] state that with probability $1 - \delta$, for some universal constant $C$, the total regret $\mathfrak{R}(T) \leq \frac{1}{T} + C \min\{d_E, T\} + 4\sqrt{d_E \beta_T T} \leq 1 + C\sqrt{d_E T} + 4\sqrt{d_E \beta_T T} = O(\sqrt{d_E(1 + \beta_T)T})$. In our settings with $\alpha = 1/T^2$, $\beta_T = 8\log(N/\delta) + 2(8C + \sqrt{8\ln(4T^2/\delta)})/T = O(\log(N/\delta))$ where $\log(N) = \Omega(1)$ for our action sets, and thus

$$\mathfrak{R}(T) = \widetilde{O}(\sqrt{d_E T \log N}).$$

---

**Algorithm 2** Eluder UCB

---

1: **Input:** Function class $\mathcal{F}$, failure probability $\delta$, parameters $\alpha, N, C$.
2: **Initialization:** $\mathcal{F}_0 \leftarrow \mathcal{F}$.
3: **for** $t$ from 1 to $T$ **do**
4:     **Select Action:**
5:     $\boldsymbol{a}_t \in \arg\max_{\boldsymbol{a} \in \mathcal{A}} \sup_{f_{\boldsymbol{\theta}} \in \mathcal{F}_{t-1}} f_{\boldsymbol{\theta}}(\boldsymbol{a})$
6:     Play action $\boldsymbol{a}_t$ and observe reward $r_t$
7:     **Update Statistics:**
8:     $\widehat{\boldsymbol{\theta}}_t \in \arg\min_{\boldsymbol{\theta}} \sum_{s=1}^{t}(f_{\boldsymbol{\theta}}(\boldsymbol{a}_s) - r_s)^2$
9:     $\beta_t \leftarrow 8\log(N/\delta) + 2\alpha t(8C + \sqrt{8\ln(4t^2/\delta)})$
10:     $\mathcal{F}_t \leftarrow \{f_{\boldsymbol{\theta}} : \sum_{s=1}^{t}(f_{\boldsymbol{\theta}} - f_{\widehat{\boldsymbol{\theta}}_t})^2(\boldsymbol{a}_s) \leq \beta_t\}$

---

**Applications in our settings**    We show that in our settings Theorem 2.1 will obtain the rates listed in Table 1.

**The covering numbers**

**Lemma B.1.** *The log-covering-number (of radius $\alpha$ with $\alpha \ll 1$, under $\|\cdot\|_{\infty}$) of the function classes are:* $\log N(\mathcal{F}_{SYM}) = O(dk\log\frac{k}{\alpha})$, $\log N(\mathcal{F}_{ASYM}) = O(dk\log\frac{k}{\alpha})$, $\log N(\mathcal{F}_{EV}) = O(dk\log\frac{k}{\alpha})$, *and* $\log N(\mathcal{F}_{LR}) = O(dk\log\frac{k}{\alpha})$.

*Proof.* Let $S_{\xi}^d$ denote a minimal $\xi$-covering of $\mathbb{S}^{d-1}$ (under $\|\cdot\|_2$) for $0 < \xi < \frac{1}{10}$, and $|S_{\xi}^d| = O(d\log 1/\xi)$ (see for example [62]). Then we can construct the coverings in our settings from $S_{\xi}^d$:

- $\mathcal{F}_{SYM}$: let $\xi = \frac{\alpha}{kp}$, and for $k$ copies of $S_{\xi}^d$, we can construct a covering of $\mathcal{F}_{SYM}$ with size $|S_{\xi}^d|^k$. Specifically, let the covering be $S_{SYM} = \{g(\boldsymbol{a}) = \sum_{j=1}^{k} \lambda_j (\boldsymbol{u}_j^{\top}\boldsymbol{a})^p : (\boldsymbol{u}_1, \boldsymbol{u}_2, \cdots, \boldsymbol{u}_k) \in S_{\xi}^d \times S_{\xi}^d \times \cdots \times S_{\xi}^d\}$, then for each $f(\boldsymbol{a}) = \sum_{j=1}^{k} \lambda_j(\boldsymbol{v}_j^{\top}\boldsymbol{a})^p \in \mathcal{F}_{SYM}$, as we can find $\boldsymbol{u}_j \in S_{\xi}^d$ that $\|\boldsymbol{u}_j - \boldsymbol{v}_j\|_2 \leq \xi$,

$$\sup_{\boldsymbol{a}}[f(\boldsymbol{a}) - g(\boldsymbol{a})] \leq \sup_{\boldsymbol{a}}[\sum_{j=1}^{k}|\lambda_j||\boldsymbol{u}_j^{\top}\boldsymbol{a} - \boldsymbol{v}_j^{\top}\boldsymbol{a}||\sum_{q=0}^{p-1}(\boldsymbol{u}_j^{\top}\boldsymbol{a})^q(\boldsymbol{v}_j^{\top}\boldsymbol{a})^{p-q-1}|] \leq pk\xi = \alpha;$$

- $\mathcal{F}_{ASYM}$: let $\xi = \frac{\alpha}{kp}$, and for $kp$ copies of $S_{\xi}^d$, let the covering be $S_{ASYM} = \{g(\boldsymbol{a}) = \sum_{j=1}^{k} \lambda_j \prod_{q=1}^{p}(\boldsymbol{u}_j(q)^{\top}\boldsymbol{a}(q)) : (\boldsymbol{u}_1(1), \boldsymbol{u}_1(2), \cdots, \boldsymbol{u}_1(p), \boldsymbol{u}_2(1), \cdots, \boldsymbol{u}_k(p)) \in S_{\xi}^d \times S_{\xi}^d \times \cdots \times S_{\xi}^d\}$ with size $|S_{\xi}^d|^{kp}$. Then for each $f(\boldsymbol{a}) = \sum_{j=1}^{k} \lambda_j \prod_{q=1}^{p}(\boldsymbol{v}_j(q)^{\top}\boldsymbol{a}(q)) \in \mathcal{F}_{ASYM}$, as we can find $\boldsymbol{u}_j(q) \in S_{\xi}^d$ that $\|\boldsymbol{u}_j(q) - \boldsymbol{v}_j(q)\|_2 \leq \xi$,

$$\begin{aligned}\sup_{\boldsymbol{a}}[f(\boldsymbol{a}) - g(\boldsymbol{a})] &\leq \sup_{\boldsymbol{a}}[\sum_{j=1}^{k}|\lambda_j|\sum_{q=1}^{p}|\boldsymbol{u}_j(q)^{\top}\boldsymbol{a} - \boldsymbol{v}_j(q)^{\top}\boldsymbol{a}| \cdot \\ & \quad |\prod_{r<q}(\boldsymbol{u}_j(r)^{\top}\boldsymbol{a})\prod_{r>q}(\boldsymbol{v}_j(r)^{\top}\boldsymbol{a})|] \\ &\leq pk\xi = \alpha;\end{aligned}$$

- $\mathcal{F}_{EV}$: the construction follows that of $\mathcal{F}_{SYM}$ by taking $p = 2$;

- $\mathcal{F}_{LR}$: taking the construction of $\mathcal{F}_{SYM}$ with $p = 2$ and $\xi = \frac{\alpha}{2k}$, for $\boldsymbol{N} = \sum_{j=1}^{k} \lambda_j \boldsymbol{u}_j \boldsymbol{u}_j^{\top}$ and $\boldsymbol{M} = \sum_{j=1}^{k} \lambda_j \boldsymbol{v}_j \boldsymbol{v}_j^{\top}$ with $\|\boldsymbol{u}_j - \boldsymbol{v}_j\|_2 \leq \xi$, we know $\|\boldsymbol{N} - \boldsymbol{M}\|_F \leq \left\|\boldsymbol{N} - \sum_{j=1}^{k} \lambda_j \boldsymbol{u}_j \boldsymbol{v}_j^{\top}\right\|_F + \left\|\sum_{j=1}^{k} \lambda_j \boldsymbol{u}_j \boldsymbol{v}_j^{\top} - \boldsymbol{M}\right\|_F \leq \sum_{j=1}^{k} 2|\lambda_j|\xi \leq \alpha$. Then $\sup_{\boldsymbol{A}}[f_{\boldsymbol{M}}(\boldsymbol{A}) - f_{\boldsymbol{N}}(\boldsymbol{A})] \leq \sup_{\boldsymbol{A}} \|\boldsymbol{M} - \boldsymbol{N}\|_F \cdot \|\boldsymbol{A}\|_F \leq \alpha$.

Then we can bound the covering numbers in Theorem 2.1. Notice that in the settings the log-covering numbers are only different by constant factors. $\square$

**The eluder dimensions**

**Lemma B.2.** *The $\epsilon$-eluder-dimension ($\epsilon < 1$) $d_E$ of the function classes are: $d_E(\mathcal{F}_{SYM}) = \widetilde{\Theta}(d^p)$ (for $k \geq p$), $d_E(\mathcal{F}_{ASYM}) = \widetilde{\Theta}(d^p)$, $d_E(\mathcal{F}_{EV}) = \widetilde{\Theta}(d^2)$, and $d_E(\mathcal{F}_{LR}) = \widetilde{\Theta}(d^2)$. In the settings WLOG we assume the top eigenvalue is $r^* = \lambda_1 = 1$ as we are mostly interested in the cases where $r^* > \epsilon$.*

*Proof.* The upper bounds for the eluder dimension can be given by the linear argument. [62] show that the $d$-dimension linear model $\{f_{\boldsymbol{\theta}}(\boldsymbol{a}) = \boldsymbol{\theta}^\top \boldsymbol{a}\}$ has $\epsilon$-eluder-dimension $O(d \log \frac{1}{\epsilon})$. In all of these settings, we can find feature maps $\phi$ and $\psi$ so that $\mathcal{F} = \{f_{\boldsymbol{\theta}}(\boldsymbol{a}), f_{\boldsymbol{\theta}}(\boldsymbol{a}) = \phi(\boldsymbol{\theta})^\top \psi(\boldsymbol{a}), \|\phi(\boldsymbol{\theta})\|_2 \leq k, \|\psi(\boldsymbol{a})\|_2 \leq k\}$. Then the eluder dimensions will be bounded by the corresponding linear dimension as an original $\epsilon$-independent sequence $\{\boldsymbol{a}_i\}$ will induce an $\epsilon$-independent sequence $\{\psi(\boldsymbol{a}_i)\}$ in the linear model. Therefore for matrices ($\mathcal{F}_{LR}$ and $\mathcal{F}_{EV}$) the eluder dimension is $O(d^2 \log \frac{k}{\epsilon})$ and for the tensors ($\mathcal{F}_{SYM}$ and $\mathcal{F}_{ASYM}$) it is $O(d^p \log \frac{k}{\epsilon})$.

Then we consider the lower bounds. We provide the following example of $O(1)$-independent sequences to bound the eluder dimension in our settings up to a log factor.

- $\mathcal{F}_{SYM}$: the sequence is $\{\boldsymbol{a}_i = (\boldsymbol{e}_{i_1}, \boldsymbol{e}_{i_2}, \cdots, \boldsymbol{e}_{i_p}) : i = (i_1, i_2, \cdots, i_p) \in [d]^p\}$. For $f_j(\boldsymbol{a}) = \prod_{q=1}^{p} \boldsymbol{e}_{j_q}^\top \boldsymbol{a}(q)$, $f_j(\boldsymbol{a}_i)$ is only 1 when $i = j$ and 0 otherwise. Then each $\boldsymbol{a}_i$ is 1-independent to the predecessors on $f_i$ and zero, and thus the eluder dimension is lower bounded by $d^p$.

- $\mathcal{F}_{ASYM}$: for $p \leq d$ and $k \geq p$, the sequence is $\{\boldsymbol{a}_i = \frac{1}{\sqrt{p}}(\boldsymbol{e}_{i_1} + \boldsymbol{e}_{i_2} + \cdots + \boldsymbol{e}_{i_p}) : i = (i_1, i_2, \cdots, i_p) \in [d]^p, i_1 < i_2 < \cdots < i_p\}$. There are tensors $f_j$ and $g_j$ of CP-rank $k$ that $(f_j - g_j)(\boldsymbol{a}) = \prod_{q=1}^{p}(\boldsymbol{e}_{j_q}^\top \boldsymbol{a})$ where $j_1 < j_2 < \cdots < j_p$, $(f_j - g_j)(\boldsymbol{a}_i)$ is only 1 when $i = j$ and 0 otherwise. Then each $\boldsymbol{a}_i$ is 1-independent to the predecessors on $f_i$ and $g_i$, and thus the eluder dimension is lower bounded by $\binom{d}{p}$.

- $\mathcal{F}_{EV}$: the sequence is $\{\boldsymbol{a}_i = \frac{1}{\sqrt{2}}(\boldsymbol{e}_{i_1} + \boldsymbol{e}_{i_2}) : i = (i_1, i_2) \in [d]^2, i_1 \leq i_2\}$. For $f_j(\boldsymbol{a}) = \frac{1}{2}\boldsymbol{a}^\top(\boldsymbol{e}_{j_1} + \boldsymbol{e}_{j_2})(\boldsymbol{e}_{j_1} + \boldsymbol{e}_{j_2})^\top \boldsymbol{a}$ and $g_j(\boldsymbol{a}) = \frac{1}{2}\boldsymbol{a}^\top(\boldsymbol{e}_{j_1} - \boldsymbol{e}_{j_2})(\boldsymbol{e}_{j_1} - \boldsymbol{e}_{j_2})^\top \boldsymbol{a}$ with $j_1 \leq j_2$, $(f_j - g_j)(\boldsymbol{a}_i)$ is only 1 when $i = j$ and 0 otherwise. Then each $\boldsymbol{a}_i$ is 1-independent to the predecessors on $f_i$ and $g_i$, and thus the eluder dimension is lower bounded by $\binom{d}{2}$.

- $\mathcal{F}_{LR}$: the sequence is $\{\boldsymbol{A}_i = \frac{1}{2}\boldsymbol{e}_{i_1}\boldsymbol{e}_{i_2}^T + \boldsymbol{e}_{i_1}\boldsymbol{e}_{i_2}^T : i = (i_1, i_2) \in [d]^2, i_1 \leq i_2\}$. For $f_j(\boldsymbol{A}) = \langle \frac{1}{2}(\boldsymbol{e}_{j_1}\boldsymbol{e}_{j_2}^T + \boldsymbol{e}_{j_2}\boldsymbol{e}_{j_1}^T), \boldsymbol{A} \rangle$ with $j_1 \leq j_2$, $f_j(\boldsymbol{A}_i)$ is only 1 when $i = j$ and 0 otherwise. Then each $\boldsymbol{A}_i$ is 1-independent to the predecessors on $f_i$ and zero, and thus the eluder dimension is lower bounded by $\binom{d}{2}$.

$\square$

Then we are all set for the results in the first line of 1. Notice that when we choose $\alpha = O(1/T^2)$ and $\epsilon = O(1/T^2)$ in our analysis of Algorithm 2, the regret upper bound would only expand by $\log(T)$ factors.

## C Omitted Proofs for Quadratic Reward

In this section we include all the omitted proof of the theorems presented in the main paper.

### C.1 Omitted Proofs of Main Results for Stochastic Bandit Eigenvector Problem

*Proof of Theorem 3.3.* Notice in Algorithm 1, for each iterate $\boldsymbol{a}$, its next iterate $\boldsymbol{y}$ satisfies

$$\boldsymbol{y} = \frac{1}{n_s} \sum_{i=1}^{n_s} (\boldsymbol{a}/2 + \boldsymbol{z}_i/2)^\top \boldsymbol{M}(\boldsymbol{a}/2 + \boldsymbol{z}_i/2)\boldsymbol{z}_i + \eta_i \boldsymbol{z}_i$$

$$= \frac{m_s}{n_s} \sum_{i=1}^{n_s} (\frac{1}{4}\boldsymbol{a}^\top \boldsymbol{M}\boldsymbol{a} + \frac{1}{2}\boldsymbol{a}^\top \boldsymbol{M}\boldsymbol{z}_i + \eta_i)\boldsymbol{z}_i.$$

Therefore $\mathbb{E}[\boldsymbol{y}] = \frac{1}{2}\boldsymbol{M}\boldsymbol{a}$. We can write $2\boldsymbol{y} = \boldsymbol{M}\boldsymbol{a} + \boldsymbol{g}$ where $\boldsymbol{g} := \frac{m_s}{n_s}\sum_{i=1}^{n_s}(\frac{1}{2}\boldsymbol{a}^\top\boldsymbol{M}\boldsymbol{a} + 2\eta_i)\boldsymbol{z}_i$. With Claim D.12 and Claim D.11 we get that $\|\boldsymbol{g}\| \leq C\sqrt{\frac{m_s\log^2(n/\delta)\log(d/\delta)d}{n_s}}$. Therefore with our choice of $n_s \geq \widetilde{\Theta}(\frac{d^2}{\varepsilon_s^2(\lambda_1 - |\lambda_2|)^2})$ we guarantee $\|\boldsymbol{g}\| \leq \varepsilon_s(\lambda_1 - |\lambda_2|)$. Therefore it satisfies the requirements for noisy power method, and by applying Corollary C.4, we have with $L = O(\kappa\log(d/\varepsilon))$ iterations we will be able to find $\|\widehat{\boldsymbol{a}} - \boldsymbol{a}^*\| \leq \varepsilon$. By setting $\delta < 0.1/L$ in the algorithm we can guarantee the whole process succeed with high probability. Altogether it is sufficient to take $Ln_s = \widetilde{O}(\kappa d^2/(\varepsilon\Delta)^2)$ actions to get an $\varepsilon$-optimal arm.

Finally to get the cumulative regret bound, we apply Claim D.7 with $A = \frac{d^2\kappa}{\Delta^2}$ and $a = 2$. Therefore we set $\varepsilon = A^{1/4}T^{-1/4} = \frac{d^{1/2}\kappa^{1/4}}{\Delta^{1/2}T^{1/4}}$ and get:

$$\mathrm{Reg}(T) \lesssim T^{1/2}A^{1/2}r^* = \sqrt{\frac{d^2\kappa}{\Delta^2}T}r^* = \sqrt{d^2\kappa^3 T}.$$

$\square$

**Corollary C.1** (Formal statement for Corollary 3.6). *In Algorithm 1, by setting $\alpha = 1 - \varepsilon^2/2$, one can get $\varepsilon$-optimal reward with a total of $\widetilde{O}(d^2\lambda_1^2/\varepsilon^4)$ total samples to get $\boldsymbol{a}$ such that $r^* - f(\boldsymbol{a}) \leq \varepsilon$. Therefore one can get an accumulative regret of $\widetilde{O}(\lambda_1^{3/5}d^{2/5}T^{4/5})$.*

*Proof of Lemma 3.6.* In order to find an arm with $\lambda_1\varepsilon^2$-optimal reward, one will want to recover an arm that is $\varepsilon/2$-close (meaning to find an $\boldsymbol{a}$ such that $\tan\theta(\boldsymbol{V}_l, \boldsymbol{a}) \leq \varepsilon/2$) to the top eigenspace $\mathrm{span}(\boldsymbol{v}_1, \cdots \boldsymbol{v}_l)$, where $l$ satisfies $\lambda_l \geq \lambda_1 - \widetilde{\varepsilon}$ and $\lambda_{l+1} \leq \lambda_1 - \widetilde{\varepsilon}$. Here we set $\widetilde{\varepsilon} := \lambda_1\varepsilon^2/2$. We first show 1) this is sufficient to get an $\lambda_1\varepsilon$-optimal reward, and next show 2) how to set parameter to achieve this.

To get 1), we write $\boldsymbol{V}_l = [\boldsymbol{v}_1, \cdots \boldsymbol{v}_l] \in \mathbb{R}^{d\times l}$ and $\boldsymbol{V}_l^\perp = [\boldsymbol{v}_{l+1}, \cdots \boldsymbol{v}_k]$. When $\tan\theta(\boldsymbol{V}_l, \boldsymbol{a}_T) = \|\boldsymbol{V}^\perp\boldsymbol{a}\|/\|\boldsymbol{V}\boldsymbol{a}\| \leq \varepsilon/2$, from the proof of Claim D.6, we get $r^* - f(\boldsymbol{a}) \leq \min\{\lambda_1, \lambda_1 2(\varepsilon/2)^2 + \widetilde{\varepsilon}\} = \lambda_1\varepsilon^2$.

Now to get 2), we note that in each iteration we try to conduct the power iteration to find an action $\tan\theta(\boldsymbol{V}_l, \widehat{\boldsymbol{a}}) \leq \varepsilon/2$ and with eigengap $\geq \widetilde{\varepsilon} := \lambda_1\varepsilon^2/2$. Therefore it is sufficient to let $\|\boldsymbol{g}\| \leq 0.1\widetilde{\varepsilon}\varepsilon$ and $|\boldsymbol{v}_1^\top\boldsymbol{g}| \leq 0.1\widetilde{\varepsilon}\frac{1}{\sqrt{d}}$, and thus $n_s \geq \widetilde{\Theta}(\frac{d^2}{\varepsilon^2\widetilde{\varepsilon}^2}) \leq \widetilde{\Theta}(d^2/\lambda_1^2\varepsilon^6)$. Together we need $\lambda_1/\widetilde{\epsilon}\log(2d/\varepsilon)n_s = \widetilde{\Theta}(d^2/\lambda_1^2\varepsilon^8)$ samples to get an $\lambda_1\varepsilon^2$-optimal reward. Namely we get $\widetilde{\varepsilon}$-optimal reward with $\widetilde{O}(d^2\lambda_1^2/\widetilde{\varepsilon}^4)$ samples.

Finally by applying Claim D.7 we get:

$$\mathfrak{R}(T) \lesssim (d^2\lambda_1^2)^{\frac{1}{5}}T^{\frac{4}{5}}\lambda_1^{\frac{1}{5}} \leq \widetilde{O}(\lambda_1^{3/5}d^{2/5}T^{4/5}).$$

$\square$

**Theorem C.2** (Formal statement of Theorem 3.7). *In Algorithm 3, if we set $n = \widetilde{\Theta}(\frac{d^2\lambda_1^2}{\varepsilon^2\lambda_k^2}), m = d\log(n/\delta), L = \Theta(\log(d/\varepsilon)), \delta = 0.1/L$, we will be able to identify an action $\widehat{\boldsymbol{a}}$ that yield at most $\varepsilon$-regret with probability $0.9$. Therefore by applying the standard PAC to regret conversion as discussed in Claim D.7 we get a cumulative regret of $\widetilde{O}(\lambda_1^{1/3}k^{1/3}(\widetilde{\kappa}dT)^{2/3})$ for large enough $T$, where $\widetilde{\kappa} = \lambda_1/|\lambda_k|$.*

*On the other hand, we set $n = \widetilde{\Theta}(\frac{d^2k^2}{\varepsilon^2})$ and keep the other parameters. If we play Algorithm 3 $k$ times by setting $k' = 2, 4, 6, \cdots 2k$ and select the best output among them, we can get a gap-free cumulative regret of $\widetilde{O}(\lambda_1^{1/3}k^{4/3}(dT)^{2/3})$ for large enough $T$ with high probability.*

*Proof of Theorem 3.7.* First we show the first setting identify an $\varepsilon$-optimal reward with $\widetilde{O}(\widetilde{\kappa}^2d^2k\epsilon^{-2})$ samples.

Similarly as Theorem 3.8, when setting $n \geq \widetilde{\Theta}(d^2/(\sigma_k^2\widetilde{\epsilon}^2))$, we can find $\boldsymbol{X}_L$ that satisfies $\|(\boldsymbol{X}_L\boldsymbol{X}_L^\top - \boldsymbol{I})\boldsymbol{U}\| \leq \widetilde{\epsilon}$, and therefore we recover an $\boldsymbol{Y}_L = \boldsymbol{M}\boldsymbol{X}_{L-1} + \boldsymbol{G}_L$ with $\|\boldsymbol{G}_L\| \leq \sigma_k\widetilde{\varepsilon}$ and

---

**Algorithm 3** Gap-free Subspace Iteration for Bilinear Bandit

---

1: **Input:** Quadratic reward $f : \mathcal{X} \to \mathbb{R}$ generating noisy reward, failure probability $\delta$, error $\varepsilon$.
2: **Initialization:** Set $k' = 2k$. Initial candidate matrix $\boldsymbol{X}_0 \in \mathbb{R}^{d \times k'}$, $\boldsymbol{X}_0(j) \in \mathbb{R}^d, j = 1, 2, \cdots k'$ is the $j$-th column of $X_0$ and are i.i.d sampled on the unit sphere $\mathbb{S}^{d-1}$ uniformly. Sample variance $m$, # sample per iteration $n$, total iteration $L$.
3: **for** Iteration $l$ from 1 to $L$ **do**
4:     **for** $s$ from 1 to $k'$ **do**
5:         **Noisy subspace iteration:**
6:         Sample $\boldsymbol{z}_i \sim \mathcal{N}(0, 1/mI_d), i = 1, 2, \cdots n_s$.
7:         Calculate tentative rank-1 arms $\widetilde{\boldsymbol{a}}_i = \frac{1}{2}(\boldsymbol{X}_{l-1}(s) + \boldsymbol{z}_i)$.
8:         Conduct estimation $\boldsymbol{Y}_l(s) \leftarrow 4m/n \sum_{i=1}^n (f(\widetilde{\boldsymbol{a}}_i) + \eta_i)\boldsymbol{z}_i.$ ($\boldsymbol{Y}_l \in \mathbb{R}^{d \times k'}$)
9:     Let $\boldsymbol{Y}_l = \boldsymbol{X}_l \boldsymbol{R}_l$ be a QR-factorization of $\boldsymbol{Y}_l$
10:     Update target arm $\boldsymbol{a}_l \leftarrow \arg\max_{\|\boldsymbol{a}\|=1} \boldsymbol{a}^\top \boldsymbol{Y}_l \boldsymbol{X}_{l-1}^\top \boldsymbol{a}$.
11: **Output:** $\boldsymbol{a}_L$.

---

$\|\boldsymbol{Y}_L \boldsymbol{X}_{L-1}^\top - \boldsymbol{M}\|_2 = \|\boldsymbol{M}\boldsymbol{X}_{L-1}\boldsymbol{X}_{L-1}^\top - \boldsymbol{M} + \boldsymbol{G}_L \boldsymbol{X}_{L-1}^\top\|_2 \leq (\lambda_1 + |\lambda_k|)\widetilde{\varepsilon}$. Therefore by definition of $\boldsymbol{a}_L, \boldsymbol{a}_L^\top \boldsymbol{Y}_L \boldsymbol{X}_{L-1}^\top \boldsymbol{a}_L = \max_{\|\boldsymbol{a}\|=1} \boldsymbol{a}^\top (\boldsymbol{M}\boldsymbol{X}_{L-1}\boldsymbol{X}_{L-1}^\top + \boldsymbol{G}_L \boldsymbol{X}_{L-1}^\top)\boldsymbol{a} \geq \lambda_1 - (\lambda_1 + |\lambda_k|)\widetilde{\varepsilon}$. Therefore $\boldsymbol{a}_L^\top \boldsymbol{M} \boldsymbol{a}_L \geq \lambda_1 - 2(\lambda_1 + |\lambda_k|)\widetilde{\varepsilon}$. Therefore we set $2(\lambda_1 + |\lambda_k|)\widetilde{\varepsilon} = \epsilon$, i.e., $\widetilde{\varepsilon} = 0.5\epsilon/(\lambda_1 + |\lambda_k|)$ which will get a total sample of $T = \widetilde{\Theta}(kn) = \widetilde{\Theta}(d^2 \widetilde{\kappa}^2 k \varepsilon^{-2})$. Then by applying Claim D.7 we get the cumulative regret bound.

Next we show how to estimate the action with $\widetilde{O}(d^2 k^4 \varepsilon^{-2})$ samples. To achieve this result, we need to slightly alter Algorithm 3 where we respectively set $k' = 2, 4, 6, \cdots 2k$ and keep the best arm among the $k$ outputs. We argue that among all the choices of $k'$, at least for one $l \in [k], k' = 2l$, we have $|\lambda_l| - |\lambda_{l+1}| \geq \lambda_1/k$. Notice with similar argument as above, when we set $n = \widetilde{\Theta}(d^2 \lambda_l^{-2} \widetilde{\varepsilon}^{-2}) \leq \widetilde{\Theta}(d^2 k^2 \lambda_1^{-2} \widetilde{\varepsilon}^{-2})$ we can get $\|\boldsymbol{G}\| \leq \widetilde{\varepsilon}\lambda_l$ as required by Corollary C.4, the total number of iterations $L = O(\sigma_l/(\sigma_l - \sigma_{l+1})) \log(2d/\epsilon) = \widetilde{O}(k)$. Finally by setting $\widetilde{\varepsilon} = \epsilon/(4\lambda_1)$ we get the overall samples we required is $\widetilde{O}(k^2 n) = \widetilde{O}(d^2 k^4 \epsilon^{-2})$.

For both settings, directly applying our arguments in the PAC to regret conversion: Claim D.7 will finish the proof. $\qquad\square$

## C.2   Omitted Details of Main Results of Low-Rank Linear Reward

---

**Algorithm 4** Subspace Iteration Exploration for Low-rank Linear Reward.

---

1: **Input:** Quadratic function $f : \mathcal{A} \to \mathbb{R}$ with noisy reward, failure probability $\delta$, error $\varepsilon$.
2: **Initialization:** Set $k' = 2k$. Initial candidate matrix $\boldsymbol{X}_0 \in \mathbb{R}^{d \times k'}$, $\boldsymbol{X}_0(j) \in \mathbb{R}^d, j = 1, 2, \cdots k'$ is the $j$-th column of $\boldsymbol{X}_0$ and are i.i.d sampled on the unit sphere $\mathbb{S}^{d-1}$ uniformly. Sample variance $m$, # sample per iteration $n$, total iteration $L$.
3: **for** Iteration $l$ from 1 to $L$ **do**
4:     Sample $\boldsymbol{z}_i \sim \mathcal{N}(0, 1/mI_d), i = 1, 2, \cdots n$.
5:     **for** $s$ from 1 to $k'$ **do**
6:         **Noisy subspace iteration:**
7:         Calculate tentative rank-1 actions $\widetilde{\boldsymbol{A}}_i = \boldsymbol{X}_{l-1}(s)\boldsymbol{z}_i^\top$.
8:         Conduct estimation $\boldsymbol{Y}_l(s) \leftarrow m/n \sum_{i=1}^n (\langle \boldsymbol{M}, \widetilde{\boldsymbol{A}}_i \rangle + \eta_{i,s})\boldsymbol{z}_i.$ ($\boldsymbol{Y}_l \in \mathbb{R}^{d \times k'}$)
9:     Let $\boldsymbol{Y}_l = \boldsymbol{X}_l \boldsymbol{R}_l$ be a QR-factorization of $\boldsymbol{Y}_l$
10:     Update target action $\boldsymbol{A}_l \leftarrow \boldsymbol{Y}_l \boldsymbol{X}_l^\top$.
11: **Output:** $\widehat{\boldsymbol{A}} = \boldsymbol{A}_L/\|\boldsymbol{A}_L\|_F$

---

**Theorem C.3** (Formal statement of Theorem 3.8). *In Algorithm 4, for large enough constants $C_n, C_L, C_m$, let $n = C_n d^2 \log^2(d/\delta)\sigma_k^{-2}\varepsilon^{-2}$, $m = C_m d \log(n/\delta)$, and $L = C_L \log(d/\varepsilon)$, $\boldsymbol{X}_L$*

*satisfies* $\|(\boldsymbol{I} - \boldsymbol{X}_L\boldsymbol{X}_L^\top)\boldsymbol{V}\| \le \varepsilon/4$, *and the output* $\widehat{\boldsymbol{A}}$ *satisfies* $\|\widehat{\boldsymbol{A}} - \boldsymbol{A}^*\|_F \le \|\boldsymbol{M}\|_F\varepsilon$. *Altogether to get an $\varepsilon$-optimal action, it is sufficient to have total sample complexity of* $T \le \widetilde{O}(d^2k\lambda_k^{-2}\varepsilon^{-2})$.

*Proof of Theorem 3.8 .* Let $\boldsymbol{M} = \boldsymbol{V}\boldsymbol{\Sigma}\boldsymbol{V}^\top$. From Claim C.6 we get that for each noisy subspace iteration step we get $\boldsymbol{Y}_l = \boldsymbol{M}\boldsymbol{X}_l + \boldsymbol{G}_l$ with $5\|\boldsymbol{G}_l\| \le \varepsilon\sigma_k$ and $\|\boldsymbol{V}^\top\boldsymbol{G}\| \le \sigma_k\sqrt{k}/3\sqrt{d} \le \sigma_k(\sqrt{2k} - \sqrt{k})/2\sqrt{d}$. Therefore we can apply Corollary C.4, and get $\|\boldsymbol{V}(\boldsymbol{X}_L\boldsymbol{X}_L^\top - \boldsymbol{I})\| \le \varepsilon/4$ with $O(\log 2d/\epsilon)$ steps. Therefore we have:

$$\begin{aligned}
\|\boldsymbol{A}_L - \boldsymbol{M}\|_F =& \|(\boldsymbol{M}\boldsymbol{X}_L + \boldsymbol{G}_L)\boldsymbol{X}_L^\top - \boldsymbol{M}\|_F = \|\boldsymbol{V}^\top\boldsymbol{\Sigma}\boldsymbol{V}(\boldsymbol{X}_L\boldsymbol{X}_L^\top - \boldsymbol{I})) + \boldsymbol{G}_L\boldsymbol{X}_L^\top\|_F \\
\le& \|\boldsymbol{M}\|_F\|\boldsymbol{V}(\boldsymbol{X}_L\boldsymbol{X}_L^\top - \boldsymbol{I})\| + \|\boldsymbol{G}_L\|\|\boldsymbol{X}_L\|_F \\
\le& (\|\boldsymbol{M}\|_F + \sigma_k)\varepsilon/4 < \|\boldsymbol{M}\|_F\varepsilon/2.
\end{aligned}$$

Meanwhile, notice $\|\boldsymbol{A}^*\|_F = 1, \|\boldsymbol{M}\|_F = r^*$ and $\|\widehat{\boldsymbol{A}}\|_F = 1$. $\|\boldsymbol{A}_L/r^* - \boldsymbol{A}^*\|_F \le \varepsilon/2$. $\|\widehat{\boldsymbol{A}} - \boldsymbol{A}^*\|_F = \|\boldsymbol{A}_L/\|\boldsymbol{A}_L\|_F - \boldsymbol{A}^*\|_F = \|\text{vec}(\boldsymbol{A}_L)/\|\text{vec}(\boldsymbol{A}_L)\|_2 - \text{vec}(\boldsymbol{A}^*)\|_2$.

Write $\theta_A := \theta(\text{vec}(\boldsymbol{A}_L), \text{vec}(\boldsymbol{A}^*))$. The worst case that makes $\|\text{vec}(\widehat{\boldsymbol{A}}) - \text{vec}(\boldsymbol{A}^*)\|$ to be larger than $\|\text{vec}(\boldsymbol{A}_L/r^*) - \text{vec}(\boldsymbol{A}^*)\|$ is when $\|\text{vec}(\boldsymbol{A}_L/r^*) - \text{vec}(\boldsymbol{A}^*)\| = \sin\theta_A$ and $\|\text{vec}(\widehat{\boldsymbol{A}}) - \text{vec}(\boldsymbol{A}^*)\|$ is always $2\sin(\theta_A/2)$. Notice trivially $2\sin(\theta_A/2) \le 2\sin(\theta_A)$ Therefore we could get $\|\widehat{\boldsymbol{A}} - \boldsymbol{A}^*\|_F \le 2\|\boldsymbol{A}_L/r^* - \boldsymbol{A}^*\|_F \le \varepsilon$.

$\square$

*Proof of Corollary 3.9.* The corollary uses a special property of the strongly convex action set that ensures: $\boldsymbol{A}^* = \boldsymbol{M}/r^*$. With $\widehat{\boldsymbol{A}}$ that satisfies $\|\widehat{\boldsymbol{A}}\|_F = 1$, we have

$$\begin{aligned}
r^* - f_{\boldsymbol{M}}(\boldsymbol{A}) =& r^* - \langle\widehat{\boldsymbol{A}}, \boldsymbol{M}\rangle = r^* - \langle\widehat{\boldsymbol{A}}, r^*\boldsymbol{A}^*\rangle \\
=& \frac{r^*}{2}(2 - 2\langle\widehat{\boldsymbol{A}}, \boldsymbol{A}^*\rangle) = \frac{r^*}{2}(\|\widehat{\boldsymbol{A}}\|_F^2 + \|\boldsymbol{A}^*\|_F^2 - \langle\widehat{\boldsymbol{A}}, \boldsymbol{A}^*\rangle) \\
=& \frac{r^*}{2}\|\widehat{\boldsymbol{A}} - \boldsymbol{A}^*\|_F^2 \le \frac{r^*\varepsilon^2}{2}
\end{aligned} \tag{2}$$

Therefore, with first $T_1 = \widetilde{O}(d^2k\lambda_k^{-2}\varepsilon^{-2})$ exploratory samples we get $r^* - f(\widehat{\boldsymbol{A}}) \le r^*\varepsilon^2/2 = r^*\sqrt{\frac{d^2k}{\lambda_k^2 T}} = \sqrt{\frac{(r^*)^2d^2k}{\lambda_k^2 T}}$. Together we have:

$$\begin{aligned}
\mathfrak{R}(T) =& \sum_{t=1}^{T_1} r^* - f(\boldsymbol{A}_t) + \sum_{t=T_1+1}^{T} r^* - f(\widehat{\boldsymbol{A}}) \\
<& r^*T_1 + Tr^*\varepsilon^2 \\
\le& \widetilde{O}(\sqrt{d^2k(r^*)^2\lambda_k^{-2}T}).
\end{aligned}$$

$\square$

*Proof of Theorem 3.10.* We find an $l$ to be the smallest integer such that $\sum_{i=l+1}^{k}\sigma_i^2 \le \epsilon^2\|\boldsymbol{M}\|_F^2$. Then we have $\sigma_l \ge \epsilon/\sqrt{k-l} > \epsilon/\sqrt{k}$.

Notice that in Algorithm 4, we set $n \ge \widetilde{\Theta}(\frac{d^2k}{(r^*)^2\varepsilon^4})$ large enough such that $\|\boldsymbol{G}\|_2 \le O(\|\boldsymbol{M}\|_F\epsilon^2/\sqrt{k}) \lesssim \epsilon(\sigma_l - 0)$ and $\|\boldsymbol{U}^\top\boldsymbol{G}\|_2 \le \|\boldsymbol{M}\|_F\epsilon/\sqrt{k}\frac{\sqrt{k'} - \sqrt{k-1}}{2\sqrt{d}}$. (This comes from the argument proved in Claim C.6.)

Therefore by conducting noisy power method we get with $O(nk) = \widetilde{O}(\frac{d^2k^2}{(r^*)^2\varepsilon^4})$ samples we can get an action $\widehat{\boldsymbol{A}}$ that satisfies:

$$\|\boldsymbol{M} - \boldsymbol{X}_L\boldsymbol{X}_L^\top\boldsymbol{M}\|_F^2 \le \sum_{i=l+1}^{k}\sigma_i^2 + l\epsilon^2\sigma_l^2 \le 2\|\boldsymbol{M}\|_F^2\epsilon^2.$$

Therefore we could get $\|A^* - \widehat{A}\| \leq 2\epsilon$, and with similar argument as (2) we have $r^* - f(\widehat{A}) \leq \|M\|_F \epsilon^2$.

Therefore if we want to take a total of $T$ actions, we will set $\epsilon^6 = \widetilde{\Theta}(\frac{d^2 k^2}{(r^*)^2 T})$ and we get:

$$\mathfrak{R}(T) = \sum_{t=1}^{T_1} r^* - f(A_t) + \sum_{t=T_1+1}^{T} r^* - f(\widehat{A})$$

$$< r^* T_1 + T r^* \varepsilon^2$$

$$\leq \widetilde{O}(d^{2/3} k^{2/3} (r^*)^{1/3} T^{2/3}).$$

$\square$

### C.3 Technical Details for Quadratic Reward

**Noisy Power Method.**

**Corollary C.4** (Adapted from Corollary 1.1 from [37]). *Let $k' \geq l$. Let $U \in \mathbb{R}^{d \times l}$ represent the top $l$ singular vectors of $M$ and let $\sigma_1 \geq \cdots \geq \sigma_k > 0$ denote its singular values. Suppose $X_0$ is an orthonormal basis of a random $k'$-dimensional subspace. Further suppose that at every step of NPM we have*

$$5\|G\| \leq \epsilon(\sigma_l - \sigma_{l+1}),$$

$$\text{and } 5\|U^\top G\| \leq (\sigma_l - \sigma_{l+1}) \frac{\sqrt{k'} - \sqrt{l-1}}{2\sqrt{d}}$$

*for some fixed parameter $\epsilon < 1/2$. Then with all but $2^{-\Omega(k'+1-l)} + e^{\Omega(d)}$ probability, there exists an $L = O(\frac{\sigma_l}{\sigma_l - \sigma_{l+1}} \log(2d/\epsilon))$ so that after $L$ steps we have that $\|(I - X_L X_L^\top)U\| \leq \epsilon$.*

**Theorem C.5** (Adapted from Theorem 2.2 from [11]). *Let $U_l \in \mathbb{R}^{d \times l}$ represent the top $l$ singular vectors of $M$ and let $\sigma_1 \geq \cdots \geq \sigma_k > 0$ denote its singular values. Naturally $l \leq k$. Suppose $X_0$ is an orthonormal basis of a random $k'$-dimensional subspace where $k' \geq k$. Further suppose that at every step of NPM we have*

$$\|G\| \leq O(\epsilon \sigma_l),$$

$$\text{and } \|U_k^\top G\|_2 \leq O(\sigma_l \frac{\sqrt{k'} - \sqrt{k-1}}{2\sqrt{d}})$$

*for small enough $\epsilon$. Then with all but $2^{-\Omega(k'+1-k)} + e^{\Omega(d)}$ probability, there exists an $L = O(\log(2d/\epsilon))$ so that after $L$ steps we have that $\|(I - X_L X_L^\top)U_l\| \leq \epsilon$. Furthermore:*

$$\|M - X_L X_L^\top M\|_F^2 \leq \sum_{i=l+1}^{k} \sigma_i^2 + l\sigma^2 \sigma_l^2.$$

**Concentration Bounds.**

**Claim C.6.** *Write the eigendecomposition for $M$ as $M = U\Sigma U^\top$. In Algorithm 4, when $n \geq \widetilde{\Theta}(d^2/(\lambda_k^2 \varepsilon^2))$, the noisy subspace iteration step can be written as: $Y_l = MX_{l-1} + G_l$, where the noise term satisfies:*

$$5\|G_l\| \leq \varepsilon |\lambda_k|$$

$$5\|U^\top G_l\| \leq \varepsilon |\lambda_k| \frac{\sqrt{k}}{3\sqrt{d}}.$$

*with high probability for our choice of $n$.*

*Proof.* For compact notation, write vector $\boldsymbol{\eta}_i := [\eta_{i,1}, \eta_{i,2}, \cdots \eta_{i,k'}]^\top \in \mathbb{R}^{k'}$. We have:

$$G_l(s) = \frac{m}{n} \sum_{i=1}^{n} (z_i^\top M X_l(s)) z_i + \frac{m}{n} \sum_{i=1}^{n} \eta_{i,s} z_i - M X_l(s), \text{ therefore}$$

$$G_l = (\frac{m}{n} \sum_{i=1}^{n} [z_i z_i^\top] - I) M X_l + \frac{m}{n} \sum_{i=1}^{n} z_i \boldsymbol{\eta}_i^\top.$$

First note that for orthogonal matrix $X_l$, $\|MX_l\| \leq \lambda_1$, and $\|\frac{m}{n}\sum_{i=1}^{n}[z_i z_i^\top] - I\| \leq O(\sqrt{\frac{d+\log(1/\delta)}{n}})$. The bottleneck is from the second term and we will use Matrix Bernstein to concentrate it. Write $S_i = \frac{m}{n}z_i \eta_i^\top$. We have $\|S_i\| \leq O(\frac{\sqrt{mk'}\log(n/\delta)}{n})$ with probability $1 - \delta$ and $\mathbb{E}[\sum_i S_i S_i^\top] = \frac{mk'}{n}I_d$ and $\mathbb{E}[\sum_i S_i^\top S_i] = \frac{md}{n}I_{k'}$. Therefore with matrix Bernstein we can get that $\|\sum_i S_i\|_i \leq O(\sqrt{\frac{md}{n}}\log(d/\delta))$ with probability $1 - \delta$.

Therefore for $n \geq \widetilde{\Omega}(d^2/(\lambda_k^2 \varepsilon^2))$, we can get that $5\|G_l\| \leq \varepsilon|\lambda_k|$.

Similarly since $U^\top z_i \sim \mathcal{N}(0, \frac{1}{m}I_{k'})$, with the same argument one can easily get that $\|U^\top G_l\| \leq O(\sqrt{\frac{mk'}{n}}\log(d/\delta))$. Therefore with the same lower bound for $n$ one can get $15\|U^\top G_l\| \leq \varepsilon|\lambda_k|\sqrt{\frac{k}{d}}$. $\qquad\square$

## C.4 Omitted Proof for RL with Quadratic Q function

---

**Algorithm 5** Learn policy complete polynomial with simulator.

---

1: **Initialize:** Set $n = \widetilde{\Theta}(\widetilde{\kappa}^2 d^2 H^3/\varepsilon^2)$, Oracle to estimate $\widehat{T}_h$ from noisy observations.
2: **for** $h = H, \ldots 1$ **do**
3:     Sample $\phi(s_h^i, a_h^i), i \in [n]$ from standard Gaussian $N(0, I_d)$
4:     **for** $i \in [n]$ **do**
5:         Query $(s_h^i, a_h^i)$ and use $\pi_{h+1}, \ldots, \pi_H$ as the roll-out to get estimation $\widehat{Q}_h^{\pi_{h+1}, \ldots, \pi_H}(s_h^i, a_h^i)$
6:     Retrieve $\widehat{M}_h$ from estimation $\widehat{Q}_h^{\pi_{h+1}, \ldots, \pi_H}(s_h^i, a_h^i), i \in [n]$
7:     Set $\widehat{Q}_h(s, a) \leftarrow f_{\widehat{T}_h}$
8:     Set $\pi_h(s) \leftarrow \arg\max_{a \in \mathcal{S}} \widehat{Q}_h(s, a)$
9: **Return** $\pi_1, \ldots, \pi_H$

---

*Proof of Theorem 3.13.* With the oracle, at horizon $H$, we can estimate $\widehat{M}_H$ that is $\epsilon/H$ close to $M_H^*$ in spectral norm through noisy observations from the reward function with $\widetilde{O}(\widetilde{\kappa}^2 d^2 H^2/\varepsilon^2)$ samples. Next, for each horizon $h = H - 1, H - 1, \cdots, 1$, sample $s_i' \sim \mathbb{P}(\cdot|s, a)$, we define $\eta_i = \max_{a'} f_{\widehat{M}_{h+1}}(s_i', a') - \mathbb{E}_{s' \sim \mathbb{P}(\cdot|s,a)}\max_{a'} f_{\widehat{M}_{h+1}}(s', a')$. $\eta_i$ is mean-zero and $O(1)$-sub-gaussian since it is bounded. Denote $M_h$ as the matrix that satisfies $f_{M_h} := \mathcal{T}f_{\widehat{M}_{h+1}}$, which is well-defined due to Bellman completeness. We estimate $\widehat{M}_h$ from the noisy observations $y_i = r_h(s, a) + \max_{a'} f_{\widehat{M}_{h+1}}(s_i', a') = \mathcal{T}f_{\widehat{M}_{h+1}} + \eta_i =: f_{M_h} + \eta_i$. Therefore with the oracle, we can estimate $\widehat{M}_h$ such that $\|\widehat{M}_h - M_h\|_2 \leq \epsilon/H$ with $\Theta(\widetilde{\kappa}^2 d^2 k^2 H^2/\epsilon^2)$ bandits. Together we have:

$$
\begin{aligned}
\|f_{\widehat{M}_h} - f_{\widehat{M}_h^*}\|_\infty =& \|\widehat{M}_h - M_h^*\| \\
\leq& \|\widehat{M}_h - M_h\| + \|M_h - M_h^*\| \\
\leq& \epsilon/H + \|\mathcal{T}f_{\widehat{M}_{h+1}} - \mathcal{T}f_{M_{h+1}^*}\|_\infty \\
\leq& \epsilon/H + \|f_{\widehat{M}_{h+1}} - f_{M_{h+1}^*}\|_\infty \\
\leq& 2\epsilon/H + \|f_{\widehat{M}_{h+2}} - f_{M_{h+2}^*}\|_\infty \\
\leq& \cdots \\
\leq& (H - h)\epsilon/H.
\end{aligned}
$$

Finally for $h = 1$ we have $\|\widehat{M}_1 - M^*\| \leq \epsilon$ if we sample $n = \widetilde{\Theta}(\widetilde{\kappa}^2 d^2 k^2 H^2/\epsilon^2)$ for each $h \in [H]$. Therefore for all the $H$ timesteps we need $\Theta(\widetilde{\kappa}^2 d^2 k^2 H^3/\epsilon^2)$.

$\qquad\square$

# D Technical details for General Tensor Reward

---

**Algorithm 6** Phased elimination with zeroth order exploration.

---

1: **Input:** Function $f : \mathcal{A} \rightarrow \mathbb{R}$ of polynomial degree $p$ generating noisy reward, failure probability $\delta$, error $\varepsilon$.
2: **Initialization:** $L_0 = C_L k \log(1/\delta)$; Total number of stages $S = C_S \lceil \log(1/\varepsilon) \rceil + 1$, $\mathcal{A}_0 = \{a_0^{(1)}, a_0^{(2)}, \cdots a_0^{(L_0)}\}$ where each $a_0^{(l)}$ is uniformly sampled on the unit sphere $\mathbb{S}^{d-1}$. $\widetilde{\varepsilon}_0 = 1$.
3: **for** $s$ from 1 to $S$ **do**
4:     $\widetilde{\varepsilon}_s \leftarrow \widetilde{\varepsilon}_{s-1}/2$, $n_s \leftarrow C_n d^p \log(d/\delta)/\lambda_1^2 \widetilde{\varepsilon}_s^2$, $n_s \leftarrow n_s \cdot \log^3(n_s/\delta)$, $m_s \leftarrow C_m d \log(n_s/\delta)$, $\mathcal{A}_s = \varnothing$.
5:     **for** $l$ from 1 to $L_{s-1}$ **do**
6:         **Zeroth-order optimization:**
7:         Locate current action $\widetilde{a} = a_{s-1}^{(l)}$.
8:         **for** $\lceil (1/(1-\alpha)) \log(2d) \rceil$ times **do**
9:             Sample $z_i \sim \mathcal{N}(0, 1/m_s I_d), i = 1, 2, \cdots n_s$.
10:             Take actions $a_i = (1 - \frac{1}{2p})\widetilde{a} + \frac{1}{2p} z_i$ and observe $r_i = T(a_i) + \eta_i, i \in [n_s]$; Take actions $\frac{1}{2p} z_i$ and observe $r_i' = T(\frac{1}{2p} z_i) + \eta_i', i \in [n_s]$.
11:             Conduct estimation $y \leftarrow 1/n_s \sum_{i=1}^{n_s} (r_i - r_i') z_i$.
12:             Update the current action $\widetilde{a} \leftarrow y/\|y\|$.
13:         Estimate the expected reward for $\widetilde{a}$ through $n_s$ samples: $r_n(\widetilde{a}) = 1/n_s \sum_{i=1}^{n_s} (T(\widetilde{a}) + \eta_i)$.
14:         **Candidate Elimination:**
15:         **if** $r_n \geq \lambda_1(1 - p\widetilde{\varepsilon}_s^2)$ **then**
16:             Keep the action $\mathcal{A}_s \leftarrow \mathcal{A}_s \cup \{\widetilde{a}\}$
17:     Label the actions: $L_s = |\mathcal{A}_s|$, $\mathcal{A}_s =: \{a_s^{(1)}, \cdots a_s^{(L_s)}\}$.
18: Run UCB (Algorithm 7) with the candidate set $\mathcal{A}_S$.

---

## D.1 Technical Details for Symmetric Setting

**Lemma D.1** (Zeroth order optimization for noiseless setting). *For $p \geq 3$, suppose $0.5 a^\top v_1 > |a^\top v_j|$ for all $j \geq 2$, we have:*

$$\tan \theta(G(a), v_1) \leq \frac{1}{2} \tan \theta(a, v_1).$$

*Proof.* We first simplify $G(a) = \sum_{j=1}^r \lambda_j v_j \cdot S_j$, where

$$G(a) = \sum_{s=0}^{\lfloor (p-3)/2 \rfloor} \frac{(1 - \frac{1}{2p})^{p-2s-1} (\frac{1}{2p})^{2s+1}}{m^s} \binom{p}{2s+1} T(I^{\otimes s+1} \otimes a^{\otimes p-2s-1})$$

$$= \sum_{s=0}^{\lfloor (p-3)/2 \rfloor} \frac{(1 - \frac{1}{2p})^{p-2s-1} (\frac{1}{2p})^{2s+1}}{m^s} \binom{p}{2s+1} \sum_{j=1}^k \lambda_j (v_j^\top a)^{p-2j-1} v_j$$

$$= \sum_{j=1}^k v_j \cdot \lambda_j \overbrace{\sum_{s=0}^{\lfloor (p-3)/2 \rfloor} \frac{(1 - \frac{1}{2p})^{p-2s-1} (\frac{1}{2p})^{2s+1}}{m^s} \binom{p}{2s+1} (v_j^\top a)^{p-2s-1}}^{S_j :=}$$

$$= \sum_{j=1}^k S_j v_j.$$

Notice for even $p$,

$$
S_j = \lambda_j (\boldsymbol{v}_j^\top \boldsymbol{a})^3 \cdot \sum_{s=0}^{p/2-2} \frac{(1 - \frac{1}{2p})^{p-2s-1}(\frac{1}{2p})^{2s+1}}{m^s} \binom{p}{2s+1} (\boldsymbol{v}_j^\top \boldsymbol{a})^{p-2s-4}
$$

$$
= \lambda_j (\boldsymbol{v}_j^\top \boldsymbol{a})^3 \cdot \sum_{r=0}^{p/2-2} \frac{(1 - \frac{1}{2p})^{2r+3}(\frac{1}{2p})^{p-3-2r}}{m^{p/2-2-r}} \binom{p}{p-2r-3} (\boldsymbol{v}_j^\top \boldsymbol{a})^{2r}.
$$

$$
\text{(let } 2r = p - 4 - 2s\text{)}
$$

$$
\frac{S_j}{\lambda_j (\boldsymbol{v}_j^\top \boldsymbol{a})^3} = \sum_{r=0}^{p/2-2} \frac{(1 - \frac{1}{2p})^{2r+3}(\frac{1}{2p})^{p-3-2r}}{m^{p/2-2-r}} \binom{p}{p-2r-3} (\boldsymbol{v}_j^\top \boldsymbol{a})^{2r}
$$

$$
\text{(Divide both sides by } \lambda_j (\boldsymbol{v}_j^\top \boldsymbol{a})^3\text{)}
$$

$$
\leq \sum_{r=0}^{p/2-2} \frac{(1 - \frac{1}{2p})^{2r+3}(\frac{1}{2p})^{p-3-2r}}{m^{p/2-2-r}} \binom{p}{p-2r-3} (\boldsymbol{v}_1^\top \boldsymbol{a})^{2r}.
$$

$$
\text{(Since the first term is constant and } |\boldsymbol{v}_j^\top \boldsymbol{a}| \leq \boldsymbol{v}_1^\top \boldsymbol{a} \text{ for } r \geq 1\text{)}
$$

$$
= \frac{S_1}{\lambda_1 (\boldsymbol{v}_1^\top \boldsymbol{a})^3}.
$$

Therefore for even $p \geq 4$:

$$
|S_j| \leq \frac{|\lambda_j|}{\lambda_1} \frac{|\boldsymbol{v}_j^\top \boldsymbol{a}|^3}{|\boldsymbol{v}_1^\top \boldsymbol{a}|^3} S_1 \leq \frac{1}{4} \frac{|\boldsymbol{v}_j^\top \boldsymbol{a}|}{|\boldsymbol{v}_1^\top \boldsymbol{a}|} S_1, \forall j \geq 2. \tag{3}
$$

Similarly for odd $p$, we have:

$$
S_j = \lambda_j (\boldsymbol{v}_j^\top \boldsymbol{a})^2 \cdot \sum_{s=0}^{(p-3)/2} \frac{(1 - \frac{1}{2p})^{p-2s-1}(\frac{1}{2p})^{2s+1}}{m^s} \binom{p}{2s+1} (v_j^\top a)^{p-2s-3}
$$

$$
= \lambda_j (\boldsymbol{v}_j^\top \boldsymbol{a})^2 \cdot \sum_{r=0}^{(p-3)/2} \frac{(1 - \frac{1}{2p})^{2r+2}(\frac{1}{2p})^{p-2-2r}}{m^{(p-3)/2-r}} \binom{p}{p-2-2r} (\boldsymbol{v}_j^\top \boldsymbol{a})^{2r},
$$

$$
\text{(Let } r = (p-3)/2 - s\text{)}
$$

$$
\frac{S_j}{\lambda_j (\boldsymbol{v}_j^\top \boldsymbol{a})^2} = \sum_{r=0}^{(p-3)/2} \frac{(1 - \frac{1}{2p})^{2r+2}(\frac{1}{2p})^{p-2-2r}}{m^{(p-3)/2-r}} \binom{p}{p-2-2r} (\boldsymbol{v}_j^\top \boldsymbol{a})^{2r}
$$

$$
\text{(Divide both sides by } \lambda_j (\boldsymbol{v}_j^\top \boldsymbol{a})^2\text{)}
$$

$$
\leq \sum_{r=0}^{(p-3)/2} \frac{(1 - \frac{1}{2p})^{2r+2}(\frac{1}{2p})^{p-2-2r}}{m^{(p-3)/2-r}} \binom{p}{p-2-2r} (\boldsymbol{v}_1^\top \boldsymbol{a})^{2r}
$$

$$
\text{(Since the first term is constant and } |\boldsymbol{v}_j^\top \boldsymbol{a}| \leq \boldsymbol{v}_1^\top \boldsymbol{a} \text{ for } r \geq 1\text{)}
$$

$$
= \frac{S_1}{\lambda_1 (\boldsymbol{v}_1^\top \boldsymbol{a})^2}.
$$

Therefore for odd $p$ we have:

$$
|S_j| \leq \frac{|\lambda_j|}{\lambda_1} \frac{|\boldsymbol{v}_j^\top \boldsymbol{a}|^2}{|\boldsymbol{v}_1^\top \boldsymbol{a}|^2} S_1 \leq \frac{1}{2} \frac{|\boldsymbol{v}_j^\top \boldsymbol{a}|}{|\boldsymbol{v}_1^\top \boldsymbol{a}|} S_1, \forall j \geq 2. \tag{4}
$$

Write $\boldsymbol{V} = [\boldsymbol{v}_2, \boldsymbol{v}_3, \cdots, \boldsymbol{v}_k] \in \mathbb{R}^{d \times k}$ be the complement for $\boldsymbol{v}_1$. Therefore for any $\boldsymbol{x}$ without normalization, one can conveniently represent $|\tan \theta(\boldsymbol{x}, \boldsymbol{v}_1)|$ as $\|\boldsymbol{V}^\top \boldsymbol{x}\|_2 / |\boldsymbol{v}_1^\top \boldsymbol{x}|$.

$$\|\boldsymbol{V}^\top G(\boldsymbol{a})\|^2 = \sum_{j=2}^{k} S_j^2 \tag{5}$$

$$\leq \sum_{j=2}^{k} \frac{|\boldsymbol{v}_j^\top \boldsymbol{a}|^2}{4|\boldsymbol{v}_1^\top \boldsymbol{a}|^2} S_1^2 \tag{from (4),(3)}$$

$$= \frac{1}{4} \tan^2 \theta(\boldsymbol{v}_1, \boldsymbol{a})(\boldsymbol{v}_1^\top G(\boldsymbol{a}))^2. \tag{6}$$

Therefore for $p \geq 3$, $\tan \theta(G(\boldsymbol{a}), \boldsymbol{v}_1) \leq \frac{1}{2} \tan \theta(\boldsymbol{a}, \boldsymbol{v}_1)$. $\qquad\square$

### D.1.1 Proof Sketch of Theorem 3.14

**Definition D.2** (Zeroth order gradient function). For some scalar $m$, we define an empirical operator $G_n : \mathcal{A} \to \mathcal{A}$ that is similar to the zeroth-order gradient of $f$ through $n$ samples:

$$G_n(\boldsymbol{a}) := \frac{m}{n} \sum_{i=1}^{n} \left( T\left( \left( (1 - \frac{1}{2p})\boldsymbol{a} + \frac{1}{2p}\boldsymbol{z}_i \right)^{\otimes p} \right) - T(\frac{1}{2p}\boldsymbol{z}_i) \right) \boldsymbol{z}_i + (\eta_i - \eta_i')\boldsymbol{z}_i.$$

where $\boldsymbol{z}_i \sim \mathcal{N}(0, \frac{1}{m}\boldsymbol{I})$ and $\eta_i, \eta_i'$ are independent zero-mean 1-sub-Gaussian noise. Therefore we have:

$$\mathbb{E}[G_n(\boldsymbol{a})] = m\mathbb{E}[\sum_{l=0}^{p-1} \binom{p}{l} T((1 - \frac{1}{2p})^{p-l} \boldsymbol{a}^{\otimes(p-l)} \otimes (\frac{1}{2p})^l \boldsymbol{z}^{\otimes l}) \boldsymbol{z}]$$

(Due to symmetry of Gaussian only for odd $l =: 2s + 1$ expectation is nonzero)

$$= (1 - \frac{1}{2p})^{p-2s-1}(\frac{1}{2p})^{2s+1}[ \sum_{s=0}^{\lfloor p/2-1 \rfloor} m^{-s} \binom{p}{2s+1} T(\boldsymbol{a}^{\otimes(p-2s-1)} \otimes \boldsymbol{I}^{\otimes s+1})]$$

Note that for even $p$ the last term (when $s = p/2 - 1$) is $T(\boldsymbol{a} \otimes \boldsymbol{I}^{\otimes p/2}) = \sum_{j=1}^{k} \lambda_j(\boldsymbol{a}^\top \boldsymbol{v}_j)\boldsymbol{v}_j$. While all other terms will push the iterate towards the optimal action at a superlinear speed, the last term perform a matrix multiplication and the convergence speed will depend on the eigengap. Therefore for $p \geq 4$ we will remove the extra bias in the last term that is orthogonal to $\boldsymbol{v}_1$ and will treat it as noise. (Notice for quadratic function $s = 0 = p/2 - 1$ is the only term in $\mathbb{E}[G_n(\boldsymbol{a})]$. This is the distinction between $p = 2$ and larger $p$, and why its convergence depends on eigengap.)

We further define $G(\boldsymbol{a})$ as the population version of $G_n(\boldsymbol{a})$ by removing this undesirable bias term that will be treated as noise:

$$G(\boldsymbol{a}) = \begin{cases} \mathbb{E}[G_n] - \frac{(\frac{1}{2p})^{p-1}(1-\frac{1}{2p})p}{m^{p/2-1}} \sum_{j=2}^{k} \lambda_j(\boldsymbol{v}_j^\top \boldsymbol{a})\boldsymbol{v}_j, & \text{when } p \text{ is even} \\ \mathbb{E}[G_n], & \text{when } p \text{ is odd.} \end{cases}$$

$$= \sum_{s=0}^{\lfloor (p-3)/2 \rfloor} \frac{(\frac{1}{2p})^{2s+1}}{m^s} \binom{p}{2s+1} T(\boldsymbol{I}^{\otimes s+1} \otimes ((1 - \frac{1}{2p})\boldsymbol{a})^{\otimes p-2s-1})$$

$$= \frac{1}{2}(1 - \frac{1}{2p})^{p-1} T(\boldsymbol{I}, \boldsymbol{a}^{\otimes p-1}) + O(1/m).$$

We define $G(\boldsymbol{a})$ to push the action $\boldsymbol{a}$ towards the $\boldsymbol{v}_1$ direction with at least linear convergence rate. More precisely, their angle $\tan \theta(G(\boldsymbol{a}), \boldsymbol{v}_1)$ will converge linearly to 0 for proper initialization with the dynamics $\boldsymbol{a} \to G(\boldsymbol{a})$. An easy way to see that is when $p = 2$ or 3, $G$ is conducting (3-order tensor) power iteration. For higher-order problems, this operation $G$ is equivalent to the summation of $p, p - 2, p - 4, \cdots$-th order tensor product and hence the linear convergence.

The estimation error $G_n(\boldsymbol{a}) - G(\boldsymbol{a})$ will be treated as noise (which is not mean zero when $p$ is even but will be small enough: $O((2p)^{-p}m^{-(p-1)/2})$). Therefore the iterative algorithm with $\boldsymbol{a} \to G_n(\boldsymbol{a})$ will converge to a small neighborhood of $\boldsymbol{v}_1$ depending on the estimation error. This estimation error is controlled by the choice of sample size $n$ in each iteration. We now provide the proof sketch:

**Lemma D.3** (Initialization for $p \geq 3$; Corollary C.1 from [70] ). *For any $\eta \in (0, 1/2)$, with $L = \Theta(k \log(1/\eta))$ samples $\mathcal{A} = \{\boldsymbol{a}^{(1)}, \boldsymbol{a}^{(2)}, \cdots \boldsymbol{a}^{(L)}\}$ where each $\boldsymbol{a}^{(l)}$ is sampled uniformly on the sphere $\mathbb{S}^{d-1}$. At least one sample $\boldsymbol{a} \in \mathcal{A}$ satisfies*

$$\max_{j \neq 1} |\boldsymbol{v}_j^\top \boldsymbol{a}| \leq 0.5 |\boldsymbol{v}_1^\top \boldsymbol{a}|, \text{ and } |\boldsymbol{v}_1^\top \boldsymbol{a}| \geq 1/\sqrt{d}. \tag{7}$$

*with probability at least $1 - \eta$.*

**Lemma D.4** (Iterative progress). *Let $\alpha = 1/2$ for $p \geq 3$ in Algorithm 6. Consider noisy operation $\boldsymbol{a}^+ \to G(\boldsymbol{a}) + \boldsymbol{g}$. If the error term $\boldsymbol{g}$ satisfies:*

$$\|\boldsymbol{g}\| \leq \min\{\frac{0.025}{p} \lambda_1 (\boldsymbol{v}_1^\top \boldsymbol{a})^{p-2}, 0.1\lambda_1 \widetilde{\varepsilon}\}$$
$$+ 0.03\lambda_1 |\sin \theta(\boldsymbol{v}_1, \boldsymbol{a})| (\boldsymbol{v}_1^\top \boldsymbol{a})^{p-2},$$
$$|\boldsymbol{v}_1^\top \boldsymbol{g}| \leq 0.05\lambda_1 (\boldsymbol{v}_1^\top \boldsymbol{a})^{p-1}.$$

*Suppose $\boldsymbol{a}$ satisfies $0.5|\boldsymbol{v}_1^\top \boldsymbol{a}| \geq \max_{j \geq 2} |\boldsymbol{v}_j^\top \boldsymbol{a}|$, we have:*

$$\tan \theta(\boldsymbol{a}^+, \boldsymbol{v}_1) \leq 0.8 \tan \theta(\boldsymbol{a}, \boldsymbol{v}_1) + \widetilde{\varepsilon}.$$

We can also bound $\boldsymbol{g}$ by standard concentration plus an additional small bias term.

**Lemma D.5** (Estimation error bound for $G$). *For fixed value $\delta \in (0, 1)$ and large enough universal constant $c_1, c_2, c_m, c_n$, when $m = c_m d \log(n/\delta), n \geq c_n d \log(d/\delta)$, we have*

$$\|\boldsymbol{g}\| \equiv \|G_n(\boldsymbol{a}) - G(\boldsymbol{a})\| \leq c_1 \sqrt{\frac{d^2 \log^3(n/\delta) \log(d/\delta)}{n}} + e\lambda_2 |\sin \theta(\boldsymbol{a}, \boldsymbol{v}_1)|,$$

$$|\boldsymbol{v}_1^\top \boldsymbol{g}| \equiv |\boldsymbol{v}_1^\top G_n(\boldsymbol{a}) - \boldsymbol{v}_1^\top G(\boldsymbol{a})| \leq c_2 \sqrt{\frac{d \log^3(n/\delta) \log(d/\delta)}{n}}.$$

*with probability $1 - \delta$. $e = 0$ for odd $p$ and $e = (2p)^{-(p-1)} m^{-(p/2-1)}$ for even $p$.*

Together we are able to prove Theorem 3.14:

*Proof of Theorem 3.14.* Initially with high probability there exists an $\boldsymbol{a}_0 \in \mathcal{A}_0$ such that Eqn. (7) holds, i.e., $\boldsymbol{v}_1^\top \boldsymbol{a}_0 \geq 1/\sqrt{d}$ and $\boldsymbol{v}_1^\top \boldsymbol{a}_0 \geq 2|\boldsymbol{v}_j^\top \boldsymbol{a}_0|, \forall j \geq 2$.

Next, from Lemma D.5, the extra bias term is bounded by $e\lambda_2 |\sin \theta(\boldsymbol{a}, \boldsymbol{v}_1)|$
$\leq 0.03\lambda_1 (\boldsymbol{v}_1^\top \boldsymbol{a})^{p-2} |\sin \theta(\boldsymbol{a}, \boldsymbol{v}_1)|$ since $e = (2p)^{-p+1} m_s^{-p/2+1}$ and with our choice of variance $m_s \geq d \geq (\boldsymbol{v}_1^\top \boldsymbol{a})^{-2}$, plus $p \geq 3$. Next with our setting of $n_s = \widetilde{\Theta}(d^p/(\lambda_1^2 \widetilde{\varepsilon}_t^2))$, the error term $\|\mathbb{E}[G(\boldsymbol{a})] - G_n(\boldsymbol{a})\|$ is upper bounded by $\widetilde{O}(\sqrt{\frac{d^2}{n}}) \leq 0.025\lambda_1 d^{-(p-2)/2} \widetilde{\varepsilon}_s/p + 0.1\lambda_1 \widetilde{\varepsilon}_s$. Meanwhile $|\boldsymbol{v}_1^\top \boldsymbol{g}| \leq \widetilde{O}(\sqrt{\frac{d}{n_s}}) \leq 0.05\lambda_1 (\boldsymbol{v}_1^\top \boldsymbol{a})^{p-1}$.

This meets the requirements for Theorem D.4 and therefore $\tan \theta(G_n(\boldsymbol{a}_0), \boldsymbol{v}_1) \leq 0.8 \tan \theta(\boldsymbol{a}_0, \boldsymbol{v}_1) + 0.1\lambda_1 \widetilde{\varepsilon}_s$. Therefore after $l$ steps will have

$$\tan \theta(G_n^l(\boldsymbol{a}_0), \boldsymbol{v}_1) \leq 0.8^l \tan \theta(\boldsymbol{a}_0, \boldsymbol{v}_1) + \sum_{i=1}^{l} 0.8^i \cdot 0.1\widetilde{\varepsilon}_s$$
$$\leq 0.8^l \tan \theta(\boldsymbol{a}_0, \boldsymbol{v}_1) + 0.5\widetilde{\varepsilon}_s.$$

Notice initially $\tan \theta(\boldsymbol{a}_0, \boldsymbol{v}_1) \leq 1/(\boldsymbol{v}_1^\top \boldsymbol{a}_0) \leq \sqrt{d}$. Therefore after at most $l = O(\log_2(\tan \theta(\boldsymbol{a}_0, \boldsymbol{v}_1))) \leq O(\log_2(d))$ steps, we will have $\tan(G_n^l(\boldsymbol{a}_0), \boldsymbol{v}_1) \leq \widetilde{\varepsilon}_0/2 = \widetilde{\varepsilon}_1$. With the same argument, the progress also holds for $s > 0$ with even smaller $l$. $\square$

*Proof of Lemma D.5.* We first estimate $G_n(\boldsymbol{a}) - \mathbb{E}[G_n(\boldsymbol{a})]$, which is want we want for even $p$. For odd $p$ we will need to analyze an extra bias term that is orthogonal to $\boldsymbol{v}_1$, $\boldsymbol{e} := \frac{(\frac{1}{2p})^{p-1}(1-\frac{1}{2p})p}{m^{p/2-1}} \sum_{j=2}^{k} \lambda_j (\boldsymbol{v}_j^\top \boldsymbol{a}) \boldsymbol{v}_j$; and we have $G_n(\boldsymbol{a}) - \mathbb{E}[G_n(\boldsymbol{a})] = G_n(\boldsymbol{a}) - G(\boldsymbol{a}) + \boldsymbol{e}$.

We decompose $G_n(\boldsymbol{a})$ as $G_n(\boldsymbol{a}) = \sum_{s=1}^{k} G_n^{(s)} + N$, where $G_n^{(s)} := \frac{m}{n} \sum_{i=1}^{n} \binom{p}{s} T(((1 - 0.5/p)\boldsymbol{a})^{\otimes p-s} \otimes (\boldsymbol{z}_i/(2p))^{\otimes s})\boldsymbol{z}_i$. The noise term $N := \frac{m}{n} \sum \epsilon_i \boldsymbol{z}_i$.

$$
\begin{aligned}
G_n^{(s)} :=& \frac{m}{n} \sum_{i=1}^{n} \binom{p}{s} T(((1 - 0.5/p)\boldsymbol{a})^{\otimes p-s} \otimes (\boldsymbol{z}_i/(2p))^{\otimes s})\boldsymbol{z}_i \\
=& \frac{m}{n}(1 - \frac{1}{2p})^{p-s}(\frac{1}{2p})^s \binom{p}{s} \sum_{i=1}^{n} \sum_{j=1}^{k} \lambda_j (\boldsymbol{a}^\top \boldsymbol{v}_j)^{p-s}(\boldsymbol{z}_i^\top \boldsymbol{v}_j)^s \boldsymbol{z}_i.
\end{aligned}
$$

$$
\begin{aligned}
\mathbb{E}[G_n^{(s)}] =& m(1 - \frac{1}{2p})^{p-s}(\frac{1}{2p})^s \binom{p}{s} \sum_{j=1}^{k} \lambda_j (\boldsymbol{a}^\top \boldsymbol{v}_j)^{p-s} \mathbb{E}[(\boldsymbol{z}^\top \boldsymbol{v}_j)^s \boldsymbol{z}] \\
=& \begin{cases} (1 - \frac{1}{2p})^{p-s}(\frac{1}{2p})^s m \binom{p}{s} \sum_{j=1}^{k} \lambda_j (\boldsymbol{a}^\top \boldsymbol{v}_j)^{p-s} \frac{1}{m^{(s+1)/2}}(s)!! \boldsymbol{v}_j, & \text{for odd } s, \\ 0, & \text{for even } s \end{cases} \\
=& \begin{cases} (1 - \frac{1}{2p})^{p-s}(\frac{1}{2p})^s \frac{s!!}{m^{(s-1)/2}} \binom{p}{s} \sum_{j=1}^{k} \lambda_j (\boldsymbol{a}^\top \boldsymbol{v}_j)^{p-s} \boldsymbol{v}_j, & \text{for odd } s, \\ 0, & \text{for even } s \end{cases}
\end{aligned}
$$

$$
G_n^{(s)} - \mathbb{E}[G_n^{(s)}] = m(1 - \frac{1}{2p})^{p-s}(\frac{1}{2p})^s \binom{p}{s} \sum_{j=1}^{k} \lambda_j (\boldsymbol{a}^\top \boldsymbol{v}_j)^{p-s} \boldsymbol{g}_{n,s}(j),
$$

where $\boldsymbol{g}_{n,s}(j) := \frac{1}{n} \sum_{i=1}^{n} (\boldsymbol{z}_i^\top \boldsymbol{v}_j)^s \boldsymbol{z}_i - \mathbb{E}[(\boldsymbol{z}^\top \boldsymbol{v}_j)^s \boldsymbol{z}]$.

Notice the scaling in each $G_n^{(s)}$ is $(1 - \frac{1}{2p})^{p-s}(\frac{1}{2p})^s \binom{p}{s} \leq (\frac{1}{2p})^s p^s/(s!) < 2^{-s}$ decays exponentially. In Claim D.12 we give bounds for $g_{n,s}(j)$. We note the bound for each $g_{n,s}$ also decays with $s$.

Therefore the bottleneck of the upper bound mostly depend on $\boldsymbol{g}_{n,0}$ and $\boldsymbol{v}_1^\top \boldsymbol{g}_{n,0}$, and we get:

$$
\begin{aligned}
\|G_n - \mathbb{E}[G_n]\| \leq& C_1 \lambda_1 \sqrt{\frac{(d + \log(1/\delta))d \log(n/\delta)}{n}} + N, \\
|\boldsymbol{v}_1^\top G_n - \mathbb{E}[\boldsymbol{v}_1^\top G_n]| \leq& C_2 \lambda_1 \sqrt{\frac{d \log(n/\delta)(1 + \log(1/\delta))}{n}} + \boldsymbol{v}_1^\top N.
\end{aligned}
$$

Next from Claim D.11, the noise term

$$
N \leq C_3 \sqrt{\frac{m \log(n/\delta)(d + \log(n/\delta)) \log(d/\delta)}{n}},
$$

$$
|v_1^\top N| \leq C_4 \sqrt{m \frac{\log^2(n/\delta) \log(d/\delta)}{n}},
$$

Finally $\boldsymbol{e}$ is very small: $\|\boldsymbol{e}\| \leq \frac{1}{m^{p/2-1}(2p)^{(p-1)}} \lambda_2 \|\boldsymbol{V}^\top \boldsymbol{a}\| = \lambda_2 \frac{1}{m^{p/2-1}(2p)^{(p-1)}} \sin\theta(\boldsymbol{a}, \boldsymbol{v}_1)$. $|\boldsymbol{e}^\top \boldsymbol{v}_1| = 0$.

Together we can bound $G_n(\boldsymbol{a}) - G(\boldsymbol{a})$ and finish the proof. $\qquad\square$

*Proof of Lemma D.4.* From Lemma D.1 we have: $|\tan\theta(G(\boldsymbol{a}),\boldsymbol{v}_1)| \leq 1/2|\tan\theta(\boldsymbol{a},\boldsymbol{v}_1)|$. Let $\boldsymbol{V} = [\boldsymbol{v}_2,\cdots\boldsymbol{v}_k]$. For any $p \geq 2$, we have:

$$\begin{aligned}
|\tan\theta(\boldsymbol{a}^+,\boldsymbol{v}_1)| &= \frac{\|\boldsymbol{V}^\top\boldsymbol{a}^+\|_2}{|\boldsymbol{v}_1^\top\boldsymbol{a}^+|}\\
&= \frac{\|\boldsymbol{V}^\top(G(\boldsymbol{a})+\boldsymbol{g})\|}{|\boldsymbol{v}_1^\top(G(\boldsymbol{a})+\boldsymbol{g})|}\\
&\leq \frac{\|\boldsymbol{V}^\top G(\boldsymbol{a})\| + \|\boldsymbol{V}^\top\boldsymbol{g}\|}{|\boldsymbol{v}_1^\top G(\boldsymbol{a})| - |\boldsymbol{v}_1^\top\boldsymbol{g}|}\\
&\leq \frac{1/2|\tan\theta(\boldsymbol{a},\boldsymbol{v}_1)||\boldsymbol{v}_1^\top G(\boldsymbol{a})| + \|\boldsymbol{g}\|}{|\boldsymbol{v}_1^\top G(\boldsymbol{a})| - |\boldsymbol{v}_1^\top\boldsymbol{g}|}\\
&= \alpha|\tan\theta(\boldsymbol{a},\boldsymbol{v}_1)|\frac{S_1}{S_1 - \|\boldsymbol{v}_1^\top\boldsymbol{g}\|} + \frac{\|\boldsymbol{g}\|}{S_1 - \|\boldsymbol{v}_1^\top\boldsymbol{g}\|},
\end{aligned}$$

where $S_1 := \boldsymbol{v}_1^\top G(\boldsymbol{a}) = \boldsymbol{v}_1^\top G(\boldsymbol{a}) = \lambda_1\sum_{s=0}^{\lfloor(p-3)/2\rfloor}\frac{(1-\frac{1}{2p})^{p-2s-1}(\frac{1}{2p})^{2s+1}}{m^s}\binom{p}{2s+1}(\boldsymbol{v}_1^\top\boldsymbol{a})^{p-2s-1} \geq \lambda_1(1-\frac{1}{2p})^{p-1}(\frac{1}{2p})p(\boldsymbol{v}_1^\top\boldsymbol{a})^{p-1} \geq \frac{\lambda_1}{4}(\boldsymbol{v}_1^\top\boldsymbol{a})^{p-1}$. The inequality comes from keeping only the first term where $s = 0$. With the assumption that $|\boldsymbol{v}_1^\top\boldsymbol{g}| \leq 0.05\lambda_1(\boldsymbol{v}_1^\top\boldsymbol{a})^{p-1}$, we have $|\boldsymbol{v}_1^\top\boldsymbol{g}| \leq 0.2S_1$. Therefore

$$\begin{aligned}
|\tan\theta(\boldsymbol{a}^+,\boldsymbol{v}_1)| &\leq 1.25/2|\tan\theta(\boldsymbol{a},\boldsymbol{v}_1)| + \frac{\|\boldsymbol{g}\|}{S_1 - |\boldsymbol{v}_1^\top\boldsymbol{g}|}\\
&\leq 1.25/2|\tan\theta(\boldsymbol{a},\boldsymbol{v}_1)| + 5/4\frac{\|\boldsymbol{g}\|}{S_1}\\
&\leq 1.25/2|\tan\theta(\boldsymbol{a},\boldsymbol{v}_1)| + 5\frac{\|\boldsymbol{g}\|}{\lambda_1(\boldsymbol{v}_1^\top\boldsymbol{a})^{p-1}}.
\end{aligned}$$

Notice when $5\frac{\|\boldsymbol{g}\|}{\lambda_1(\boldsymbol{v}_1^\top\boldsymbol{a})^{p-1}} \leq \max\{0.125|\tan\theta(\boldsymbol{a},\boldsymbol{v}_1)|,\widetilde{\epsilon}\}$, which will ensure $|\tan\theta(G(\boldsymbol{a}),\boldsymbol{v}_1)| \leq (1.25/2 + 0.125)|\tan\theta(G(\boldsymbol{a}),\boldsymbol{v}_1)| + \widetilde{\epsilon}$. (We will prove this condition is satisfied when $\|\boldsymbol{g}\| \leq \min\{\frac{0.025}{p}\lambda_1(\boldsymbol{v}_1^\top\boldsymbol{a})^{p-2}, 0.1\lambda_1\widetilde{\varepsilon}\}$. We will handle the additional term in the upper bound of $\|\boldsymbol{g}\|$ later.) We divide this requirement into the following two cases. On one hand, when $|\boldsymbol{v}_1^\top\boldsymbol{a}| \leq 1 - 1/(p-1)$, $\|\boldsymbol{V}^\top\boldsymbol{a}\| \geq \sqrt{1-(1-1/(p-1))^2} > 1/p$, therefore $|\tan\theta(\boldsymbol{a},\boldsymbol{v}_1)| \geq 1/p|\boldsymbol{v}_1^\top\boldsymbol{a}|$. Therefore

$$\begin{aligned}
&5\frac{\|\boldsymbol{g}\|}{\lambda_1(\boldsymbol{v}_1^\top\boldsymbol{a})^{p-1}} \leq 0.125|\tan\theta(\boldsymbol{a},\boldsymbol{v}_1)|\\
\Longleftarrow &5\frac{\|\boldsymbol{g}\|}{\lambda_1(\boldsymbol{v}_1^\top\boldsymbol{a})^{p-1}} \leq 0.125/\left(p|\boldsymbol{v}_1^\top\boldsymbol{a}|\right)\\
\Longleftrightarrow &\|\boldsymbol{g}\| \leq 0.025\lambda_1(\boldsymbol{v}_1^\top\boldsymbol{a})^{p-2}/p.
\end{aligned}$$

On the other hand, when $|\boldsymbol{v}_1^\top\boldsymbol{a}| \geq 1 - 1/(p-1)$, $|\boldsymbol{v}_1^a|^{p-1} \geq 1/4$ when $p = 3$. Therefore $5\frac{\|\boldsymbol{g}\|}{\lambda_1(\boldsymbol{v}_1^\top\boldsymbol{a})^{p-1}} \leq 20\|\boldsymbol{g}\|/\lambda_1$. Therefore we will need $\|\boldsymbol{g}\| \leq 0.05\lambda_1\widetilde{\epsilon}$, and then the requirement that $5\frac{\|\boldsymbol{g}\|}{\lambda_1(\boldsymbol{v}_1^\top\boldsymbol{a})^{p-1}} \leq \widetilde{\epsilon}$ is satisfied.

Altogether in both cases we have: $|\tan\theta(G(\boldsymbol{a}),\boldsymbol{v}_1)| \leq 0.75|\tan\theta(G(\boldsymbol{a}),\boldsymbol{v}_1)| + \widetilde{\epsilon}$. Finally if we additionally increase $\|\boldsymbol{g}\|$ by $0.05\lambda_1(\boldsymbol{v}_1^\top\boldsymbol{a})^{p-1}$ we will have: $|\tan\theta(G(\boldsymbol{a}),\boldsymbol{v}_1)| \leq 0.8|\tan\theta(G(\boldsymbol{a}),\boldsymbol{v}_1)| + \widetilde{\epsilon}$. □

*Proof of Corollary 3.16.* As shown in Theorem 3.14 at least one action $\boldsymbol{a}$ in $\mathcal{A}_S, |\mathcal{A}_S| \leq \widetilde{O}(k)$ satisfies $\tan\theta(\boldsymbol{a},\boldsymbol{a}^*) \leq \varepsilon$ with a total of $\widetilde{O}(\frac{d^p k}{\lambda_1^2\varepsilon^2})$ steps. Therefore with Claim D.6 we have to get $\widetilde{\varepsilon}$-optimal reward we need $\widetilde{O}(\frac{d^p k}{\lambda_1\widetilde{\varepsilon}})$ steps. Notice the eluder dimension for symmetric polynomials is $d^p$ and the size of $\mathcal{A}_S$ is at most $\widetilde{O}(k)$. Then by applying Corollary D.9 we get that the total regret is at most $\widetilde{O}(\sqrt{d^p kT} + \sqrt{|\mathcal{A}_S|T}) = \widetilde{O}(\sqrt{d^p kT})$. □

## D.2 PAC to Regret Bound Relation.

**Claim D.6** (Connecting angle to regret). *When $0 < \tan\theta(\boldsymbol{a}, \boldsymbol{v}_1) \leq \zeta$, we have regret $r^* - r(\boldsymbol{a}) \leq r^* \min\{2, p\zeta^2\}$.*

*Proof.*

$$
\begin{aligned}
|\cos\theta(\boldsymbol{a}, \boldsymbol{v}_1)| =& |\boldsymbol{a}^\top \boldsymbol{v}_1| =: b, \\
|\tan\theta(\boldsymbol{a}, \boldsymbol{v}_1)| =& \frac{\sqrt{1-b^2}}{b} \leq \zeta \Leftrightarrow b \geq \frac{1}{\sqrt{\zeta^2+1}}. \\
\Rightarrow r^* - r(\boldsymbol{a}) \leq& \lambda_1 - \lambda_1 b^p \\
\leq& \lambda_1 (1 - (\zeta^2+1)^{-p/2}) \\
=& \lambda_1 \frac{(\zeta^2+1)^{p/2} - 1}{(\zeta^2+1)^{p/2}} \\
\leq& \lambda_1 ((\zeta^2+1)^{p/2} - 1) \qquad\qquad \text{(since denominator } (\zeta^2+1)^{p/2} \geq 1) \\
\leq& \lambda_1 p\zeta^2, \text{ when } \zeta^2 \leq 1/p.
\end{aligned}
$$

Additionally by definition $r^* - r(\boldsymbol{a}) \leq \lambda_1 - (-\lambda_1) = 2\lambda_1$ and thus $r^* - r(a) \leq \lambda_1 \min\{2, p\zeta^2\}$. We now derive the last inequality. When $\zeta \geq 1/p$ it is trivially true. When $\zeta \leq 1/p$, we have $(1 + \zeta^2)^{p/2} \leq 1 + p\zeta^2$ for any $p \geq 2$. Since the LHS is a convex function for $\zeta$ when $p \geq 2$ and when $\zeta = 0$ LHS=RHS and when $\zeta^2 = 1/p$ LHS is always smaller than RHS (=2).

Notice the argument is straightforward to extend to the setting where the angle is between $\boldsymbol{a}$ and subspace $V_1$ that satisfies $\forall \boldsymbol{v} \in \boldsymbol{V}_1, T(\boldsymbol{v}) \geq \lambda_1 - \epsilon$, then one also get $r^* - r(\boldsymbol{a}) \leq \lambda_1 - (\lambda_1 - \epsilon)b^p \leq \min\{\lambda_1, \lambda_1 p\zeta^2 + \epsilon b^p\} \leq \min\{\lambda_1, \lambda_1 p\zeta^2 + \epsilon\}$. $\square$

**Claim D.7** (Connecting PAC to Cumulative Regret). *Suppose we have an algorithm $\text{alg}(\zeta)$ that finds $\zeta$-optimal action $\widehat{\boldsymbol{a}}$ that satisfies $0 < \tan\theta(\boldsymbol{a}, \boldsymbol{v}_1) \leq \zeta$ by taking $A\zeta^{-a}$ actions. Here $A$ can depend on any parameters such as $d, \lambda_1$, probability error $\delta$, etc., that are not $\zeta$. Then for large enough $T$, by calling alg with $\zeta = A^{\frac{1}{a+2}} T^{-\frac{1}{a+2}} p^{-\frac{1}{a+2}}$ and playing its output action $\widehat{\boldsymbol{a}}$ for the remaining actions, one can get a cumulative regret of:*

$$
\mathfrak{R}(T) \lesssim T^{\frac{a}{a+2}} p^{\frac{a}{a+2}} A^{\frac{2}{a+2}} r^*.
$$

*Similarly, if an oracle finds $\varepsilon$-optimal action $\widehat{\boldsymbol{a}}$ that satisfies $r^* - r(\boldsymbol{a}) \leq \varepsilon$ with $B\varepsilon^{-b}$ samples, then by setting $\varepsilon = (Br^*/T)^{\frac{1}{1+b}}$, and playing the output arm for the remaining actions, one can get cumulative regret of:*

$$
\mathfrak{R}(T) \lesssim B^{\frac{1}{1+b}} T^{\frac{b}{1+b}} r^{\frac{1}{1+b}}.
$$

*Proof.* For the chosen $\zeta$, write $T_1 = A\zeta^{-a}$ be the number of actions that finds $\zeta$-optimal action. Therefore $T_1 = A^{\frac{2}{a+2}} T^{\frac{a}{a+2}} p^{\frac{a}{a+2}}$. First, when $T \geq Ap^{a/2}$, $\zeta^2 \leq 1/p$, namely $r^* - r(\boldsymbol{a}) \leq r^* p\zeta^2$. We have:

$$
\begin{aligned}
\mathfrak{R}(T) \leq& \sum_{t=1}^{T_1} 2r^* + \sum_{t=T_1+1}^{T} r^* p\zeta^2 \\
\leq& 2r^* T_1 + Tr^* p\zeta^2 \\
\leq& 3T^{\frac{a}{a+2}} p^{\frac{a}{a+2}} A^{\frac{2}{a+2}} r^*.
\end{aligned}
$$

When $T < Ap^{a/2}$, it trivially holds that $\mathfrak{R}(T) \leq 2r^* T < 2T^{\frac{a}{a+2}} p^{\frac{a}{a+2}} A^{\frac{2}{a+2}} r^*$. $\square$

**Theorem D.8** (Theorem 5.1 from [5]). *With UCB algorithm on action set with size $K$, we have with probability $1 - \delta$,*

$$
\mathfrak{R}(T) = \widetilde{O}(\min\{\sqrt{KT}\} + K).
$$

---
**Algorithm 7** UCB (Algorithm 1 in Section 5 of [5])
---
1: **Input:** Stochastic reward function $f$, failure probability $\delta$, action set $\mathcal{A}$ with finite size $K$.
2: **for** $t$ from 1 to $T - 1 - K$ **do**
3:     Execute arm $I_t = \arg\max_{i \in [K]} \left( \widehat{\mu}^t(i) + \sqrt{\frac{\log(TK/\delta)}{N^t(i)}} \right)$. Here $N^t(\boldsymbol{a}) = 1 + \sum_{i=1}^t \mathbf{1}\{I_i = \boldsymbol{a}\}$; and $\widehat{\mu}^t(\boldsymbol{a}) = \frac{1}{N^t(\boldsymbol{a})} \left( r_a + \sum_{i=1}^t \mathbf{1}\{I_i = \boldsymbol{a}\} r_i \right)$.
4:     Observe $r_{I_t}$
---

**Corollary D.9.** *With the same setting of Claim D.7, except that now the algorithm alg($\varepsilon$) finds **a set** $\mathcal{A}$ of size $S$ where at least one action $\boldsymbol{a} \in \mathcal{A}$ satisfies $r^* - f(\boldsymbol{a}) \leq \varepsilon$. Then all argument in Claim D.7 still hold by adding $\widetilde{O}(\sqrt{ST})$ on the RHS of each regret bound.*

*Proof.* Suppose we run *alg* for $T_1$ steps and achieve $\varepsilon$-optimal reward.

Let $r_\varepsilon := \max_{\boldsymbol{a} \in \mathcal{A}} f(\boldsymbol{a})$. Therefore with UCB on mutiarm bandit we have: $\sum_{t=T_1+1}^T r_\varepsilon - f(\boldsymbol{a}_t) \leq \widetilde{O}(\sqrt{ST})$ by Theorem D.8.

From the statement $r_\varepsilon \geq r^* - \zeta$. Therefore $\sum_{t=T_1+1}^T r^* - f(\boldsymbol{a}_t) \leq \widetilde{O}(\sqrt{ST}) + \varepsilon(T - T_1)$. Therefore

$$\mathfrak{R}(T) \leq \sum_{t=1}^{T_1} 2r^* + \varepsilon(T - T_1) + \widetilde{O}(\sqrt{ST}).$$

With the same choices of $T_1$ in Claim D.7, the same conclusion still holds with an additional term of $\widetilde{O}(\sqrt{ST})$. $\qquad\square$

For symmetric tensor problems the set size is $\widetilde{O}(k)$ and therefore we will have an additional $\sqrt{kT}$ term which will be subsumed in our regret bound.

### D.3 Variance and Noise Concentration

**Lemma D.10** (Vector Bernstein; adapted from Theorem 7.3.1 in [65]). *Consider a finite sequence $\{\boldsymbol{x}_k\}_{k=1}^n$ be i.i.d randomly generated samples, $x_k \in \mathbb{R}^d$, and assume that $\mathbb{E}[\boldsymbol{x}_k] = 0$, $\|\boldsymbol{x}_k\| \leq L$, and covariance matrix of $x_k$ is $\Sigma$. Then it satisfies that when $n \geq \log d/\delta$, we have:*

$$\left\| \frac{\sum_{i=1}^n \boldsymbol{x}_i}{n} \right\| \leq C \sqrt{\frac{(\|\Sigma\| + L^2)\log d/\delta}{n}},$$

*with probability $1 - \delta$.*

**Claim D.11** (Noise concentration). *Let independent samples $\boldsymbol{z}_i \sim \mathcal{N}(0, 1/m I_d)$ and $\epsilon_i \sim \mathcal{N}(0, 1)$. With probability $1 - \delta, \delta \in (0, 1)$:*

$$\left\| \frac{m}{n} \sum_{i=1}^n \epsilon_i \boldsymbol{z}_i \right\| \leq C \sqrt{\frac{m \log(n/\delta)(d + \log(n/\delta))\log(d/\delta)}{n}}$$

$$\left| \frac{m}{n} \sum_{i=1}^n \epsilon_i \boldsymbol{z}_i^\top \boldsymbol{v}_1 \right| \leq C' \sqrt{\frac{m \log^2(n/\delta)\log(d/\delta)}{n}}.$$

*Proof.* We use the Vector Bernstein Lemma D.10. The covariance matrix for $\boldsymbol{x}_i = \epsilon_i \boldsymbol{z}_i$ satisfies $\mathbb{E}[\boldsymbol{x}_i \boldsymbol{x}_i^\top] = 1/m I_d$. $\boldsymbol{x}_i$ is mean zero. $\|\epsilon_i \boldsymbol{z}_i\|^2 = \epsilon_i^2 \|\boldsymbol{z}_i\|^2$. Notice $\epsilon_i^2 \sim \chi(1) \lesssim 1 + \log(1/\delta)$ and $m\boldsymbol{z}_i^\top \boldsymbol{z}_i \sim \chi(d) \lesssim d + \log(1/\delta)$. Therefore by directly applying Vector Bernstein $\|\epsilon_i \boldsymbol{z}_i\| \leq c\sqrt{\frac{(1+\log(1/\delta))(d+\log(1/\delta))}{m}}$ with probability $1 - \delta$. By union bound we have: for all $i$, $\|\epsilon_i \boldsymbol{z}_i\| \leq c\sqrt{\frac{\log(n/\delta)(d+\log(n/\delta))}{m}}$ with probability $1 - \delta$. Therefore

$$\left\| \frac{1}{n} \sum_{i=1}^n \epsilon_i \boldsymbol{z}_i \right\| \leq C \sqrt{\frac{\log(n/\delta)(d + \log(n/\delta))\log(d/\delta)}{mn}},$$

with probability $1 - \delta$. Similarly

$$\left| \frac{1}{n} \sum_{i=1}^{n} \epsilon_i \boldsymbol{z}_i^\top \boldsymbol{v}_1 \right| \leq C \sqrt{\frac{\log(n/\delta)(1 + \log(n/\delta)) \log(d/\delta)}{mn}}$$

$$= C' \sqrt{\frac{\log^2(n/\delta) \log(d/\delta)}{mn}},$$

$\square$

**Claim D.12.** *Let $\{\boldsymbol{z}_i\}_{i=1}^{n}$ be i.i.d samples from $\mathcal{N}(0, 1/mI_d)$. Let $g_{n,s}(j) := \frac{1}{n} \sum_{i=1}^{n} (\boldsymbol{z}_i^\top \boldsymbol{v}_j)^s \boldsymbol{z}_i - \mathbb{E}[(\boldsymbol{z}^\top \boldsymbol{v}_j)^s \boldsymbol{z}]$. We have:*

$$\|g_{n,0}(j)\| \lesssim \sqrt{\frac{d + \log(1/\delta)}{nm}},$$

$$|\boldsymbol{v}_1^\top g_{n,0}(j)| \lesssim \sqrt{\frac{1 + \log(1/\delta)}{nm}},$$

$$|\boldsymbol{v}_1^\top g_{n,1}(j)| \leq \|g_{n,1}(j)\| \lesssim \sqrt{\frac{d + \log(1/\delta)}{m^2 n}}, \text{ when } n \geq d\log(1/\delta),$$

$$|\boldsymbol{v}_1^\top g_{n,s}(j)| \leq \|g_{n,s}(j)\| \lesssim \sqrt{\frac{\log(d/\delta)}{d^s n}}, \text{ when } n \geq \log(d/\delta), m \geq c_0 d\log(n/\delta), s \geq 2.$$

*For any $j \in [k]$.*

We mostly care about the correct concentration for smaller $s$. For larger $s$ a very loose bound will already suffice our requirement.

*Proof of Claim D.12.* For $s = 0$, $nm\|\frac{1}{n} \sum_{i=1}^{n} \boldsymbol{z}_i\|^2 \sim \chi(d)$, therefore $\|\frac{1}{n} \sum_{i=1}^{n} \boldsymbol{z}_i\| \lesssim \sqrt{\frac{d + \log(1/\delta)}{nm}}$. $nm(\frac{1}{n} \sum_{i=1}^{n} \boldsymbol{z}_i^\top \boldsymbol{v}_1)^2 \sim \chi(1)$. Therefore $|\frac{1}{n} \sum_{i=1}^{n} \boldsymbol{z}_i^\top \boldsymbol{v}_1| \lesssim \sqrt{\frac{1 + \log(1/\delta)}{nm}}$.

For $s = 1$, due to standard concentration for covariance matrices (see e.g. [65, 22] ), we have:

$$m\|(\frac{1}{n} \sum_{i=1}^{n} \boldsymbol{z}_i \boldsymbol{z}_i^\top - \mathbb{E}[\boldsymbol{z}\boldsymbol{z}^\top])\| \leq \max\{\sqrt{\frac{d + \log(2/\delta)}{n}}, \frac{d + \log(2/\delta)}{n}\}.$$

Therefore when $n \geq d\log(1/\delta)$, both results

$$\|g_{n,1}(j)\| \lesssim \sqrt{\frac{d + \log(1/\delta)}{m^2 n}} \|\boldsymbol{v}_j\|,$$

$$= \sqrt{\frac{d + \log(1/\delta)}{m^2 n}}, \text{ and}$$

$$\|\boldsymbol{v}_1^\top g_{n,1}(j)\| \lesssim \sqrt{\frac{d + \log(1/\delta)}{m^2 n}} \|\boldsymbol{v}_1\|\|\boldsymbol{v}_j\|$$

$$= \sqrt{\frac{d + \log(1/\delta)}{m^2 n}}$$

hold.

For larger $s \geq 2$, with probability $1 - \delta$, $|\boldsymbol{z}_i^\top \boldsymbol{v}_j| \leq C\sqrt{\log(n/\delta)/m} = Cc_0/\sqrt{d} \leq 1/\sqrt{d}$. When $m \geq c_0 d\log(n/\delta)$, for small enough $c_0$ we have $|\boldsymbol{z}_i^\top \boldsymbol{v}_j| \leq 1/\sqrt{d}$ and $\|\boldsymbol{z}_i\| \leq 1$ for all $i \in [n]$. Therefore $\|(\boldsymbol{z}_i^\top \boldsymbol{v}_j)^s \boldsymbol{z}_i\| \leq d^{-s/2}$ We can use vector Bernstein, i.e., Lemma D.10 to get:

$$\|g_{n,s}(j)\| \leq C_1 \sqrt{\frac{\log(d/\delta)}{d^s n}}.$$

Therefore we have:

$$|g_{n,s}(j)^\top \boldsymbol{v}_1| \leq C_1 \sqrt{\frac{\log(d/\delta)}{d^s n}}.$$

$\square$

### D.3.1 The asymmetric setting

Now we consider the asymmetric tensor problem with reward $f : \mathcal{A} \to \mathbb{R}$. The input space $\mathcal{A}$ consists of $p$ vectors in a unit ball: $\vec{a} = (\boldsymbol{a}(1), \boldsymbol{a}(2), \cdots \boldsymbol{a}(p)) \in \mathcal{A}, \|\boldsymbol{a}(s)\| \leq 1, \forall s \in [p]$. $f(\vec{a}) = \boldsymbol{T}(\otimes_{s=1}^p \boldsymbol{a}(s)) + \eta$. Tensor $\boldsymbol{T} = \sum_{j=1}^k \lambda_j \boldsymbol{v}_j(1) \otimes \boldsymbol{v}_j(2) \cdots \otimes \boldsymbol{v}_j(p)$. For each $s \in [p]$, $\{\boldsymbol{v}_1(s), \boldsymbol{v}_2(s), \cdots \boldsymbol{v}_k(s)\}$ are orthonormal vectors. We order the eigenvalues such that $\lambda_1 \geq |\lambda_2| \cdots \geq |\lambda_k|$. Therefore the optimal reward is $\lambda_1$ and can be achieved by $\boldsymbol{a}^*(s) = \boldsymbol{v}_1(s), s \in [p]$. In this section we only consider $p \geq 3$ and leave the quadratic and low-rank matrix setting to the next section.

**Theorem D.13.** *For $p \geq 3$, by conducting alternating power iteration, one can get a $\varepsilon$-optimal reward with a total $\widetilde{O}\left((2k)^p \log^p(p/\delta) d^p \lambda_1^{-1} \varepsilon^{-1}\right)$ actions; therefore the regret bound is at most $\widetilde{O}(\sqrt{k^p d^p T})$.*

This setting is actually much easier than the symmetric setting. Notice by replacing one slice of $\vec{a}$ by random Gaussian $\boldsymbol{z}_i \sim \mathcal{N}(0, 2/d \log(d/\delta))$, one directly gets $\boldsymbol{T}(\boldsymbol{a}(1), \cdots \boldsymbol{a}(s-1), \boldsymbol{I}, \boldsymbol{a}(s+1), \cdots \boldsymbol{a}(p))$ on each slice with $1/n \sum_i f(\boldsymbol{a}(1), \cdots \boldsymbol{a}(s-1), \boldsymbol{z}_i, \boldsymbol{a}(s+1), \cdots \boldsymbol{a}(p))\boldsymbol{z}_i$ which is tensor product. We defer the proof to Appendix D.4.

### D.4 Omitted Details for Asymmetric Tensors

---

**Algorithm 8** Phased elimination with alternating tensor product.

---

1: **Input:** Stochastic reward $r : (B_1^d)^{\otimes p} \to \mathbb{R}$ of polynomial degree $p$, failure probability $\delta$, error $\varepsilon$.
2: **Initialization:** $L_0 = C_L k \log(1/\delta)$; Total number of stages $S = C_S \lceil \log(1/\varepsilon) \rceil + 1$, $\mathcal{A}_0 = \{\boldsymbol{a}_0^{(1)}, \boldsymbol{a}_0^{(2)}, \cdots \boldsymbol{a}_0^{(L_0)}\} \subset (B_1^d)^{\otimes p}$ where each $\boldsymbol{a}_0^{(l)}(j), j \in [p]$ is uniformly sampled on the unit sphere $\mathbb{S}^{d-1}$. $\widetilde{\varepsilon}_0 = 1$.
3: **for** $s$ from 1 to $S$ **do**
4: $\quad \widetilde{\varepsilon}_s \leftarrow \widetilde{\varepsilon}_{s-1}/2$, $n_s \leftarrow C_n d^p \log(d/\delta)/\widetilde{\varepsilon}_s^2$, $n_s \leftarrow n_s \cdot \log^3(n_s/\delta)$, $m_s \leftarrow C_m d \log(n/\delta)$, $\mathcal{A}_s = \varnothing$.
5: $\quad$ **for** $l$ from 1 to $L_{s-1}$ **do**
6: $\quad\quad$ **Tensor product update:**
7: $\quad\quad$ Locate current arm $\widetilde{\boldsymbol{a}} = \boldsymbol{a}_{s-1}^{(l)}$.
8: $\quad\quad$ **for** $\lceil (\lambda_1/\Delta) \log(2d) \rceil$ times **do**
9: $\quad\quad\quad$ **for** $j$ from 1 to $p$ **do**
10: $\quad\quad\quad\quad$ Sample $\boldsymbol{z}_i \sim \mathcal{N}(0, 1/m_s I_d), i = 1, 2, \cdots n_s$.
11: $\quad\quad\quad\quad$ Calculate tentative arm $\boldsymbol{a}_i \leftarrow \widetilde{\boldsymbol{a}}, \boldsymbol{a}_i(j) = (1 - \widetilde{\varepsilon}_s)\widetilde{\boldsymbol{a}}(j) + \widetilde{\varepsilon}_s \boldsymbol{z}_i$
12: $\quad\quad\quad\quad$ Conduct estimation $\boldsymbol{y} \leftarrow 1/n_s \sum_{i=1}^{n_s} r_{\epsilon_i}(\boldsymbol{a}_i)\boldsymbol{z}_i$.
13: $\quad\quad\quad\quad$ Update the current arm $\widetilde{\boldsymbol{a}}(j) \leftarrow \boldsymbol{y}/\|\boldsymbol{y}\|$.
14: $\quad\quad$ Estimate the expected reward for $\widetilde{\boldsymbol{a}}$ through $n_s$ samples: $r_n = 1/n_s \sum_{i=1}^{n_s} r_{\epsilon_i}(\widetilde{\boldsymbol{a}})$.
15: $\quad\quad$ **Candidate Elimination:**
16: $\quad\quad$ **if** $r_n \geq \lambda_1(1 - p\widetilde{\varepsilon}_s)$ **then**
17: $\quad\quad\quad$ Keep the arm $\mathcal{A}_s \leftarrow \mathcal{A}_s \cup \{\widetilde{\boldsymbol{a}}\}$
18: $\quad$ Label the arms: $L_s = |\mathcal{A}_s|, \mathcal{A}_t =: \{\boldsymbol{a}_s^{(1)}, \cdots \boldsymbol{a}_s^{(L_s)}\}$.
19: Run Algorithm 7 with $\mathcal{A}_S$.

---

**Lemma D.14** (Asymmetric Tensor Initialization). *With probability $1 - \delta$, with $L = \widetilde{\Theta}((2k)^p \log^p(p/\delta))$ random initializations $\mathcal{A}_0 = \{\boldsymbol{a}_0^{(0)}, \boldsymbol{a}_0^{(1)}, \cdots \boldsymbol{a}_0^{(L)}\}$, there exists an initialization $\boldsymbol{a}_0 \in \mathcal{A}_0$ that satisfies:*

$$\alpha \boldsymbol{a}_0(s)^\top \boldsymbol{v}_1^{(s)} \geq |\boldsymbol{a}_0(s)^\top \boldsymbol{v}_j^{(s)}|, \forall j \geq 2 \& j \in [k], \forall s \in [p], \tag{8}$$

$$\boldsymbol{a}_0(s)^\top \boldsymbol{v}_1^{(s)} \geq 1/\sqrt{d}.$$

*with some constant $\alpha < 1$.*

*Proof.* This lemma simply comes from applying Lemma D.3 for $p$ times and we need $\geq 2k \log_2(p\delta)$ to ensure the condition for each $\boldsymbol{a}_0(s), s \in [p]$ holds. Therefore together we will need $(2k \log_2(p/\delta))^p$ samples. $\qquad\square$

**Lemma D.15** (Asymmetric tensor progress). *For each $a$ that satisfies Eqn. (8) with constant $\alpha < 1$, we have:*

$$\tan\theta(\boldsymbol{T}(\boldsymbol{a}(1),\cdots\boldsymbol{a}(s-1),\boldsymbol{I},\boldsymbol{a}(s+1),\cdots\boldsymbol{a}(p)),\boldsymbol{v}_1^{(s)}) \le \alpha\tan\theta(\boldsymbol{a}_j,\boldsymbol{v}_1^{(j)}),$$

*for any $j$ that is in $[p]$ but is not $s$. When $n \ge \Theta(d^p \log(d/\delta)\log^3(n/\delta)/\widetilde{\varepsilon}^2)$ and $m = \Theta(d\log(n/\delta))$, we have:*

$$\tan\theta(\boldsymbol{T}(\boldsymbol{a}(1),\cdots\boldsymbol{a}(s-1),\boldsymbol{I},\boldsymbol{a}(s+1),\cdots\boldsymbol{a}(p)),\boldsymbol{v}_1^{(s)})$$
$$\le (1+\alpha)/2\tan\theta(\boldsymbol{a}_j,\boldsymbol{v}_1^{(j)}) + \widetilde{\varepsilon}, \forall j \in [p]\&j \ne s.$$

The remaining proof is a simpler version for the symmetric tensor setting on conducting noisy power method with the good initialization and iterative progress.

Finally due to the good initialization that satisfies (8) and together with Lemma D.15 we can finish the proof for Theorem D.13.

# E    Proof of Theorem 3.21

## E.1    Additional Notations

Here, we briefly introduce complex and real algebraic geometry. This section is based on [56, 64, 12, 69].

An (affine) **algebraic variety** is the common zero loci of a set of polynomials, defined as $V = Z(S) = \{\boldsymbol{x} \in \mathbb{C}^n : f(\boldsymbol{x}) = 0, \forall f \in S\} \subseteq \mathbb{A}^n = \mathbb{C}^n$ for some $S \subseteq \mathbb{C}[x_1,\cdots,x_n]$. A **projective variety** $U$ is a subset of $\mathbb{P}^n = (\mathbb{C}^{n+1} \setminus \{0\})/\sim$, where $(x_0,\cdots,x_n) \sim k(x_0,\cdots,x_n)$ for $k \ne 0$ and $S$ is a set of homogeneous polynomials of $(n+1)$ variables.

For an affine variety $V$, its **projectivization** is the variety $\mathbb{P}(V) = \{[\boldsymbol{x}] : x \in V\} \subseteq \mathbb{P}^{n-1}$, where $[\boldsymbol{x}]$ is the line corresponding to $\boldsymbol{x}$.

The **Zariski topology** is the topology generated by taking all varieties to be the closed sets.

A set is **irreducible** if it is not the union of two proper closed subsets.

A **variety is irreducible** if and only if it is irreducible under the Zariski topology.

The **algebraic dimension** $d = \dim V$ of a variety $V$ is defined as the length of the longest chain $V_0 \subset V_1 \subset \cdots \subset V_d = V$, such that each $V_i$ is irreducible.

A variety $V$ is said to be **admissible** to a set of linear functions $\{\ell_\alpha : \mathbb{C}^d \to \mathbb{C}\}_{\alpha \in I}$, if for every $\ell_\alpha$, we have $\dim(V \cap \{\boldsymbol{x} \in \mathbb{C}^d : \ell_\alpha(\boldsymbol{x}) = 0\}) < \dim V$.

A map $f = (f_1,\cdots,f_m) : \mathbb{A}^n \to \mathbb{A}^m$ is **regular** if each $f_i$ is a polynomial.

A **algebraic set** is the common real zero loci of a set of polynomials.

For a complex variety $V \subseteq \mathbb{A}^n$, its real points form a algebraic set $V_{\mathbb{R}}$.

For an algebraic set $V_{\mathbb{R}}$, its real dimension $d = \dim_{\mathbb{R}} V_{\mathbb{R}}$ is the maximum number $d$ such that $V_{\mathbb{R}}$ is locally semi-algebraically homeomorphic to the unit cube $(0,1)^d$, details can be found in [12].

## E.2    Proof of Sample Complexity

**Lemma E.1** ([69], Theorem 3.2). *For $i = 1,\ldots,T$, let $L_i : \mathbb{C}^n \times \mathbb{C}^m \to \mathbb{C}$ be bilinear functions and $V_i$ be varieties given by homogeneous polynomials in $\mathbb{C}^n$. Let $V = V_1 \times \cdots \times V_T \subseteq (\mathbb{C}^n)^N$. Let $W \subseteq \mathbb{C}^m$ be a variety given by homogeneous polynomials. In addition, we assume $V_i$ is admissible with respect to the linear functions $\{f^{\boldsymbol{w}}(\cdot) = L_i(\cdot,\boldsymbol{w}) : \boldsymbol{w} \in W \setminus \{0\}\}$. When $T \ge \dim W$, let $\delta = T - \dim W + 1 \ge 1$. Then there exists a subvariety $Z \subseteq V$ with $\dim Z \le \dim V - \delta$ such that for any $(\boldsymbol{x}_1,\ldots,\boldsymbol{x}_T) \in V \setminus Z$ and $\boldsymbol{w} \in W$, if $L_1(\boldsymbol{x}_1,\boldsymbol{w}) = \cdots = L_T(\boldsymbol{x}_T,\boldsymbol{w}) = 0$, then $\boldsymbol{w} = 0$.*

**Lemma E.2** ([69], Lemma 3.1). *Let $V$ be an algebraic variety in $\mathbb{C}^d$. Then $\dim_{\mathbb{R}} V_{\mathbb{R}} \le \dim V$.*

**Lemma E.3.** *Let $W$ be a vector space. For vectors $\boldsymbol{x}_1,\cdots,\boldsymbol{x}_T$, if the map $f : \boldsymbol{w} \mapsto (\langle\boldsymbol{x}_1,w\rangle,\ldots,\langle\boldsymbol{x}_T,\boldsymbol{w}\rangle)$ is not injective over $W - W := \{\boldsymbol{w}_1 - \boldsymbol{w}_2 : \boldsymbol{w}_1,\boldsymbol{w}_2 \in W\}$, then there exists $\boldsymbol{v} \in W$ such that $f(\boldsymbol{v}) = 0$.*

*Proof.* Suppose $f(\boldsymbol{w}_1) = f(\boldsymbol{w}_2)$. Let $\boldsymbol{v} = \boldsymbol{w}_1 - \boldsymbol{w}_2$. Then $\boldsymbol{v} \in W - W$ and $f(\boldsymbol{v}) = f(\boldsymbol{w}_1) - f(\boldsymbol{w}_2) = 0$. $\qquad\square$

**Definition E.4** (Tensorization). Let $f$ be a polynomial of $x_1, \cdots, x_d$ with degree $\deg f \leq p$. Then every $p$-tensor $\mathbf{W}_f$ satisfying $\langle \mathbf{W}_f, \mathbf{X}_{\boldsymbol{x}} \rangle = f(\boldsymbol{x})$ is said to be a *tensorization* of the polynomial $f$, where $\mathbf{X}_{\boldsymbol{x}}$ is the tensorization of $\boldsymbol{x}$ itself:

$$\mathbf{X}_{\boldsymbol{x}} = \begin{pmatrix} 1 \\ \boldsymbol{x} \end{pmatrix}^{\otimes p}. \tag{9}$$

Let $\mathcal{F}$ be a class of polynomials. A variety of tensorization of $\mathcal{F}$ is defined to be an irreducible closed variety defined by homogeneous polynomials $W$, such that for every $f \in \mathcal{F}$, there is a tensorization $\mathbf{W}_f$ of $f$, such that $W \ni \mathbf{W}_f$ contains its tensorization. Note that neither tensorization of $f$ nor variety of tensorization of $\mathcal{F}$ is unique.

We define the variety of tensorization of $\boldsymbol{x}$ as follows. (Note that this is uniquely defined.) Consider the regular map

$$\varphi_1 : \mathbb{C}^d \to \mathbb{C}^{(d+1)^p}, \qquad \boldsymbol{x} \mapsto \begin{pmatrix} 1 \\ \boldsymbol{x} \end{pmatrix}^{\otimes p}, \tag{10}$$

the tensorization of $\boldsymbol{x}$ is defined as $V_i = \mathbb{P}(\overline{\operatorname{Im} \varphi_1})$.

Note that $V_i$ is irreducible because $\varphi_1$ is regular and $\mathbb{C}^d$ is irreducible. By [56, Theorem 9.9], its dimension is given by

$$\dim V_i \leq \dim \overline{\operatorname{Im} \varphi_1} + 1 \leq \dim \mathbb{C}^d + 1 = d + 1. \tag{11}$$

**Lemma E.5.** *For any non-zero polynomial $f \neq 0$ with $\deg f \leq p$. Let $W_f$ be a tensorization of $f$. Then $V_i$ is admissible with respect to $\{L_i(\cdot) = \langle \cdot, W_f \rangle\}$.*

*Proof.* Since $V_i$ is irreducible and $L_i$ is a linear function, it suffices to verify that $\langle \mathbf{X}_{\boldsymbol{x}}, W_f \rangle \neq 0$ [69]. But according to Definition E.4, $\langle \mathbf{X}_{\boldsymbol{x}}, W_f \rangle \neq 0$ is equivalent to

$$f(\boldsymbol{x}) = \left\langle \mathbf{W}_f, \begin{pmatrix} 1 \\ \boldsymbol{x} \end{pmatrix}^{\otimes p} \right\rangle \neq 0. \tag{12}$$

Since $f \neq 0$, we must have $f(\boldsymbol{x}) \neq 0$ for some $\boldsymbol{x}$, which gives a non-zero $\mathbf{X}_{\boldsymbol{x}} \neq 0$ for the above equation: $\langle \mathbf{X}_x, W_f \rangle \neq 0$, and we conclude that $V_i$ is admissible. $\qquad\square$

**Lemma E.6.** *Let $V \subset \mathbb{C}^n$ be a (Zariski) closed proper subset, $V \neq \mathbb{C}^n$. Then $V$ is a null set, i.e. it has (Lebesgue) measure zero.*

*Proof.* Suppose $V = Z(S)$ is the vanishing set for some $S \subseteq \mathbb{C}[x_1, \cdots, x_n]$. Since $V \neq \mathbb{C}^n$, let $f \in S$, we have $V \subseteq Z(f)$, so it suffices to show $\operatorname{Leb}(Z(f)) = 0$, which is because $Z(f) = f^{-1}(0), \operatorname{Leb}(\{0\}) = 0, f$ is a continuous function (under Euclidean topology), and $\operatorname{Leb}(\{\boldsymbol{x} : \nabla f(\boldsymbol{x}) = 0\}) = 0$. $\qquad\square$

**Theorem E.7.** *Assume that the reward function class is a class of polynomials $\mathcal{F}$. Let $W$ be (one of) its variety of tensorization. If we sample $T \geq \dim W$ times, and the sample points satisfying $(\boldsymbol{x}_1, \cdots, \boldsymbol{x}_T) \in (\mathbb{C}^d)^T \setminus Z$ for some null set $Z$. Then we can uniquely determine the reward function $f$ from the observed rewards $(f(\boldsymbol{x}_1), \cdots, f(\boldsymbol{x}_T))$.*

*Proof.* Let $n = m = (d+1)^p, L_i(\boldsymbol{x}, \boldsymbol{w}) = \langle \boldsymbol{x}, \boldsymbol{w} \rangle, V = V_1 \times \cdots \times V_T$, where $V_i$ is as in Definition E.4. By [64, Example 1.33], we have $\dim V \leq (d+1)T$. Since $W$ is a vareity of tensorization, by Lemma E.5, $V_i$ is admissible with respect to $\{L_i(\cdot, \mathbf{W}) : \mathbf{W} \in W\}$.

We are now ready to apply Lemma E.1, which gives that when $T \geq \dim W$, there exists subvariety $Z \subset V$ with $\dim Z < \dim V \leq rT$, and for any $(\mathbf{X}_1, \cdots, \mathbf{X}_T) \in V \setminus Z$ and any $\mathbf{W} \in W$, if $\langle \mathbf{X}_1, \mathbf{W} \rangle = \cdots = \langle \mathbf{X}_T, \mathbf{W} \rangle = 0$, then $\mathbf{W} = 0$. By Lemma E.3, we have for every $(\mathbf{X}_1, \cdots, \mathbf{X}_T) \in V \setminus Z$, the map $\mathbf{W} \mapsto (\langle \mathbf{X}_1, \mathbf{W} \rangle, \cdots, \langle \mathbf{X}_T, \mathbf{W} \rangle)$ is injective, so $\mathbf{W}_f$ and thus $f$ can be uniquely recovered from the observed rewards.

Finally, we show that $(\varphi_1^{-1} \times \cdots \times \varphi_1^{-1})(Z)$ is a null set, where $\varphi_1$ is as in (10). According to the proof of Lemma E.1 by [69], we find that $Z$ is also defined by homogeneous polynomials. We take the slice $Z' = \{\boldsymbol{x} \in Z : x_{11} = \cdots = x_{T1} = 1\}$, $V' = \{\boldsymbol{x} \in V : x_{11} = \cdots = x_{T1} = 1\}$, (here $x_{ij}$ is the $j$-th coordinate of $\boldsymbol{x}_i$), then $Z', V'$ are varieties. Since $\dim Z < \dim V$, we have $\dim Z' = \dim Z - T < \dim V - T = \dim V'$ and $Z' \subset V'$.

Now consider the regular map $\varphi_1' : V' \to (\mathbb{C}^d)^T$,

$$\left( \begin{pmatrix} 1 \\ \boldsymbol{x}_1 \end{pmatrix}^{\otimes p}, \cdots, \begin{pmatrix} 1 \\ \boldsymbol{x}_1 \end{pmatrix}^{\otimes p} \right) \mapsto (\boldsymbol{x}_1, \cdots, \boldsymbol{x}_T). \tag{13}$$

Then $\varphi_1'(Z'), \varphi_1'(V')$ are both varieties. By [56, Lemma 9.9], we have $\dim \overline{\varphi_1'(Z')} \leq \dim Z$. Since $\varphi_1'(V) = (\mathbb{C}^d)^T$ and $\dim \varphi_1'(V') \leq \dim V' = \dim V - T \leq (d+1)T - T$, we have $\dim V = \dim \varphi_1'(V) = dT$ and as a result, $\dim \overline{\varphi_1'(Z)} \leq \dim Z < \dim V = dT$. By Lemma E.6, $\overline{\varphi_1'(Z)}$ is a null set. Since $(\boldsymbol{x}_1, \cdots, \boldsymbol{x}_T) \notin \overline{\varphi_4'(Z)}$ implies that $(\varphi_1(\boldsymbol{x}_1), \cdots, \varphi_1(\boldsymbol{x}_T)) \notin Z$, we conclude the proof. $\qquad\square$

Theorem E.7 is stated for complex sample points. Next we extend it to the real case.

**Lemma E.8.** *In Lemma E.1, if we assume in addition that $\dim_{\mathbb{R}} V_{\mathbb{R}} = \dim V$, then the conclusion can be enhanced to ensure that $Z$ is a real subvariety and $\dim_{\mathbb{R}} Z < \dim_{\mathbb{R}} V_{\mathbb{R}}$.*

**Lemma E.9.** *Let $V \subset \mathbb{R}^n$ be a (Zariski) closed proper subset, $V \neq \mathbb{R}^n$. Then $V$ is a null set.*

The proof of Lemma E.9 is the same as that of Lemma E.6.

**Theorem E.10.** *We can additionally assume $\boldsymbol{x}_i \in \mathbb{R}^d$ in Theorem E.7.*

*Proof.* We verify that $\dim V = \dim_{\mathbb{R}} V_{\mathbb{R}}$, where $V$ is defined in the proof of Theorem E.7, but this follows clearly by [12, Corollary 2.8.2]. We conclude the proof by applying Lemma E.8. $\qquad\square$

Finally, we apply Theorem E.10 to two concrete classes of polynomials, namely Examples 3.22 and 3.23. For Example 3.22, we construct its variety of tensorization of $\mathcal{R}_{\mathcal{V}}$ as follows. We first construct the tensorization of each polynomial. We define

$$\mathbf{W}_f = \sum_{i=1}^r a_i \begin{pmatrix} 1 \\ \boldsymbol{w}_i \end{pmatrix}^{\otimes p_i} \otimes \begin{pmatrix} 1 \\ 0 \end{pmatrix}^{\otimes(p-p_i)}. \tag{14}$$

Next we construct the variety of tensorization $W$. Consider the map $\varphi_2 : (\mathbb{C}^d)^r \to \mathbb{C}^{(d+1)^p}$,

$$\varphi_2(\boldsymbol{w}_1, \cdots, \boldsymbol{w}_r) = \sum_{i=1}^r \begin{pmatrix} 1 \\ \boldsymbol{w}_i \end{pmatrix}^{\otimes p_i} \otimes \begin{pmatrix} 1 \\ 0 \end{pmatrix}^{\otimes(p-p_i)}, \tag{15}$$

and let $Y = \mathbb{P}(\overline{\operatorname{Im} \varphi_2})$. Similar to $V_i$, we can prove that $Y$ is an irreducible closed variety defined by homogeneous polynomials with $\dim Y \leq dr + 1$. Next consider the map $\varphi_2' : (\mathbb{C}^d)^{2r} \to \mathbb{C}^{(d+1)^p}$,

$$\varphi_2'(\boldsymbol{w}_1, \cdots, \boldsymbol{w}_{2r}) = \varphi_2(\boldsymbol{w}_1, \cdots, \boldsymbol{w}_r) - \varphi_2(\boldsymbol{w}_{r+1}, \cdots, \boldsymbol{w}_{2r}) \tag{16}$$

and let $W = \mathbb{P}(\overline{\operatorname{Im} \varphi_2'})$. Similar to $Y$, we can prove that $W$ is an irreducible closed variety defined by homogeneous polynomials with $\dim W \leq 2dr + 1$. Together with Theorem E.10, we can conlude that the optimal action for Example 3.22 can be uniquely determined using at most $2dr + 1$ samples.

For Example 3.23, we construct $W$ as follows. Let

$$\boldsymbol{U} = \begin{pmatrix} \boldsymbol{w}_1 & \cdots & \boldsymbol{w}_k \end{pmatrix}, \qquad q = \sum_{I \subseteq [k]:|I| \leq p} a_I x^I,$$

then we construct the tensorization of each polynomial by

$$\mathbf{W}_f = \sum_{I \subseteq [k]:|I| \leq p} a_I \bigotimes_{i \in I} \begin{pmatrix} 1 \\ \boldsymbol{w}_i \end{pmatrix} \otimes \begin{pmatrix} 1 \\ 0 \end{pmatrix}^{\otimes(p-|I|)}. \tag{17}$$

Then we have $f(\boldsymbol{x}) = \langle \mathbf{W}_f, \mathbf{X}_{\boldsymbol{x}} \rangle$. To reduce the dimension of $W$ and get better sample complexity bound, we construct in a manner slightly different from what we did for Example 3.22. Consider the map $\varphi_3 : (\mathbb{C}^d)^k \times \mathbb{C}^{(k+1)^p} \to \mathbb{C}^{(d+1)^p}$,

$$(\boldsymbol{w}_1, \cdots, \boldsymbol{w}_k) \times (a_I : I \subseteq [k], |I| \leq p) \mapsto \mathbf{W}_f, \tag{18}$$

where $\mathbf{W}_f$ is as defined in (17). Let $Y = \mathbb{P}(\overline{\operatorname{Im} \varphi_3})$ and $W = \mathbb{P}(\overline{\operatorname{Im} \varphi_3 - \operatorname{Im} \varphi_3})$. We end up with $\dim Y = \leq dk + (k+1)^p + 1$, $\dim W \leq 2(dk + (k+1)^p) + 1$. So we conlude that the optimal action for Example 3.23 can be uniquely determined using at most $2dk + 2(k+1)^p + 1$ samples.

# F Omitted Proof for Lower Bounds with UCB Algorithms

In this section, we provide the proof for the lower bounds for learning with UCB algorithms in Subsection G.0.1.

**Notation** Recall that we use $\Lambda$ to denote the subset of the $p$-th multi-indices $\Lambda = \{(\alpha_1, \ldots, \alpha_p) | 1 \leq \alpha_1 < \cdots < \alpha_p \leq d\}$. For an $\alpha = (\alpha_1, \ldots, \alpha_p) \in \Lambda$, denote $\boldsymbol{M}_\alpha = \boldsymbol{e}_{\alpha_1} \otimes \cdots \otimes \boldsymbol{e}_{\alpha_p}$, $\boldsymbol{A}_\alpha = (\boldsymbol{e}_{\alpha_1} + \cdots + \boldsymbol{e}_{\alpha_p})^{\otimes p}$. The model space $\mathcal{M}$ is a subset of rank-1 $p$-th order tensors, which is defined as $\mathcal{M} = \left\{ \boldsymbol{M}_\alpha | \alpha \in \Lambda \right\}$. We define the core action set $\mathcal{A}_0$ as $\mathcal{A}_0 = \{ \boldsymbol{e}_{\alpha_1} + \cdots + \boldsymbol{e}_{\alpha_p} | \alpha \in \Lambda \}$. The action set $\mathcal{A}$ is the convex hull of $\mathcal{A}_0$: $\mathcal{A} = \operatorname{conv}(\mathcal{A}_0)$. Assume that the ground-truth parameter is $\boldsymbol{M}^* = \boldsymbol{M}_{\alpha^*} \in \mathcal{M}$. At round $t$, the algorithm chooses an action $\boldsymbol{a}_t \in \mathcal{A}$, and gets the **noiseless** reward $r_t = r(\boldsymbol{M}^*, \boldsymbol{a}_t) = \langle \boldsymbol{M}^*, (\boldsymbol{a}_t)^{\otimes p} \rangle = \prod_{i=1}^p \langle \boldsymbol{e}_{\alpha_i^*}, \boldsymbol{a}_t \rangle$.

## F.1 Proof for Theorem G.2

We introduce a lemma showing that if the action set is **restricted** to the core action set $\mathcal{A}_0$, then at least $|\mathcal{A}_0| - 1 = \binom{d}{p} - 1$ actions are needed to identify the ground-truth.

**Lemma F.1.** *If the actions are restricted to $\mathcal{A}_0$, then for the noiseless degree-$p$ polynomial bandits, any algorithm needs to play at least $\binom{d}{p} - 1$ actions to determine $\boldsymbol{M}^*$ in the worst case. Furthermore, the worst-case cumulative regret at round $T$ can be lower bounded by*

$$\mathfrak{R}(T) \geq \min\{T, \binom{d}{p} - 1\}.$$

*proof of Lemma F.1.* For any $\alpha$ and $\alpha'$, the reward of playing $\boldsymbol{e}_{\alpha_1} + \cdots + \boldsymbol{e}_{\alpha_p}$ when the ground-truth model is $\boldsymbol{M}'_\alpha$ is

$$\langle \boldsymbol{M}'_\alpha, (\boldsymbol{e}_{\alpha_1} + \cdots + \boldsymbol{e}_{\alpha_p})^{\otimes p} \rangle = \prod_{i=1}^p \langle \boldsymbol{e}_{\alpha'_i}, \boldsymbol{e}_{\alpha_1} + \cdots + \boldsymbol{e}_{\alpha_p} \rangle$$

$$= \prod_{i=1}^p \mathbb{I}\{\alpha'_i \in \alpha\}$$

$$= \begin{cases} 1, & \text{if } \alpha = \alpha' \\ 0, & \text{otherwise .} \end{cases}$$

Hence, no matter how the algorithm adaptively chooses the actions, in the worst case $\binom{d}{p} - 1$ actions are needed to determine $\boldsymbol{M}^*$. Also notice that the reward for $\boldsymbol{e}_{\alpha_1} + \cdots + \boldsymbol{e}_{\alpha_p}$ is zero if $\alpha \neq \alpha^*$. Therefore the regret lower bound follows. $\square$

Next, we show that even when the action set is unrestricted, any UCB algorithm fails to explore in an unrestricted way. This is because the optimistic mechanism forbids the algorithm to play an informative action that is known to be low reward for all models in the confidence set. We first recall the definition of UCB algorithms.

**UCB Algorithms**  The UCB algorithms sequentially maintain a confidence set $\mathcal{C}_t$ after playing actions $\boldsymbol{a}_1, \ldots, \boldsymbol{a}_t$. Then UCB algorithms play

$$\boldsymbol{a}_{t+1} \in \arg\max_{\boldsymbol{a} \in \mathcal{A}} \mathrm{UCB}_t(\boldsymbol{a}),$$

where

$$\mathrm{UCB}_t(\boldsymbol{a}) = \max_{\boldsymbol{M} \in \mathcal{C}_t} \langle \boldsymbol{M}, (\boldsymbol{a})^{\otimes p} \rangle.$$

*proof of Theorem G.2.*  We prove that even if the action set is unrestrcited, the optimistic mechanism in the UCB algorithm above forces it to choose actions in the restricted action set $\mathcal{A}_0$.

Assume $\boldsymbol{M}^* = \boldsymbol{M}_{\alpha^*}$. Next we show that for all $\boldsymbol{a} \in \mathcal{A} - \mathcal{A}_0$ (where the minus sign should be understood as set difference), we have

$$\mathrm{UCB}_t(\boldsymbol{a}) < 1.$$

For all $\boldsymbol{a} \in \mathcal{A}$, since $\mathcal{A} = \mathrm{conv}(\mathcal{A}_0)$, we can write

$$\boldsymbol{a} = \sum_{\alpha \in \Lambda} p_\alpha (\boldsymbol{e}_{\alpha_1} + \cdots + \boldsymbol{e}_{\alpha_p}),$$

where $\sum_{\alpha \in \Lambda} p_\alpha = 1$ and $p_\alpha \geq 0$. Therefore,

$$\begin{aligned}
\mathrm{UCB}_t(\boldsymbol{a}) &= \max_{\boldsymbol{M} \in \mathcal{C}_t} \langle \boldsymbol{M}, (\boldsymbol{a})^{\otimes p} \rangle \\
&\leq \max_{\boldsymbol{M} \in \mathcal{M}} \langle \boldsymbol{M}, (\boldsymbol{a})^{\otimes p} \rangle \\
&= \max_{\alpha'} \langle \boldsymbol{M}_{\alpha'}, (\boldsymbol{a})^{\otimes p} \rangle \\
&= \max_{\alpha'} \prod_{i=1}^{p} \langle \boldsymbol{e}_{\alpha_i'}, \boldsymbol{a} \rangle.
\end{aligned}$$

Plug in the expression of $\boldsymbol{a}$, we have

$$\begin{aligned}
\langle \boldsymbol{e}_{\alpha_i'}, \boldsymbol{a} \rangle &= \sum_{\alpha} p_\alpha \langle \boldsymbol{e}_{\alpha_i'}, \boldsymbol{e}_{\alpha_1} + \cdots + \boldsymbol{e}_{\alpha_p} \rangle \\
&= \sum_{\alpha} p_\alpha \mathbb{I}\{\alpha_i' \in \alpha\} \\
&\leq \sum_{\alpha} p_\alpha = 1.
\end{aligned}$$

Therefore, for any fixed $\alpha' = (\alpha_1', \ldots, \alpha_p')$,

$$\begin{aligned}
\prod_{i=1}^{p} \langle \boldsymbol{e}_{\alpha_i'}, \boldsymbol{a} \rangle &= \left( \sum_{\alpha} p_\alpha \mathbb{I}\{\alpha_1' \in \alpha\} \right) \cdots \left( \sum_{\alpha} p_\alpha \mathbb{I}\{\alpha_p' \in \alpha\} \right) \\
&\leq 1,
\end{aligned}$$

where the equality holds if and only if for any $p_\alpha > 0$, $\alpha = \alpha'$, which is equivalent to $\boldsymbol{a} = \boldsymbol{e}_{\alpha_1'} + \cdots + \boldsymbol{e}_{\alpha_p'}$. Therefore, if $\boldsymbol{a} \in \mathcal{A} - \mathcal{A}_0$, for any $\alpha' \in \Lambda$, we have $\prod_{i=1}^{p} \langle \boldsymbol{e}_{\alpha_i'}, \boldsymbol{a} \rangle < 1$. This means

$$\mathrm{UCB}_t(\boldsymbol{a}) < 1.$$

Meanwhile, we can see that for the action $\boldsymbol{a}^* = \boldsymbol{e}_{\alpha_1^*} + \cdots + \boldsymbol{e}_{\alpha_p^*} \in \mathcal{A}_0$,

$$\begin{aligned}
\mathrm{UCB}_t(\boldsymbol{a}^*) &= \max_{\boldsymbol{M} \in \mathcal{C}_t} \langle \boldsymbol{M}, (\boldsymbol{a}^*)^{\otimes p} \rangle \\
&\geq \langle \boldsymbol{M}^*, (\boldsymbol{a}^*)^{\otimes p} \rangle && (\boldsymbol{M}^* \in \mathcal{C}_t) \\
&= \langle \boldsymbol{M}^*, \boldsymbol{A}_{\alpha^*} \rangle = 1.
\end{aligned}$$

Therefore, we see that $(\mathcal{A} - \mathcal{A}_0) \cap \arg\max_{\boldsymbol{a} \in \mathcal{A}} \mathrm{UCB}_t(\boldsymbol{a}) = \varnothing$, which means $\boldsymbol{a}_{t+1} \in \mathcal{A}_0$ for all $t \geq 0$. Therefore, by Lemma F.1, the theorem holds.

$\square$

## F.2 $O(d)$ Actions via Solving Polynomial Equations

Firstly, we verify that the model falls into the category of Example 3.23 with $k = p$. For every $\alpha \in \Lambda$, the reward of playing $\boldsymbol{a}$ when the ground-truth model is $\boldsymbol{M}_\alpha$ is

$$\langle \boldsymbol{M}_\alpha, (\boldsymbol{a})^{\otimes p} \rangle = \prod_{i=1}^{p} \langle \boldsymbol{e}_{\alpha_i}, \boldsymbol{a} \rangle,$$

which can be written as $q_0(\boldsymbol{U}_\alpha \boldsymbol{a})$, where $q_0(x_1, \ldots, x_p) = x_1 x_2 \cdots x_p$ and $\boldsymbol{U}_\alpha \in \mathbb{R}^{p \times d}$ is a matrix with $\boldsymbol{e}_{\alpha_i}$ as the $i$-th row.

Secondly, we show that since the ground-truth model is $p$-homogenous, we can extend the action set to $\mathrm{conv}(\mathcal{A}, \boldsymbol{0})$. This is because for every action of the form $c\boldsymbol{a}$, where $0 \le c \le 1$ and $\boldsymbol{a} \in \mathcal{A}$, the reward is $c^p$ times the reward at $\boldsymbol{a}$. Therefore, to get the reward at $c\boldsymbol{a}$, we only need to play at $\boldsymbol{a}$ and multiply the reward by $c^p$.

Notice that $\mathrm{conv}(\mathcal{A}, \boldsymbol{0})$ is of positive Lebesgue measure. By Theorem 3.21, we know that only $2(dk + (p+1)^p) = O(d)$ actions are needed to determine the optimal action almost surely.

# G  Proof of Section 3.3.2

We present the proof of Theorem 3.18 in the following.

*Proof.* We overload the notation and use $[d]$ to denote the set $\{e_1, e_2, \ldots, e_d\}$. The hard instances are chosen in $\Delta \cdot [d]^p$, i.e. $(\boldsymbol{\theta}_1, \ldots, \boldsymbol{\theta}_p) = \Delta \cdot (\widehat{\boldsymbol{\theta}}_1, \ldots, \widehat{\boldsymbol{\theta}}_p)$ where $(\widehat{\boldsymbol{\theta}}_1, \ldots, \widehat{\boldsymbol{\theta}}_p) \in [d]^p$. For a group of vectors $\boldsymbol{\theta}_1, \ldots, \boldsymbol{\theta}_p \in [d]$, we use

$$\mathrm{supp}(\boldsymbol{\theta}_1, \ldots, \boldsymbol{\theta}_p) := (\max_{i \in [p]} (\boldsymbol{\theta}_i)_1, \ldots, \max_{i \in [p]} (\boldsymbol{\theta}_i)_d) \in \{0, 1\}^d$$

to denote the support of these vectors. We use $\boldsymbol{a}^{(t)} \in \mathbb{R}^d$ to denote the action in $t$-th episode.

We use $\mathbb{P}_{(\boldsymbol{\theta}_1, \ldots, \boldsymbol{\theta}_p)}$ to denote the measure on outcomes induced by the interaction of the fixed policy and the bandit paramerised by $r = \prod_{i=1}^{p} (\boldsymbol{\theta}_i^\top \boldsymbol{a}) + \epsilon$. Specifically, We use $\mathbb{P}_0$ to denote the measure on outcomes induced by the interaction of the fixed policy and the pure noise bandit $r = \epsilon$.

$$\mathfrak{R}(d, p, T)$$

$$\ge \frac{1}{d^p} \sum_{(\boldsymbol{\theta}_1, \ldots, \boldsymbol{\theta}_p) \in \Delta \cdot [d]^p} \mathbb{E}_{(\boldsymbol{\theta}_1, \ldots, \boldsymbol{\theta}_p)} \left[ T\Delta^p / p^{p/2} - \sum_{t=1}^{T} \prod_{i=1}^{p} (\boldsymbol{\theta}_i^\top \boldsymbol{a}^{(t)}) \right]$$

$$= \frac{\Delta^p}{d^p} \sum_{(\boldsymbol{\theta}_1, \ldots, \boldsymbol{\theta}_p) \in \Delta \cdot [d]^p} \left( T / p^{p/2} - \mathbb{E}_{(\boldsymbol{\theta}_1, \ldots, \boldsymbol{\theta}_p)} \left[ \sum_{t=1}^{T} \prod_{i=1}^{p} (\widehat{\boldsymbol{\theta}}_i^\top \boldsymbol{a}^{(t)}) \right] \right)$$

$$\ge \frac{\Delta^p}{d^p} \sum_{(\boldsymbol{\theta}_1, \ldots, \boldsymbol{\theta}_p) \in \Delta \cdot [d]^p} \left( T / p^{p/2} - \mathbb{E}_0 \left[ \sum_{t=1}^{T} \prod_{i=1}^{p} (\widehat{\boldsymbol{\theta}}_i^\top \boldsymbol{a}^{(t)}) \right] - T \| \mathbb{P}_0 - \mathbb{P}_{(\boldsymbol{\theta}_1, \ldots, \boldsymbol{\theta}_p)} \|_{\mathrm{TV}} \right)$$

$$\ge \frac{\Delta^p}{d^p} \sum_{(\boldsymbol{\theta}_1, \ldots, \boldsymbol{\theta}_p) \in \Delta \cdot [d]^p} \left( T / p^{p/2} - \mathbb{E}_0 \left[ \sum_{t=1}^{T} \prod_{i=1}^{p} (\widehat{\boldsymbol{\theta}}_i^\top \boldsymbol{a}^{(t)}) \right] - T \sqrt{D_{\mathrm{KL}}(\mathbb{P}_0 \| \mathbb{P}_{(\boldsymbol{\theta}_1, \ldots, \boldsymbol{\theta}_p)})} \right)$$

$$\ge \frac{\Delta^p}{d^p} \sum_{(\boldsymbol{\theta}_1, \ldots, \boldsymbol{\theta}_p) \in \Delta \cdot [d]^p} \left( T / p^{p/2} - \mathbb{E}_0 \left[ \sum_{t=1}^{T} \prod_{i=1}^{p} (\widehat{\boldsymbol{\theta}}_i^\top \boldsymbol{a}^{(t)}) \right] - T \sqrt{\Delta^{2p} \mathbb{E}_0 \left[ \sum_{t=1}^{T} \prod_{i=1}^{p} (\widehat{\boldsymbol{\theta}}_i^\top \boldsymbol{a}^{(t)})^2 \right]} \right)$$

$$\ge \frac{\Delta^p}{d^p} \left( \frac{d^p T}{p^{\frac{p}{2}}} - \mathbb{E}_0 \left[ \sum_{(\widehat{\boldsymbol{\theta}}_1, \ldots, \widehat{\boldsymbol{\theta}}_p) \in [d]^p} \sum_{t=1}^{T} \prod_{i=1}^{p} (\widehat{\boldsymbol{\theta}}_i^\top \boldsymbol{a}^{(t)}) \right] - T d^{\frac{p}{2}} \Delta^p \sqrt{\mathbb{E}_0 \left[ \sum_{(\widehat{\boldsymbol{\theta}}_1, \ldots, \widehat{\boldsymbol{\theta}}_p) \in [d]^p} \sum_{t=1}^{T} \prod_{i=1}^{p} (\widehat{\boldsymbol{\theta}}_i^\top \boldsymbol{a}^{(t)})^2 \right]} \right)$$

where the first step comes from

$$\text{Regret} \geq \mathbb{E}_{(\boldsymbol{\theta}_1,\ldots,\boldsymbol{\theta}_p)} \left[ T\Delta^p/p^{p/2} - \sum_{t=1}^{T} \prod_{i=1}^{p} (\boldsymbol{\theta}_i^\top \boldsymbol{a}^{(t)}) \right]$$

(the optimal action in hindsight is $\boldsymbol{a} = \text{supp}(\boldsymbol{\theta}_1,\ldots,\boldsymbol{\theta}_p)/\sqrt{p}$; the second step comes from $(\boldsymbol{\theta}_1,\ldots,\boldsymbol{\theta}_p) = \Delta \cdot (\widehat{\boldsymbol{\theta}}_1,\ldots,\widehat{\boldsymbol{\theta}}_p)$ and algebra; the third step comes from $\left| \sum_{t=1}^{T} \prod_{i=1}^{p} (\widehat{\boldsymbol{\theta}}_i^\top \boldsymbol{a}^{(t)}) \right| \leq T$; the fourth step comes from Pinsker's inequality; the fifth step comes from

$$D_{\text{KL}}(\mathbb{P}_0 || \mathbb{P}_{\boldsymbol{\theta}_1,\ldots,\boldsymbol{\theta}_p}) = \mathbb{E}_0 \left[ \sum_{t=1}^{T} D_{\text{KL}} \left( N(0,1) || N(\prod_{i=1}^{p} (\boldsymbol{\theta}_i^\top \boldsymbol{a}^{(t)}), 1) \right) \right]$$

$$= \Delta^{2p} \mathbb{E}_0 \left[ \sum_{t=1}^{T} \prod_{i=1}^{p} (\widehat{\boldsymbol{\theta}}_i^\top \boldsymbol{a}^{(t)})^2 \right]$$

and the final step comes from Jensen's inequality and algebra.

Notice that

$$\mathbb{E}_0 \left[ \sum_{(\widehat{\boldsymbol{\theta}}_1,\ldots,\widehat{\boldsymbol{\theta}}_p) \in [d]^p} \sum_{t=1}^{T} \prod_{i=1}^{p} (\widehat{\boldsymbol{\theta}}_i^\top \boldsymbol{a}^{(t)}) \right] = \mathbb{E}_0 \left[ \sum_{(j_1,\ldots,j_p) \in [d]^p} \sum_{t=1}^{T} \prod_{i=1}^{p} (a_{j_i}^{(t)}) \right]$$

$$= \mathbb{E}_0 \left[ \sum_{t=1}^{T} \prod_{i=1}^{p} (\sum_{j=1}^{d} a_j^{(t)}) \right]$$

$$\leq \mathbb{E}_0 \left[ \sum_{t=1}^{T} \prod_{i=1}^{p} \|\boldsymbol{a}^{(t)}\|_1 \right]$$

$$\leq d^{p/2} T$$

and

$$\mathbb{E}_0 \left[ \sum_{(\widehat{\boldsymbol{\theta}}_1,\ldots,\widehat{\boldsymbol{\theta}}_p) \in [d]^p} \sum_{t=1}^{T} \prod_{i=1}^{p} (\widehat{\boldsymbol{\theta}}_i^\top \boldsymbol{a}^{(t)})^2 \right] = \mathbb{E}_0 \left[ \sum_{(j_1,\ldots,j_p) \in [d]^p} \sum_{t=1}^{T} \prod_{i=1}^{p} (a_{j_i}^{(t)})^2 \right]$$

$$= \mathbb{E}_0 \left[ \sum_{t=1}^{T} \prod_{i=1}^{p} \|\boldsymbol{a}^{(t)}\|_2^2 \right]$$

$$\leq T$$

where we used $\|\boldsymbol{a}^{(t)}\|_2 \leq 1, \forall t \in [T]$. Therefore plugging back we have

$$\mathfrak{R}(d,p,T) \geq \frac{\Delta^p}{d^p} \left( \frac{d^p T}{p^{\frac{p}{2}}} - d^{\frac{p}{2}} T - T d^{\frac{p}{2}} \Delta^p \sqrt{T} \right)$$

and finally letting $\Delta^p = \sqrt{\frac{d^p}{4Tp^p}}$ leads to

$$\mathfrak{R}(d,p,T) \geq O(\sqrt{d^p T}/p^p).$$

$\square$

**Remark G.1.** *Better result $O(\sqrt{d^p T})$ holds for bandits $r = \prod_{i=1}^{p} (\boldsymbol{\theta}_i^\top \boldsymbol{a}_i) + \epsilon$ where $\boldsymbol{a}_i \in \mathbb{R}^d, \|\boldsymbol{a}_i\|_2 \leq 1$.*

For completeness, we show the proof of the above remark.

*Proof.* We overload the notation and use $[d]$ to denote the set $\{e_1, e_2, \ldots, e_d\}$. The hard instances are chosen in $\Delta \cdot [d]^p$, i.e. $(\boldsymbol{\theta}_1,\ldots,\boldsymbol{\theta}_p) = \Delta \cdot (\widehat{\boldsymbol{\theta}}_1,\ldots,\widehat{\boldsymbol{\theta}}_p)$ where $(\widehat{\boldsymbol{\theta}}_1,\ldots,\widehat{\boldsymbol{\theta}}_p) \in [d]^p$. We use $\boldsymbol{a}_i^{(t)} \in \mathbb{R}^d$ to denote the $i$-th action in $t$-th episode, where $i \in [p], t \in [T]$.

We use $\mathbb{P}_{(\boldsymbol{\theta}_1,\ldots,\boldsymbol{\theta}_p)}$ to indicate the measure on outcomes induced by the interaction of the fixed policy and the bandit paramterised by $r = \prod_{i=1}^p(\boldsymbol{\theta}_i^\top \boldsymbol{a}_i) + \epsilon$. Specifically, We use $\mathbb{P}_0$ to indicate the measure on outcomes induced by the interaction of the fixed policy and the pure noise bandit $r = \epsilon$.

$$\mathfrak{R}(d,p,T)$$

$$\geq \frac{1}{d^p} \sum_{(\boldsymbol{\theta}_1,\ldots,\boldsymbol{\theta}_p)\in\Delta\cdot[d]^p} \mathbb{E}_{(\boldsymbol{\theta}_1,\ldots,\boldsymbol{\theta}_p)}\left[T\Delta^p - \sum_{t=1}^T\prod_{i=1}^p(\boldsymbol{\theta}_i^\top \boldsymbol{a}_i^{(t)})\right]$$

$$= \frac{\Delta^p}{d^p} \sum_{(\boldsymbol{\theta}_1,\ldots,\boldsymbol{\theta}_p)\in\Delta\cdot[d]^p} \left(T - \mathbb{E}_{(\boldsymbol{\theta}_1,\ldots,\boldsymbol{\theta}_p)}\left[\sum_{t=1}^T\prod_{i=1}^p(\widehat{\boldsymbol{\theta}}_i^\top \boldsymbol{a}_i^{(t)})\right]\right)$$

$$\geq \frac{\Delta^p}{d^p} \sum_{(\boldsymbol{\theta}_1,\ldots,\boldsymbol{\theta}_p)\in\Delta\cdot[d]^p} \left(T - \mathbb{E}_0\left[\sum_{t=1}^T\prod_{i=1}^p(\widehat{\boldsymbol{\theta}}_i^\top \boldsymbol{a}_i^{(t)})\right] - T\|\mathbb{P}_0 - \mathbb{P}_{(\boldsymbol{\theta}_1,\ldots,\boldsymbol{\theta}_p)}\|_{\mathrm{TV}}\right)$$

$$\geq \frac{\Delta^p}{d^p} \sum_{(\boldsymbol{\theta}_1,\ldots,\boldsymbol{\theta}_p)\in\Delta\cdot[d]^p} \left(T - \mathbb{E}_0\left[\sum_{t=1}^T\prod_{i=1}^p(\widehat{\boldsymbol{\theta}}_i^\top \boldsymbol{a}_i^{(t)})\right] - T\sqrt{D_{\mathrm{KL}}(\mathbb{P}_0\|\mathbb{P}_{(\boldsymbol{\theta}_1,\ldots,\boldsymbol{\theta}_p)})}\right)$$

$$\geq \frac{\Delta^p}{d^p} \sum_{(\boldsymbol{\theta}_1,\ldots,\boldsymbol{\theta}_p)\in\Delta\cdot[d]^p} \left(T - \mathbb{E}_0\left[\sum_{t=1}^T\prod_{i=1}^p(\widehat{\boldsymbol{\theta}}_i^\top \boldsymbol{a}_i^{(t)})\right] - T\sqrt{\Delta^{2p}\mathbb{E}_0\left[\sum_{t=1}^T\prod_{i=1}^p(\widehat{\boldsymbol{\theta}}_i^\top \boldsymbol{a}_i^{(t)})^2\right]}\right)$$

$$\geq \frac{\Delta^p}{d^p}\left(d^pT - \mathbb{E}_0\left[\sum_{(\widehat{\boldsymbol{\theta}}_1,\ldots,\widehat{\boldsymbol{\theta}}_p)\in[d]^p}\sum_{t=1}^T\prod_{i=1}^p(\widehat{\boldsymbol{\theta}}_i^\top \boldsymbol{a}_i^{(t)})\right] - Td^{\frac{p}{2}}\Delta^p\sqrt{\mathbb{E}_0\left[\sum_{(\widehat{\boldsymbol{\theta}}_1,\ldots,\widehat{\boldsymbol{\theta}}_p)\in[d]^p}\sum_{t=1}^T\prod_{i=1}^p(\widehat{\boldsymbol{\theta}}_i^\top \boldsymbol{a}_i^{(t)})^2\right]}\right)$$

where the first step comes from

$$\mathrm{Regret} \geq \mathbb{E}_{(\boldsymbol{\theta}_1,\ldots,\boldsymbol{\theta}_p)}\left[T\Delta^p - \sum_{t=1}^T\prod_{i=1}^p(\boldsymbol{\theta}_i^\top \boldsymbol{a}_i^{(t)})\right]$$

(the optimal action in hindsight is $\boldsymbol{a}_i = \widehat{\boldsymbol{\theta}}_i$); the second step comes from $(\boldsymbol{\theta}_1,\ldots,\boldsymbol{\theta}_p) = \Delta\cdot(\widehat{\boldsymbol{\theta}}_1,\ldots,\widehat{\boldsymbol{\theta}}_p)$ and algebra; the third step comes from $\left|\sum_{t=1}^T\prod_{i=1}^p(\widehat{\boldsymbol{\theta}}_i^\top \boldsymbol{a}_i^{(t)})\right| \leq T$; the fourth step comes from Pinsker's inequality; the fifth step comes from

$$D_{\mathrm{KL}}(\mathbb{P}_0\|\mathbb{P}_{\boldsymbol{\theta}_1,\ldots,\boldsymbol{\theta}_p}) = \mathbb{E}_0\left[\sum_{t=1}^T D_{\mathrm{KL}}\left(N(0,1)\|N(\prod_{i=1}^p(\boldsymbol{\theta}_i^\top \boldsymbol{a}_i^{(t)}),1)\right)\right]$$

$$= \Delta^{2p}\mathbb{E}_0\left[\sum_{t=1}^T\prod_{i=1}^p(\widehat{\boldsymbol{\theta}}_i^\top \boldsymbol{a}_i^{(t)})^2\right]$$

and the final step comes from Jensen's inequality and algebra.

Notice that

$$\mathbb{E}_0\left[\sum_{(\widehat{\boldsymbol{\theta}}_1,\ldots,\widehat{\boldsymbol{\theta}}_p)\in[d]^p}\sum_{t=1}^T\prod_{i=1}^p(\widehat{\boldsymbol{\theta}}_i^\top \boldsymbol{a}_i^{(t)})\right] = \mathbb{E}_0\left[\sum_{(j_1,\ldots,j_p)\in[d]^p}\sum_{t=1}^T\prod_{i=1}^p\left((\boldsymbol{a}_i^{(t)})_{j_i}\right)\right]$$

$$= \mathbb{E}_0\left[\sum_{t=1}^T\prod_{i=1}^p\left(\sum_{j=1}^d(\boldsymbol{a}_i^{(t)})_j\right)\right]$$

$$\leq \mathbb{E}_0\left[\sum_{t=1}^T\prod_{i=1}^p\|\boldsymbol{a}_i^{(t)}\|_1\right]$$

$$\leq d^{p/2}T$$

and

$$\mathbb{E}_0\left[\sum_{(\widehat{\boldsymbol{\theta}}_1,\ldots,\widehat{\boldsymbol{\theta}}_p)\in[d]^p}\sum_{t=1}^T\prod_{i=1}^p(\widehat{\boldsymbol{\theta}}_i^\top \boldsymbol{a}_i^{(t)})^2\right] = \mathbb{E}_0\left[\sum_{(j_1,\ldots,j_p)\in[d]^p}\sum_{t=1}^T\prod_{i=1}^p\left((\boldsymbol{a}_i^{(t)})_{j_i}\right)^2\right]$$

$$= \mathbb{E}_0\left[\sum_{t=1}^T\prod_{i=1}^p\|\boldsymbol{a}_i^{(t)}\|_2^2\right]$$

$$\leq T$$

where we used $\|\boldsymbol{a}_i^{(t)}\|_2 \leq 1, \forall t \in [T]$. Therefore plugging back we have

$$\mathfrak{R}(d,p,T) \geq \frac{\Delta^p}{d^p}\left(d^p T - d^{\frac{p}{2}}T - Td^{\frac{p}{2}}\Delta^p\sqrt{T}\right)$$

and finally letting $\Delta^p = \sqrt{\frac{d^p}{4T}}$ leads to

$$\mathfrak{R}(d,p,T) \geq O(\sqrt{d^p T}).$$

$\square$

We present the proof of Theorem 3.19 in the following.

*Proof.* Denote the optimal action in hindsight as $\boldsymbol{a}^* = \mathrm{supp}(\boldsymbol{\theta}_1,\ldots,\boldsymbol{\theta}_p)/\sqrt{p}$. From the proof of Theorem 3.18 we know that if $T \leq \frac{1}{4p^p}\cdot\frac{d^p}{\Delta^{2p}}$, then

$$\frac{1}{d^p}\sum_{(\boldsymbol{\theta}_1,\ldots,\boldsymbol{\theta}_p)\in\Delta\cdot[d]^p}\mathbb{E}_{(\boldsymbol{\theta}_1,\ldots,\boldsymbol{\theta}_p)}\left[\prod_{i=1}^p(\boldsymbol{\theta}_i^\top \boldsymbol{a}^*) - \prod_{i=1}^p(\boldsymbol{\theta}_i^\top \boldsymbol{a}^{(t)})\right]$$

$$\geq \frac{\Delta^p}{d^p}\left(\frac{d^p}{p^{\frac{p}{2}}} - d^{\frac{p}{2}} - d^{\frac{p}{2}}\Delta^p\sqrt{T}\right)$$

$$\geq \frac{\Delta^p}{4p^{\frac{p}{2}}}$$

$$\geq \frac{1}{4}\cdot\frac{1}{d^p}\sum_{(\boldsymbol{\theta}_1,\ldots,\boldsymbol{\theta}_p)\in\Delta\cdot[d]^p}\mathbb{E}_{(\boldsymbol{\theta}_1,\ldots,\boldsymbol{\theta}_p)}\left[\prod_{i=1}^p(\boldsymbol{\theta}_i^\top \boldsymbol{a}^*)\right]$$

which indicates the following

$$\inf_\pi \sup_{(\boldsymbol{\theta}_1,\ldots,\boldsymbol{\theta}_p)}\mathbb{E}_{(\boldsymbol{\theta}_1,\ldots,\boldsymbol{\theta}_p)}\left[\frac{3}{4}\cdot\prod_{i=1}^p(\boldsymbol{\theta}_i^\top \boldsymbol{a}^*) - \prod_{i=1}^p(\boldsymbol{\theta}_i^\top \boldsymbol{a}^{(t)})\right] \geq 0.$$

$\square$

### G.0.1  Lower Bounds with UCB Algorithms

In this subsection, we construct a hard bandit problem where the rewards are noiseless degree-$p$ polynomial, and show that any UCB algorithm needs at least $\Omega(d^p)$ actions to learn the optimal action. On the contrary, Theorem 3.21 shows that by playing actions randomly, we only need $2(dk + (p+1)^p) = O(d)$ actions.

**Hard Case Construction**  Let $\boldsymbol{e}_i$ denotes the $i$-th standard orthonormal basis of $\mathbb{R}^d$, i.e., $\boldsymbol{e}_i$ has only one 1 at the $i$-th entry and 0's for other entries. We define a $p$-th multi-indices set $\Lambda$ as $\Lambda = \{(\alpha_1,\ldots,\alpha_p)|1 \leq \alpha_1 < \cdots < \alpha_p \leq d\}$. For an $\alpha = (\alpha_1,\ldots,\alpha_p) \in \Lambda$, denote $\boldsymbol{M}_\alpha = \boldsymbol{e}_{\alpha_1}\otimes\cdots\otimes\boldsymbol{e}_{\alpha_p}$. Then the model space $\mathcal{M}$ is defined as $\mathcal{M} = \{\boldsymbol{M}_\alpha|\alpha \in \Lambda\}$, which is a subset of rank-1 $p$-th order tensors. The action set $\mathcal{A}$ is defined as $\mathcal{A} = \mathrm{conv}(\{\boldsymbol{e}_{\alpha_1} + \cdots + \boldsymbol{e}_{\alpha_p}|\alpha \in \Lambda\})$. Assume that the ground-truth parameter is $\boldsymbol{M}^* = \boldsymbol{M}_{\alpha^*} \in \mathcal{M}$. The noiseless reward $r_t = r(\boldsymbol{M}^*, \boldsymbol{a}_t) = \langle\boldsymbol{M}^*, (\boldsymbol{a}_t)^{\otimes p}\rangle = \prod_{i=1}^p\langle\boldsymbol{e}_{\alpha_i^*}, \boldsymbol{a}_t\rangle$ is a polynomial of $\boldsymbol{a}_t$ and falls into the case of Example 3.23.

**UCB Algorithms**   The UCB algorithms sequentially maintain a confidence set $\mathcal{C}_t$ after playing actions $\boldsymbol{a}_1, \ldots, \boldsymbol{a}_t$. Then UCB algorithms play $\boldsymbol{a}_{t+1} \in \arg\max_{\boldsymbol{a} \in \mathcal{A}} \mathrm{UCB}_t(\boldsymbol{a})$, where $\mathrm{UCB}_t(\boldsymbol{a}) = \max_{\boldsymbol{M} \in \mathcal{C}_t} \langle \boldsymbol{M}, (\boldsymbol{a})^{\otimes p} \rangle$.

**Theorem G.2.** *Assume that for each $t \geq 0$, the confidence set $\mathcal{C}_t$ contains the ground-truth model, i.e., $\boldsymbol{M}^* \in \mathcal{C}_t$. Then for the noiseless degree-$p$ polynomial bandits, any UCB algorithm needs to play at least $\binom{d}{p} - 1$ actions to distinguish models in $\mathcal{M}$. Furthermore, the worst-case cumulative regret at round $T$ can be lower bounded by*

$$\mathfrak{R}(T) \geq \min\{T, \binom{d}{p} - 1\}.$$

Theorem G.2 shows the failure of the optimistic mechanism, which forbids the algorithm to play an informative action that is known to be of low reward for all models in the confidence set. On the contrary, the reward function class falls into the form of $q(\boldsymbol{U}\boldsymbol{a})$, therefore, by playing actions randomly[6], we only need $O(d)$ actions as Theorem 3.21 suggests.

---

[6]Careful readers may notice that $\mathcal{A}$ is of measure zero in this setting. However, since the reward function is a homogenous polynomial of degree $p$, we can actually obtain the rewards on $\mathrm{conv}(\mathcal{A}, \boldsymbol{0})$, which is of positive measure.