# OpenReview forum: "Optimal Gradient-based Algorithms for Non-concave Bandit Optimization"
_NeurIPS.cc/2021/Conference — NeurIPS 2021 Poster_

### Official Review · Reviewer_zjCr · 2021-07-16

**Rating:** 8
**Confidence:** 3

**Summary:**

This paper studied a fundamental question in non-linear bandit: suppose the mean reward function f_\theta in the model r_t = f_\theta(a_t) + noise is non-linear or only linear after some high-dimensional reparametrization of \theta, is there a good bandit algorithm to find near-optimal policies with sample complexities dictated by the intrinsic dimension? In this paper, the authors investigated the following examples:

1. low-rank linear reward;
2. bilinear reward (or RL with quadratic Bellman-complete Q-function);
3. homogeneous polynomial reward;
4. noiseless polynomial reward.

In first three examples, the zeroth-order gradient ascent algorithm achieves a PAC sample complexity with the optimal dependence on the dimension d, strictly better than the upper bound given by LinUCB when viewed as a generalized linear model, as well as the eluder dimension argument. In the last example, there is an instance where solving polynomial equations after random sampling only has sample complexity O(d), but any UCB algorithm incurs a sample complexity O(d^p).

**Main Review:**

This is a really nice paper. In my opinion, the authors look into an important problem in bandits, and obtain satisfactory answers to several important examples. The key message that for non-linear rewards, one can do much better than LinUCB or eluder dimension arguments, is an important addition to the bandit literature, and provides useful tools to study future non-linear bandits. It is also a nice discovery that the zeroth-order gradient ascent could achieve the optimal dimension dependence in several examples.

If I have to list some limitations:

1. The applicable range of the zeroth-order gradient ascent remains unclear - in the examples where it works, this method typically turns into other known methods such as noisy subspace iteration or power methods. Also, the analysis tools are pretty ad-hoc, so the possibility that this tool is only tailored for specific problems such as sparsity or PCA cannot be ruled out.

2. The noiseless polynomial reward example seems to be disparate from other examples. First the noise-free property is crucial in solving polynomial equations; second only random sampling is needed. Of course this is a really nice example to provably establish the separation compared to UCB, but I am not sure how much intuition is provided here.

3. The dependence of the sample complexity on other parameters (k, \varepsilon) is often not tight, leaving interesting questions on whether new algorithms should be used.

Some additional comments:

1. An interesting observation is that, in Table 1, the improvement on d compared with the LinUCB/eluder upper bound is always linear in d. I would greatly appreciate it if there is some intuitive explanations.

2. Line 122: argmax should be max.

3. Line 303: I think a factor of p is missing in the characterization of dim(R_V).

Post-rebuttal feedback:

I thank the authors for the response and I also looked at other reviewers' comments. I retain my opinion that this paper is a valuable contribution to the bandit community and studies a fundamental question in contextual bandits, and therefore my ratings.

**Time Spent Reviewing:**

3

---

> ### Author Response · Authors · 2021-08-10
> **Response to Reviewer zjCr**
>
> Thank you for your valuable feedback. We address the questions and comments in the following.
>
> *1: ``The applicable range of the zeroth-order gradient ascent remains unclear. In the examples where it works, this method typically turns into other known methods such as noisy subspace iteration or power methods"*
>
> *A:* We agree it is a fundamental problem to determine when the adaptations of zeroth-order gradient methods result in optimal algorithms. In our paper, we partially answer the question, in the sense that for all our settings where optimization can achieve linear or super-linear convergence (potentially after careful initialization),  we can find algorithms with optimal dependence on the problem dimension by adapting the zeroth-order gradient methods.
>
> The reviewer is correct that for the matrix problems, our algorithm turns into the power method or the noisy subspace iteration. However, this is not the case for $p\ge 3$, since the approximate gradient direction is biased in the tensor power method update, and simply bounding the bias as done in a standard zeroth-order gradient analysis would not suffice. We utilize the bias and show that it actually helps the algorithm.
>
> *2: "The noiseless polynomial reward example seems to be disparate from other examples."*
>
> *A:* In both the noisy and noiseless case, we settle the optimal dependence on dimension $d$. This is why we study both cases.
>
> *3: ``An interesting observation is that, in Table 1, the improvement on d compared with the LinUCB/eluder upper bound is always linear in d. I would greatly appreciate it if there is some intuitive explanations."*
>
> *A:* This is an interesting observation. We have shown that our results meet the lower bound, where the dependence is $d^p$. It happens that the room of improvement can be at most $O(d)$ compared to existing works.

---

### Official Review · Reviewer_VuTE · 2021-07-16

**Rating:** 5
**Confidence:** 1

**Summary:**

This work proposed algorithms based on zeroth-order optimization scheme for structure polynomial reward function in multi armed bandit problems. Solid theoretical results are presented in the paper.




**Limitations And Societal Impact:**

No  negative societal impact.

**Main Review:**

This work proposed algorithms based on zeroth-order optimization scheme for structure polynomial reward function in multi armed bandit problems. Solid theoretical results are presented in the paper.

The paper is difficult to understand.

1.	In the contribution statement, the authors say “Rather than looking at the expected regret, we are concerned with the PAC bounds on the sample complexity to identify an _-optimal arm”. Why PAC is adopted instead of expected regret.
2.	The reviewer failed to understand the main results. The reason may be twofold: first, the reviewer has a weak background on this topic; and second, his paper presents many theoretical results, which are lack of explanation. The theoretical analysis may be sound, but the reviewer suggest that the paper present the results from a higher level, or be more concentrated on some main points, so the paper is less heavy and more friendly to the readers.


**Time Spent Reviewing:**

8

---

> ### Author Response · Authors · 2021-08-10
> **Response to Reviewer Reviewer VuTE**
>
> Thank you for your valuable feedback. Our paper studies many fundamental questions in bandit problems, and proposes sample-efficient algorithms with a matching lower bound. Part of our results also refute the conjectures from [28] and [35] on the dependence of $d$ for eigenvector bandit problems and the low-rank linear reward case. We provide the response to the key questions from the reviewer and respectfully hope the reviewer would reconsider his/her score.
>
> *1: "Why PAC is adopted instead of expected regret "*
>
> A: Both PAC and expected regret are standard criteria to evaluate the sample complexity to achieve good performance. There are PAC-regret conversion arguments that can reduce the PAC results to regret results (and vice versa). In the revised version, we will also include the expected regret bounds. Thank you for your suggestion.
>
> *2: " the reviewer suggest that the paper present the results from a higher level"*
>
> A: We will add more intuitions in the revised version. For instance, in the rank-1 case ($r(\mathbf{a}) = (\mathbf{a}^T \theta^*)^2 +\eta$), former results [28,35] suffer from an $O(d^3)$ sample complexity, and in the revised version we will add a paragraph to demonstrate how we get beyond the $O(d^3)$ barrier from a more high-level way.
>
> A random action $\mathbf{a} \sim \text{Unif}( \mathbf{S}^{d-1}) $ in expectation has signal $f(\mathbf{a}) \asymp {1/d^2}$, and the noise has standard deviation $O(1)$. Thus the signal-to-noise-ratio is $O(1/d^2)$. Also, the optimal action $\theta^*$ requires $d$ bits to encode. Therefore, if we were to play non-adaptively, we would require $O(d^3)$ queries which matches the results of [28][35].
>
> To go beyond former results, we must design algorithms that are adaptive, meaning that the information in one query of $f(\mathbf{a}) +\eta$ is **strictly larger than** $1/d^2$. As an illustration of why this is possible, consider batching the time-steps into $d$ stages so that in each stage the queries decode $1$ bit more of $\theta^\ast$. At the first stage, random exploration $\mathbf{a} \sim \text{Unif}( \mathbf{S}^{d-1}) $ gives signal-to-noise-ratio $O(1/d^2)$ so we need $O(d^2)$ queries. Suppose $k$ bits of $\theta$ are decoded at $k$-th stage by $\hat \theta$, adaptive algorithms can boost the signal-to-noise-ratio to $O(k/d^2)$ by using $\hat\theta$ as bootstrap (e.g. exploring with $\hat \theta \pm a$ where $a$ is random in the unexplored subspace). In this way adaptive algorithms only need $O(d^2/k)$ queries in ($k+1$)-th stage and so the total number of queries sums up to $\sum_{k=1}^d d^2/k \approx d^2 \log d$ that improves over $O(d^3)$.
>
>
> Gradient descent and power method offer a computationally efficient and seamless way to implement the above intuition. For the iterating action $\mathbf{a}$, we estimate $\mathbf{M} \mathbf{a}$ from noisy observations and take it as our next action $\mathbf{a}^+$. With $d^2/(\Delta^2\epsilon^2)$ samples in each iteration, noisy power method enjoys linear progress. Therefore even though each iteration costs $O(d^2)$ samples, overall we only need $O(\log(d))$ iterations to find an $\epsilon$-optimal action.
>
> References:
>
> [28]Kwang-Sung Jun, Rebecca Willett, Stephen Wright, and Robert Nowak. Bilinear bandits with low-rank structure. In International Conference on Machine Learning, pages 3163–3172. PMLR, 2019.
>
> [35]Yangyi Lu, Amirhossein Meisami, and Ambuj Tewari. Low-rank generalized linear bandit problems. In International Conference on Artificial Intelligence and Statistics, pages 460–468. PMLR, 2021.

---

### Official Review · Reviewer_FgrR · 2021-07-17

**Rating:** 7
**Confidence:** 2

**Summary:**

This paper studies the zeroth-order optimization methods for the nonlinear bandit problem. It studies PAC bounds for bandit problems where the reward function is structured polynomials, including a low-rank generalized linear bandit problem and a two-layer neural network with polynomial activations bandit problem. In the low-rank generalized linear bandit setting, the authors provide a minimax optimal algorithm improving the results in [28, 35]. They further apply their algorithms in the RL with the generative model setting, achieving improved sample complexity. In the neural net setting, they propose a bandit algorithm that has sample complexity equal to the intrinsic algebraic dimension.
They also show that the standard optimistic algorithms are sub-optimal for both settings.


**Limitations And Societal Impact:**

The authors adequately addressed the limitations and potential negative societal impact of their work.

**Main Review:**

In general, this work is well organized and the results as well as the related comparisons are clearly presented.

The theoretical contributions of this work are significant. The authors not only provide the near-optimal upper bounds and the lower bounds, but also shows the analysis of the sub-optimality of the optimistic algorithms.

The basic idea of this work is based on the noisy power iteration and conversion of the zeroth order information to the function gradient by perturbation noise. However, it is interesting to see that this idea can be applied to the structured nonlinear bandit problem and can further improve the existing results.

Currently, I do not find any major weakness in this work. There are some minor comments:
1. The details of the parameter setting are missing in Algorithm 1, which makes it look too simplified. It would be clearer if the authors can move Theorem B.2 to the main text.
2. In the Line 13 of Algorithm 2, the left half of parenthesis is missing in the equation
3. Can the authors provide some intuitions or point out the main techniques that lead to the improvement of the existing results in [28,35] by the noisy power iteration?


**Time Spent Reviewing:**

5

---

> ### Author Response · Authors · 2021-08-10
> **Response to Reviewer FgrR**
>
> We thank the reviewer for the valuable feedback and we will fix all the typos and make clear the algorithm details in the revision. We hope given these clarifications you will consider increasing your score.
>
> *1: "Can the authors provide some intuitions or point out the main techniques that lead to the improvement of the existing results in [28,35] by the noisy power iteration?"*
>
> *A:* To demonstrate it, we provide an information-theory-based analysis for the rank-1 case: $r(\mathbf{a}) = (\mathbf{a}^T \theta^*)^2 +\eta$. (In this case, results in [28,35] suffer from an $O(d^3)$ sample complexity as well. )
>
> A random action $\mathbf{a} \sim \text{Unif}( \mathbf{S}^{d-1}) $ in expectation has signal $f(\mathbf{a}) \asymp {1/d^2}$, and the noise has standard deviation $O(1)$. Thus the signal-to-noise-ratio is $O(1/d^2)$. Also, the optimal action $\theta^*$ requires $d$ bits to encode. Therefore, if we were to play non-adaptively, we would require $O(d^3)$ queries which matches the results of [28][35].
>
> To go beyond former results, we must design algorithms that are adaptive, meaning that the information in one query of $f(\mathbf{a}) +\eta$ is **strictly larger than** $1/d^2$. As an illustration of why this is possible, consider batching the time-steps into $d$ stages so that in each stage the queries decode $1$ bit more of $\theta^\ast$. At the first stage, random exploration $\mathbf{a} \sim \text{Unif}( \mathbf{S}^{d-1}) $ gives signal-to-noise-ratio $O(1/d^2)$ so we need $O(d^2)$ queries. Suppose $k$ bits of $\theta$ are decoded at $k$-th stage by $\hat \theta$, adaptive algorithms can boost the signal-to-noise-ratio to $O(k/d^2)$ by using $\hat\theta$ as bootstrap (e.g. exploring with $\hat \theta \pm a$ where $a$ is random in the unexplored subspace). In this way adaptive algorithms only need $O(d^2/k)$ queries in ($k+1$)-th stage and so the total number of queries sums up to $\sum_{k=1}^d d^2/k \approx d^2 \log d$ that improves over $O(d^3)$.
>
> Gradient descent and power method offer a computationally efficient and seamless way to implement the above intuition. For the iterating action $\mathbf{a}$, we estimate $\mathbf{M} \mathbf{a}$ from noisy observations and take it as our next action $\mathbf{a}^+$. With $d^2/(\Delta^2\epsilon^2)$ samples in each iteration, noisy power method enjoys linear progress. Therefore even though each iteration costs $O(d^2)$ samples, overall we only need $O(\log(d))$ iterations to find an $\epsilon$-optimal action.
>
>
> References:
>
> [28]Kwang-Sung Jun, Rebecca Willett, Stephen Wright, and Robert Nowak. Bilinear bandits with low-rank structure. In International Conference on Machine Learning, pages 3163–3172. PMLR, 2019.
>
> [35]Yangyi Lu, Amirhossein Meisami, and Ambuj Tewari. Low-rank generalized linear bandit problems. In International Conference on Artificial Intelligence and Statistics, pages 460–468. PMLR, 2021.

---

### Official Review · Reviewer_3PPq · 2021-07-22

**Rating:** 8
**Confidence:** 4

**Summary:**

The paper studies nonlinear bandit optimization with structured polynomial reward functions, with a focus on the pure exploration setting. The paper proposes some stochastic zeroth-order gradient ascent algorithms with phase elimination and show that they attain minimax sample complexity (in terms of $1/\epsilon$ and the dimension $d$) for a class of structured polynomial reward functions (i.e., stochastic low-rank linear reward function, stochastic bilinear reward function, stochastic bilinear reward reward function). The sample complexity bounds usually improve upon the bounds obtained via existing techniques (i.e., LinUCB and eluder dimension-based arguments) by a $O(d)$ factor. Combined with matching lower bounds, these results help to reveal the right role of $d$ in the minimax complexity of the considered problems. The paper also shows the suboptimality of UCB algorithms (in terms of $d$) by proving a $\Omega(d^p)$-type lower bound for UCB in the noise-less polynomial reward case (for which solving polynomial equations usually leads to $O(d)$-type sample complexity).

**Limitations And Societal Impact:**

I have pointed out some places where the authors can make some improvement in terms of presentation.

**Main Review:**

*In the discussion phase, the area chair raised the issue that the presentation of the paper may require some improvement. I agree with this point and add some suggestions at the end of my review.*

The paper makes important contributions to nonlinear bandit optimization by illustrating the optimality of zeroth order gradient ascent for several natural structured polynomial reward functions. While existing techniques (i.e., LinUCB and eluder dimension-based arguments) are usually able to provide sample complexity that is tight in terms of $1/\epsilon$, the algorithms and techniques introduced in this paper provide improved sample complexity that is usually better by a $O(d)$ factor. Such improvements also lead to non-trivial consequences for simulator-based RL and phase retrieval. In the noise-less polynomial reward case, the paper gives an interesting result: UCB may suffer from $\Omega(d^p)$-type sample complexity, while solving polynomial equations may enable $O(d)$-type sample complexity. Overall, the results in this paper may lead to deeper understanding of nonlinear bandit optimization (in the stochastic and pure exploration case).

I have some questions/comments on the paper.
1. In line 53-54, the authors say that "Our algorithms are also computationally efficient, bringing more practical advantages." However, there is no explicit description of the computational complexity of the algorithm. Can the authors provide such descriptions (even informal)? In particular, a comparison with existing algorithms from a computational perspective would be very helpful.
2. The shorthand "NPM" appears in Table 1 without any explanation, which may cause confusion (I see that there is an explanation under Table 2, but that appears after Table 1).
3. Table 1 list results in the order of HPS, HPA, B, LL, which is different from the order of presentation. Maybe it is better to make them consistent?
4. For the case of HPS, one has a gap-dependent bound for even $p$ and a gap-free bound for odd $p$. Can the authors provide some insights on why even $p$ and odd $p$ are fundamentally different? For example, why is it difficult to obtain gap-free bound for even $p$?
5. Line 214: "Let ... is defined as:" should be changed to "Let ... be defined as:".
6. Line 238-239: Can the authors explain why the techniques in this paper can only extended to simulator-based RL? What is the difficulty to extend these techniques to online RL?
7. Line 255-256: "We now the bandit optimization procedure" --- something is missing.
8. Footnote 1: Why does this footnote appear under Corollary 3.12, but not in a place where this notation first appears?
9. Section 3.4.1: Is the procedure of solving polynomial equations computationally tractable ? If not, this should be clearly mentioned. Also, how would zeroth order gradient ascent performs in this case? Something needs to be explained here (e.g., how the results can be related to zeroth order gradient ascent); otherwise, Section 3.4 seems quite isolated from other parts.

Moreover, the paper's presentation can be improved. I have the following suggestions.
1. As some reviewers have pointed out, the paper should spend more time providing high-level intuition at the beginning. The current introduction comes in a very technical way, with many terminologies directly appearing without further explanations (e.g., phase retrieval, eluder dimension). If the authors want to directly use these technical terms at the beginning, then some background of phase retrieval and some intuition on the eluder dimension may be helpful.
2. The current Section 2 uses a quite non-standard structure --- it merges the problem setting (which is technical) and the paper outline (which needs to be clear and easy to find) together. I am fine with this structure, but is there any way to make it clearer?
3. The two tables in the paper are difficult to understand without additional efforts. I have pointed out two issues in my review:
- The shorthand "NPM" appears in Table 1 without explanations, which may cause confusion.
- Table 1 lists results in the order of HPS, HPA, B, LL, which is different from the order of presentation.
4. Section 3 contain lots of lemmas, theorems, definitions and assumptions. The presentation is split by multiple "paragraph titles", which makes the flow fragmented. My personal suggestion is to remove Section 3.2.1 (RL with simulator) from the main text and defer it to the appendix, as it is almost impossible to clearly explain the RL setting (which involves many concepts that require familiarity of the field, e.g., Bellman complete) in such a small space, and the RL setting departs from the nonlinear bandit setting which is more central in the paper. I also find the RL results to be extensions of the bandit results, so I feel that they can be deferred to the appendix.
5. As another reviewer and I have pointed out, Section 3.4 seems to be quite isolated from other parts in the main paper, and the results are of somehow different favors (the setting is noiseless, and the upper bound is provided by a computational-inefficient polynomial solving procedure rather than gradient ascent). It is also possible to defer this part to the appendix, but I do not have a strong opinion because the results seem to be also interesting.

After some removal of non-essential results, the authors should be able to save quite some space, and they can use such space to add clear explanations to their key results, as well as providing more high-level intuition. I hope that the authors can improve the presentation such that readers can better appreciate the theoretical contributions of the paper.





**Time Spent Reviewing:**

6

---

> ### Author Response · Authors · 2021-08-10
> **Response to Reviewer 3PPq**
>
> Thank you very much for the positive feedback and constructive suggestions. We will fix all the typos and grammar problems in the revised version.
>
> *1: “Add computational comparisons.”*
>
> *A:* Thank you for the great suggestion. We will add the computational complexity results in the revised version. From our algorithms, we can see that the computational complexity is at most $d$ times the respective sample complexity. This demonstrates the polynomial time complexity for our methods. By contrast, the existing LinUCB algorithms are not computationally tractable with polynomial computational complexity [1].
>
>
> *2: “Intuition for gap-dependence for even p and gap-independence for odd p.”*
>
> *A:* For a $p$-th order polynomial, our zeroth-order optimization is approximately equivalent to combining several different orders of (noisy) tensor product: starting from the $p$-th order tensor product, plus $(p-2)$-th order, $(p-4)$-th order, and so on. For odd $p$, the last term (of least order) will be the third-order tensor product, while for even $p$, the last term will be a matrix-vector product. In the current proof for even $p$, the assessment of the last term follows from the gap-dependent matrix power method analysis, which becomes a bottleneck.
>
> In the revision, we will provide gap-independent proofs for both odd and even $p$. The new results are attainable because we recently improved our results by showing that the higher orders of tensor product already make sufficient progress in each optimization step, and the matrix-vector product will not be a bottleneck in our analysis. Thus we are now able to achieve gap-independent results for all orders of $p\ge 3$.
>
> *3: “Why the paper can only extend to simulator-based RL instead of online RL”*
>
> *A:* We can easily apply our algorithms to simulator-based RL since one can choose any state-action pair as a sample at each time step. However, for online RL, we do not yet know how to freely sample the state-action pair to ensure that the feature map $\phi(s,a)$ follows an isotropic distribution, which is needed to estimate the gradient.
>
> *4: "Is the procedure of solving polynomial equations computationally tractable?"*
>
> *A:* To our knowledge, solving general polynomial equations is not tractable (NP-complete for degree above 2), and we will mention this.
>
> References:
>
> [1] Shipra Agrawal. Recent advances in multiarmed bandits for sequential decision making. In Operations Research & Management Science in the Age of Analytics, pages 167–188. INFORMS, 2019

---

### Decision · Program_Chairs · 2021-09-27

**Decision:**

Accept (Poster)

**Comment:**

The reviewers are in agreement that this paper provides important fundamental results on non-linear bandit problems.  The only score below the acceptance threshold gives the lowest possible confidence, and was accordingly downweighted to form the final decision.

Despite the general positive assessment, all reviewers agreed that the paper could be quite difficult to read for many readers.  I ask that in the camera-ready version, the authors very carefully incorporate the reviewer suggestions on improving clarity.  Here are some excerpts from the reviewer discussion, in case they were missed in the reviews:
- "the current Section 2 uses a quite non-standard structure --- it merges the problem setting (which is technical) and the paper outline (which needs to be clear and easy to find) together. The notations are also quite complicated"
- "the two tables in the paper are difficult to understand without additional efforts. I have pointed out two issues in my review"
- ", Section 3 contain lots of lemmas, theorems, definitions and assumptions. The presentation is split by multiple "paragraph titles", which makes the flow fragmented"
- "Section 3.4 seems to be quite isolated from other parts in the main paper, and the results are of somehow different favors" [and could be moved to the appendix]
- "After some removal of non-essential results, the authors should be able to save quite some space, and they should use such space to add clear explanations to their key results, as well as providing more high-level intuition."

In addition, based on the reviewer discussion and my own reading, I will mention the following (these are just examples and are far from exhaustive):
- Terms like "eluder dimension" should always be defined clearly.  For terms that are uncommon but not as central to your paper, you can consider defining them in an appendix, and cross-referencing that appendix in the main text.
- Be careful with the consistency of explanations and notation, e.g., Line 146 doesn't quite match Line 8 of Algorithm 1, and T(.) in Algorithm 2 may not have been defined
- Consider adding more citations (and possibly mentioning the main differences) when similar ideas have appeared previously, e.g., phased elimination
- Be careful for English grammar issues, e.g., wording of Theorem 3.13